# Multi-tract multi-symptom relationships in pediatric concussion

**Guido I Guberman[1]\***, **Sonja Stojanovski[2,3]**, **Eman Nishat[2,3]**, **Alain Ptito[1]**, **Danilo Bzdok[4,5,6]**, **Anne L Wheeler[2,3]**, **Maxime Descoteaux[7,8]**

[1]Department of Neurology and Neurosurgery, Faculty of Medicine, McGill University, Montreal, Canada; [2]Department of Physiology, Faculty of Medicine, University of Toronto, Toronto, Canada; [3]Neuroscience and Mental Health, The Hospital for Sick Children, Toronto, Canada; [4]McConnell Brain Imaging Centre (BIC), Montreal Neurological Institute (MNI), Faculty of Medicine, McGill University, Montreal, Canada; [5]Department of Biomedical Engineering, Faculty of Medicine, School of Computer Science, McGill University, Montreal, Canada; [6]Mila - Quebec Artificial Intelligence Institute, Montreal, Canada; [7]Department of Computer Science, Université de Sherbrooke, Sherbrooke, Canada; [8]Imeka Solutions Inc, Sherbrooke, Canada

**\*For correspondence:**
guido.guberman@mail.mcgill.ca

## Abstract

**Background:** The heterogeneity of white matter damage and symptoms in concussion has been identified as a major obstacle to therapeutic innovation. In contrast, most diffusion MRI (dMRI) studies on concussion have traditionally relied on group-comparison approaches that average out heterogeneity. To leverage, rather than average out, concussion heterogeneity, we combined dMRI and multivariate statistics to characterize multi-tract multi-symptom relationships.

**Methods:** Using cross-sectional data from 306 previously concussed children aged 9–10 from the Adolescent Brain Cognitive Development Study, we built connectomes weighted by classical and emerging diffusion measures. These measures were combined into two informative indices, the first representing microstructural complexity, the second representing axonal density. We deployed pattern-learning algorithms to jointly decompose these connectivity features and 19 symptom measures.

**Results:** Early multi-tract multi-symptom pairs explained the most covariance and represented broad symptom categories, such as a general problems pair, or a pair representing all cognitive symptoms, and implicated more distributed networks of white matter tracts. Further pairs represented more specific symptom combinations, such as a pair representing attention problems exclusively, and were associated with more localized white matter abnormalities. Symptom representation was not systematically related to tract representation across pairs. Sleep problems were implicated across most pairs, but were related to different connections across these pairs. Expression of multi-tract features was not driven by sociodemographic and injury-related variables, as well as by clinical subgroups defined by the presence of ADHD. Analyses performed on a replication dataset showed consistent results.

**Conclusions:** Using a double-multivariate approach, we identified clinically-informative, cross-demographic multi-tract multi-symptom relationships. These results suggest that rather than clear one-to-one symptom-connectivity disturbances, concussions may be characterized by subtypes of symptom/connectivity relationships. The symptom/connectivity relationships identified in multi-tract multi-symptom pairs were not apparent in single-tract/single-symptom analyses. Future studies aiming to better understand connectivity/symptom relationships should take into account multi-tract multi-symptom heterogeneity.

**Funding:** Financial support for this work came from a Vanier Canada Graduate Scholarship from the Canadian Institutes of Health Research (G.I.G.), an Ontario Graduate Scholarship (S.S.), a Restracomp Research Fellowship provided by the Hospital for Sick Children (S.S.), an Institutional Research Chair in Neuroinformatics (M.D.), as well as a Natural Sciences and Engineering Research Council CREATE grant (M.D.).

## Editor's evaluation

This manuscript aims to address an important issue in the study of concussion: both the brain damage caused by concussion, as well as the behavioral symptoms that result vary widely across individuals. The study uses novel and interesting methods to relate multi-variate diffusion MRI data with multi-variate symptom-related data. The methods of analysis are sophisticated and well-executed and the results are quite interesting. The methods developed here could have a broad impact in their application to the many other neurological diseases that have heterogeneous outcomes.

## Introduction

Concussion afflicts approximately 600 per 100,000 individuals every year (*Cassidy et al., 2004*). It is associated with several psychiatric conditions, such as post-traumatic stress disorder and attention-deficit/hyperactivity disorder (ADHD) (*Orlovska et al., 2014*). Its incidence rate is rising in children and adolescents (*Zhang et al., 2016*), and compared to adult populations, the impact of concussions on pediatric brains is understudied (*Mayer et al., 2018*). Despite considerable funding devoted to clinical and basic research, no major advances in therapeutics have been achieved to date (*Kenzie et al., 2017*). A root cause of this stagnation appears to be a contradiction: while all concussions are treated equally in clinical trials and research studies, they are characterized by extensive heterogeneity in their pathophysiology, clinical presentation, symptom severity and duration (*Kenzie et al., 2017*; *Hawryluk and Bullock, 2016*). Concussion heterogeneity across patients has been identified as a major hurdle in advancing concussion care (*Kenzie et al., 2017*; *Hawryluk and Bullock, 2016*).

Due to shearing forces transmitted during injury, the brain's white matter is especially vulnerable to concussion (*Armstrong et al., 2016*; *Bigler and Maxwell, 2012*). Decades of research have studied white matter structure in individuals who sustain concussions. However, most studies continue to assume consistent, one-to-one structure/symptom relationships and employ traditional group comparisons (*Dodd et al., 2014*; *Hulkower et al., 2013*), averaging out the diffuse and likely more idiosyncratic patterns of brain structure abnormalities in favor of shared ones. Hence, the extant literature suggests that a large proportion of the clinical and research studies have not adequately accounted for clinical and neuropathological concussion heterogeneity.

To remedy this shortcoming, a growing number of studies aim to parse the clinical heterogeneity in concussions by algorithmically partitioning patients into discrete subgroups based on symptoms (*Langdon et al., 2020*; *Si et al., 2018*; *Yeates et al., 2019*). Other studies aim instead to account for heterogeneity in white matter structure alterations (*Stojanovski et al., 2019b*; *Taylor et al., 2020*; *Ware et al., 2017*). *Ware et al., 2017* built individualized maps of white matter abnormalities which revealed substantial inter-subject variability in traumatic axonal injury and minimal consistency of subject-level effects. *Taylor et al., 2020* computed a multivariate summary measure of white matter structure across 22 major white matter bundles which achieved better classification accuracy of concussed patients from healthy controls compared to single tract measures. Hence, studies have attempted to address heterogeneity in symptoms and in white matter structure across concussed patients.

However, white matter alterations due to concussion are diffuse and can elicit several symptoms that may interact with each other in complex and variable ways (*Kenzie et al., 2017*; *Hawryluk and Bullock, 2016*; *Iverson, 2019*). For instance, two individuals may suffer a concussion and develop sleep problems. The first may have damaged white matter tracts related to sleep/wakefulness control, whereas the second may have damaged tracts related to mood, causing depression-like symptoms, which include sleep problems. These two individuals will thus display a common symptom but will have overall different symptom profiles and different white matter damage profiles. This example illustrates

**eLife digest** Concussions can damage networks of connections in the brain. Scientists have spent decades and millions of dollars studying concussions and potential treatments. Yet, no new treatments are available or in the pipeline. A major reason for this stagnation is that no two concussions are exactly alike. People affected by concussions may have different genetic or socioeconomic backgrounds. The nature of the injury or how its effects change over time may also vary among people with concussions.

One central question facing scientists is whether there are multiple types of concussions. If so, what distinguishes them and what characteristics do they share. Some studies have looked at differences among subgroups of patients with concussions. But questions remain about whether – beyond differences between the patients – the brain injury itself differs and what impact that has on symptoms or patient trajectory.

To better characterize different types of concussion, Guberman et al. analyzed diffusion magnetic resonance imaging scans from 306 nine or ten-year-old children with a previous concussion. The children were participants in the Adolescent Brain Cognitive Development Study. Using specialized statistical techniques, the researchers outlined subgroups of concussions in terms of connections and symptoms and studied how many of these subgroups each patient had. Some types of injury were linked with a category of symptoms like cognitive, mood, or physical symptoms. Some types of damage were linked with specific symptoms. Guberman et al. also found that one symptom, sleep problems, was part of many different injury subtypes. Sleep problems may occur in different patients for different reasons. For example, one patient with sleep difficulties may have experienced damage in brain regions controlling sleep and wakefulness. Another person with sleep problems may have injured parts of the brain responsible for mood and may have depression, which causes excessive sleepiness and difficulties waking up.

Guberman et al. suggest a new way of thinking about concussions. If more studies confirm these concussion subgroups, scientists might use them to explore which types of therapies might be beneficial for patients with specific subgroups. Developing subgroup-targeted treatments may help scientists overcome the challenges of trying to develop therapies that work across a range of injuries. Similar disease subgrouping strategies may also help researchers study other brain diseases that may vary from patient to patient.

an additional, hitherto ignored source of heterogeneity, whereby a variety of white matter structure alterations ('multi-tract') may be related to a variety of symptoms ('multi-symptoms') in different ways. This disease-specific type of heterogeneity will henceforth be referred to as multi-tract multi-symptom heterogeneity for brevity. Parsing concussion heterogeneity requires accounting for these dynamic, multi-tract multi-symptom relationships.

The objective of the present study was to leverage advanced diffusion MRI (dMRI) methods as well as a double-multivariate approach to parse multi-tract multi-symptom heterogeneity in a large sample of previously concussed children. Multi-tract multi-symptom relationships captured more information than traditional univariate approaches. Expression of multi-tract connectivity features was not driven by sociodemographic strata, injury characteristics, or clinical subgroups. Analyses comparing clinical subgroups defined by the presence of attention-deficit/hyperactivity disorder showed that multi-tract multi-symptom analyses identified disease-specific connectivity patterns that were missed by single-tract single-symptom approaches.

## Materials and methods
### Participants

Data in this study were obtained from the world's largest child development study of its kind – the ongoing longitudinal Adolescent Brain Cognitive Development Study (ABCD Study; https://abcd-study.org/), data release 2.0 (https://data-archive.nimh.nih.gov/abcd). The ABCD Study acquired data from 11,874 children aged 9–10 years (mean age = 9.49 years) from across the United States (48% girls; 57% Caucasian, 15% African American, 20% Hispanic, 8% other) (*Volkow et al., 2018*).

Additional information about the ABCD Study can be found in *Garavan et al., 2018*. This dataset is administered by the National Institutes of Mental Health Data Archive and is freely available to all qualified researchers upon submission of an access request. All relevant instructions to obtain the data can be found in https://nda.nih.gov/abcd/request-access. The Institutional Review Board of the McGill University Faculty of Medicine and Health Sciences reviewed the application and confirmed that no further ethics approvals were required.

## History of concussion

Parents completed a modified version of the Ohio State University TBI Identification Method (OSU-TBI-ID) (*Corrigan and Bogner, 2007*). We included participants who reported a head injury without loss of consciousness but with memory loss and/or a head injury with loss of consciousness for less than 30 min (n = 434). Due to missing or incomplete data, corrupted files, data conversion errors, and images rated by the ABCD Study team as being of poor quality, the final sample of participants with usable data was 345. After processing, images were visually inspected by two trained independent raters (G.I.G., S.S.). Images that were deemed of low quality after processing by both raters were removed (n = 39), leading to a final sample of 306 participants. We randomly divided the sample into a discovery dataset (70%, n = 214) and a replication dataset (30%, n = 92). *Figure 1* summarizes the subject selection procedure.

## Symptom-oriented measures

To probe various aspects of concussion symptomatology, we used items collected from assessments available in the ABCD dataset. These items, as well as the concussion symptom they are meant to probe are outlined in *Table 1*.

## MRI acquisition

MRI scans were acquired across 21 sites, with data coming from 28 different scanners. Details about the acquisition protocols and image specifications are outlined in *Casey et al., 2018*. Multi-shell dMRI scans had 96 diffusion-weighted directions, with 6 directions of b = 500 s/mm$^2$, 15 directions of b = 1000 s/mm$^2$, 15 directions of b = 2000 s/mm$^2$, and 60 directions of b = 3000 s/mm$^2$. The b = 2000 shell was excluded from the data processing. In addition, scans had 6 or 7 b = 0 s/mm$^2$ images, depending on scanner type. Lastly, a reverse b0 image was included for each participant.

## Processing

We used Tractoflow (*Theaud et al., 2020*) to process dMRI and T1-weighted scans. Tractoflow is a novel diffusion MRI processing pipeline, incorporating state-of-the-art functions from FSL, Dipy, and MRtrix into NextFlow. The processing steps are summarized in *Theaud et al., 2020*. Important deviations from the default parameters utilized by Tractoflow are as follows: 1. We used gray-white matter interface seeding, as this method accounts for the length bias introduced by white-matter seeding; (*Girard et al., 2014*) 2. We used 24 seeds-per-voxel with the objective of obtaining approximately 2 million streamlines across the entire brain. We used the b = 0, 500, and 1000 shells to perform tensor fitting, and the b = 0 and 3000 shells to perform Constrained Spherical Deconvolution (CSD) (*Descoteaux et al., 2009*; *Tournier et al., 2007*). We obtained group-average fiber-response functions from voxels with high ( > 0.70) fractional anisotropy (FA). Lastly, we created tractograms using a probabilistic particle-filtering tractography algorithm (*Girard et al., 2014*).

## Connectivity matrices

The post-processing workflow is illustrated in *Figure 2*. To construct connectivity matrices, we used Freesurfer on McGill's CBrain platform (*Sherif et al., 2014*) to fit the Desikan-Killiani Tourvile (DKT) (*Klein and Tourville, 2012*) and *aseg* atlases onto the processed T1-images that had been transformed to DWI space during processing (*Figure 2A*). We applied these parcellations and extracted diffusion measures using *connectoflow* (version 1.0.0) (https://github.com/scilus/connectoflow; *Rheault and Houde, 2021*). This novel pipeline uses *scilpy* (version 1.0.0) scripts (https://github.com/scilus/scilpy; *Sherbrooke Connectivity Imaging Lab, 2022*) (wrappers of *Dipy*) implemented in Nextflow to split parcellations into individual labels, apply them to tractograms to create individual bundles, and then extract diffusion measures across them. To implement *connectoflow*, we first removed redundant

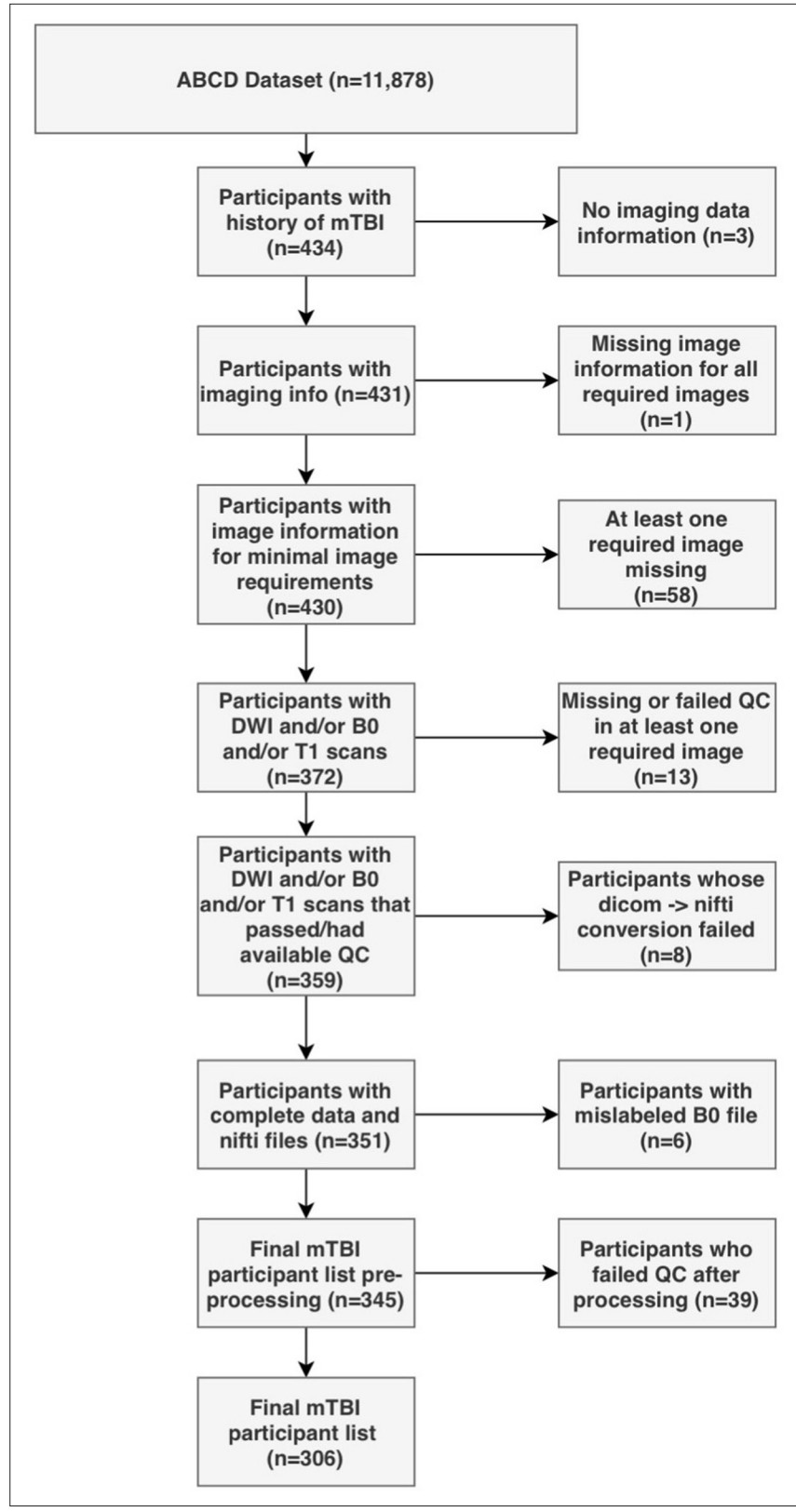

**Figure 1.** Flowchart describing the participant selection procedure.

**Table 1.** Table outlining all behavioral measures used in analyses, along with the corresponding symptom they reflect.

| Questionnaire - Description | Symptom Measured | Respondent |
|---|---|---|
| CBCL – Headaches | Headaches | Parent |
| CBCL – Nausea, feels sick | Nausea | Parent |
| CBCL – Vomiting, throwing up | Vomiting | Parent |
| CBCL – Feels dizzy or lightheaded | Dizziness | Parent |
| CBCL – Overtired without good reason | Fatigue | Parent |
| SDS – The child experiences daytime sleepiness | Drowsiness | Parent |
| SDS – The child has difficulty getting to sleep at night | Trouble falling asleep | Parent |
| CBCL – Sleep more than most kids during day and/or night | Sleep more than usual | Parent |
| CBCL – Sleeps less than most kids | Sleep less than usual | Parent |
| CBCL – Depression (DSM) T score | Sadness | Parent |
| CBCL – Anxiety Disorder (DSM) T score | Nervousness | Parent |
| CBCL – Attention Problems T score | Difficulty concentrating | Parent |
| CBCL Aggression T score | Irritability | Child |
| NIH Toolbox Picture Sequence Memory Test – Fully-Corrected T-score | Sequence Memory (difficulty remembering) | Child |
| NIH Toolbox List Sorting Working Memory Test – Fully-Corrected T-score | Working memory (difficulty remembering) | Child |
| RAVLT Short Delay Trial VI – Total Correct | Short recall (difficulty remembering) | Child |
| RAVLT Long Delay Trial VII – Total Correct | Long recall (difficulty remembering) | Child |
| NIH Toolbox Dimensional Change Card Sort Test – Fully-Corrected T-score | Executive function (feeling "foggy") | Child |
| NIH Toolbox Pattern Comparison Processing Speed Test – Fully-Corrected T-score | Processing speed (feeling "slow") | Child |

CBCL: Child Behavior Checklist. SDS: Sleep Disturbance Scale. NIH: National Institutes of Health. DSM: Diagnostics and Statistics Manual. RAVLT: Ray Auditory Verbal Learning Test.

and irrelevant labels from the fitted atlas (a list of retained labels is supplied in *Supplementary file 1*), yielding a final atlas with 76 labels. We then thresholded matrices such that a connection was only retained if it was found to be successfully reconstructed (defined as the presence of at least one streamline) across 90% of participants (*Guberman et al., 2020a*). We then performed a procedure to minimize the impact of spurious streamlines on our results (see "Accounting for spurious streamlines" paragraph below). We then randomly divided the sample into a discovery dataset (70%, n = 214) and a replication dataset (30%, n = 92). Every step hereafter was performed separately for each dataset. On each dataset, we weighted thresholded connectomes by FA, mean, radial, and axial diffusivities (MD, RD, AD respectively), apparent fiber density along fixels (AFDf), and number of fiber orientations (NuFO) (*Figure 2B*). The first four measures are derived from the tensor model, whereas the latter two are based on fiber orientation distribution functions (fODFs) obtained from CSD (*Raffelt et al., 2012*; *Dell'Acqua et al., 2013*). Simulation studies have shown that AFD is more specifically related to axonal density, and by computing it along 'fixels' (fiber elements), axonal density specific to particular fiber populations can be studied independently of crossing fibers (*Raffelt et al., 2012*).

## Additional data transformations

We imputed missing connectivity (prior to the PCA), symptom, and nuisance data (sex, pubertal stage, handedness, scanner) by randomly selecting non-missing data from other participants in the same dataset. We reverse-coded cognitive scores, such that increasing scores in all symptom data reflected more problems. From connectivity and symptom data, we regressed out the following nuisance variables: sex, pubertal stage, scanner (only for connectivity data), and handedness. An illustration of the impact of regressing out scanner from connectivity data can be found in *Appendix 1—figure 1*.

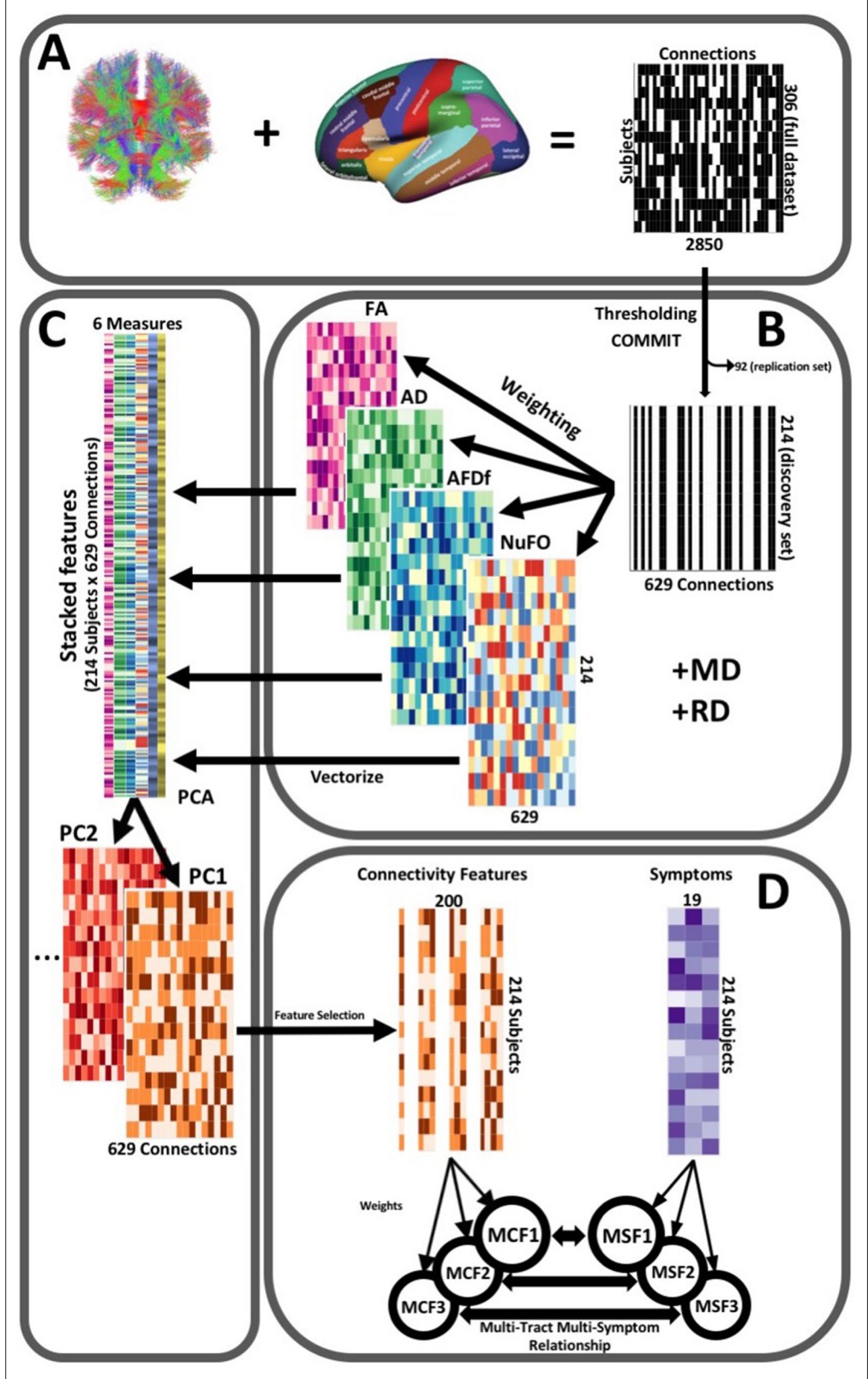

**Figure 2.** Illustration of the study's post-processing pipeline. (**A**). We applied the DKT parcellation onto each tractogram, thus building a binary connectivity matrix that displayed for all 306 subjects in the full dataset (rows), whether (black) or not (white) a streamline existed between each pair of labels (columns). (**B**). We thresholded connectomes using the full dataset, only keeping connections that existed across 90% of participants (a threshold

*Figure 2 continued on next page*

*Figure 2 continued*

of 100% is illustrated here for simplicity). On these connections, we also filtered streamlines by computing COMMIT weights. This technique assigns weights to streamlines depending on how well they explain the diffusion signal. We identified connections as spurious if all their streamlines had a COMMIT weight of 0. We only retained connections that were found to be non-spurious across 90% of participants in the full dataset. We then split the dataset into a discovery set (n = 214) and a replication set (n = 92). Using the discovery set, we then constructed connectomes of 6 scalar diffusion measures (Fractional Anisotropy (FA), Axial Diffusivity (AD), Mean Diffusivity (MD), Radial Diffusivity (RD), Apparent Fiber Density along fixels (AFDf), and Number of Fiber Orientations (NuFO)), by computing the average measure across each connection. (**C**). We stacked all columns from each connectivity matrix, creating vectors of every pair of subject and connection, and then joined together these vectors. We then performed principal component analysis (PCA) on these matrices. Principal component (PC) scores were calculated for each subject/connection combination, thus reconstructing connectomes weighted by PC scores. (**D**). From each these new connectomes, we selected 200 connections based on Pearson correlations with symptom-oriented measures. We then performed partial least squares correlation on each of these PC-weighted features and symptom measures, which allowed us to obtain pairs of multi-tract connectivity features ('MCF') and multi-symptom features ('MSF'). Each multivariate feature is composed of linear combinations (weighted sums, illustrated by the black arrows called 'weights') of variables from its corresponding feature set.

## Accounting for spurious streamlines

Most tractography techniques, including the one presently used, depend on propagating the local diffusion model across voxels. This approach has inherent limitations in voxels where the local model lacks the information necessary to inform on the appropriate path for streamline propagation, leading inevitably to the creation of spurious streamlines (*Maier-Hein et al., 2017*; *Girard et al., 2020*). To minimize the impact of spurious connections on our results, we implemented an approach called Convex optimization modeling for microstructure informed tractography (COMMIT) (*Daducci et al., 2015*). This technique assigns weights to streamlines based on how they explain the diffusion signal. After running COMMIT, we identified streamlines with weights of 0, which signified that these streamlines were not necessary to explain the dMRI signal because they were spurious or redundant. When all the streamlines of a connection had a weight of 0, the entire connection was considered spurious. We identified non-spurious connections and only kept those that were found to be non-spurious across 90% of participants in the full dataset.

## Principal components analysis

Although individual diffusion measures are related to different aspects of neuropathology, together they provide more information than when considered separately (*Guberman et al., 2020a*). A recent framework based on principal component analysis (PCA) has been proposed to combine diffusion measures into biologically interpretable indices of white matter structure (*Chamberland et al., 2019*). We therefore performed PCA on the concatenated set of standardized measures across subjects in the discovery set and connections that passed COMMIT filtering, generating connectivity matrices weighted by principal component (PC) scores (*Figure 2C*).

## Pattern-learning pipeline

### Feature selection

Given the constraints on the number of connectivity features that can be included in the partial least squares correlation (PLSc) analysis, we performed a univariate feature selection based on Pearson correlations. This solution is becoming increasingly adopted for high-dimensional variable sets (*Figure 2D*; *Boulesteix, 2004*; *Wang et al., 2020*). From the connections that passed COMMIT filtering, we selected the 200 connectivity features most correlated with any symptom score, to maximize the number of features included. Given our discovery dataset size, selecting 200 connectivity features corresponded to 93% of our sample, a level of granularity comparable to other recent neuroimaging studies employing a feature selection step prior to multivariate analyses (*Dinga et al., 2019*; *Drysdale et al., 2017*).

### PLSc

We performed PLSc analyses in R using the *tepPLS* function from the *texposition* package (*Beaton et al., 2014*). PLSc involves singular value decomposition on the covariance matrix between connectivity

and symptom features, creating pairs of multivariate connectivity and multivariate symptom features called multi-tract multi-symptom relationships. Each multi-tract multi-symptom relationship encapsulates a linear combination of connectivity features ('multi-tract features'), a linear combination of symptom scores ('multi-symptom features'), and an eigenvalue (reflective of the amount of explained covariance between connectivity and symptom features). Each multi-tract multi-symptom relationship is constructed so as to explain a successively smaller portion of the covariance between symptoms and connectivity features. We constructed the largest number of possible multi-tract multi-symptom relationships, given the dimensionality of the behavioral variable set (k = 19) (*Figure 2D*).

## Selection and interpretation of multi-tract multi-symptom pairs

To reduce the number of multi-tract multi-symptom pairs to retain for interpretation, we performed permutation testing by randomly shuffling row labels for the symptom features, without replacement, repeating the PLSc and computing eigenvalues at every permutation (2000 iterations). We calculated p-values as the proportion of permutations that yielded eigenvalues that exceeded the original amount.

To interpret symptom and connectivity weights of significant ($p < 0.05$) multi-tract multi-symptom pairs, we performed bootstrap analyses (2000 iterations), using the *BOOT4PLSC* command from the *texposition* package. At each iteration, labels for data were drawn with replacement, the entire PLSc was repeated and the weights for all pairs were obtained. Although the pairs are expected to differ between iterations, they are always ordered by the percentage of covariance in inputs they explain. This process yields a sampling distribution of weights for each connectivity and symptom feature (*McIntosh and Lobaugh, 2004*). The ratio of the original weights to the standard error of each measure's bootstrap distribution can be interpreted as a z-score, which yielded so-called 'bootstrap ratios'. We used a value of 1.96 to determine which variables significantly contributed to each particular significant pair.

## Comparison of multivariate against univariate approaches

To compare information captured by the PLSc and univariate approaches, we identified, among the 214 participants from the discovery set, those that had obtained a psychiatric diagnosis. Parents of all participants completed the Kiddie-Schedule for Affective and Psychiatric Disorders in School Age Children (KSADS), a gold-standard tool to assess the presence of pediatric psychiatric disorders (*Kaufman et al., 1997*). We divided the sample into clinical subgroups based on whether they had obtained a diagnosis of attention-deficit/hyperactivity disorder (ADHD). We selected this diagnosis because its behavioral manifestations can be easily related to some of the presently-studied concussion symptoms (e.g.: attention problems). It was also the second-most common diagnosis in our sample (33/214). Using a threshold of $p < 0.05$, we computed univariate comparisons of connectivity (PC scores) between individuals with and those without a diagnosis of ADHD, thus identifying putative 'ADHD-related' univariate connectivity features.

We were interested in comparing how many of these features were also found to significantly contribute to each multi-tract connectivity feature. To do so, we computed a measure of percent overlap as follows:

$$\%Overlap = \frac{C_{sig}}{(S_u + S_m - C_{sig})} \times 100,$$

where $C_{sig}$ refers to the number of connections flagged as significant in both approaches, $S_u$ to the number of connections flagged as significant in the univariate approach, and $S_m$ to the number of connections flagged as significant in the multivariate approach. This measure can account for the apparent high overlap that can arise when $S_u$ and $S_m$ are not equivalent in size.

## Relation to TBI-related and sociodemographic factors

We assessed whether expression of multi-tract connectivity features was related to injury-specific and sociodemographic factors ('external' variables). Injury-related variables included: the time between the last-documented injury and testing, the cause of injury, and the total number of documented mTBIs. Sociodemographic variables included: sex, total combined family income in the last 12 months, and race/ethnicity. We used the following categories for race/ethnicity: 'Asian' (Asian Indian, Chinese,

Filipino, Japanese, Korean, Vietnamese, Other Asian), AIAN ('American Indian'/Native American, Alaska Native), NHPI (Native Hawaiian, Guamanian, Samoan, Other Pacific Islander), Non-Hispanic White, Non-Hispanic Black, Hispanic, Other, and Multiple (*Heeringa, 2020*). To illustrate the influence of these sociodemographic factors we created scatter plots illustrating expression of connectivity latent factors color-coded by sociodemographic factors (*Appendix 1—figure 2*). In addition, we calculated correlations between multi-tract or multi-symptom feature expression and binary (or dummy-coded) variables representing these 'external' variables. These simple yet straightforward analyses allowed us to quantify the strength of the relationship between multivariate feature expression and external variables.

### Analyses on the replication dataset

To assess the robustness of our analyses, we first computed the percentage of connectivity/symptom covariance explained in the replication set by the first multi-tract multi-symptom pair of both PLSc analyses performed on the discovery set. We then selected, from the replication set, the same 200 connectivity features originally selected in the discovery set, and projected them, along with symptom features, onto the latent spaces obtained using the discovery set. To assess whether differences existed in multi-tract multi-symptom expression between participants from each set, we performed correlations comparing multi-tract multi-symptom feature expression against a binary variable indexing the dataset. Finally, we reran our feature selection procedure as well as the PLSc analyses on the dreplication set, and compared the number of connectivity features that coincided in both analyses. We also performed correlations comparing the loadings of every corresponding multi-tract and multi-symptom feature, as well as the expression of these features.

### Data availability

Data from the ABCD Study can be accessed by qualified researchers (see *Participants* section above for details). Scripts, supporting documents, and other information necessary to implement all aspects of data organization, preparation, and analysis can be found in https://github.com/GuidoGuberman/Multi-tract-multi-symptom-relationships-in-pediatric-concussion, (copy archived at swh:1:rev:4c-30fa113b2e0d24305a6e82fe8af54a3ed5af1a; *Guberman, 2022*).

## Results
### Sample

Out of 434 participants with a history of mild TBI (mTBI, used interchangeably with the term 'concussion' in this manuscript), 306 (127F/179 M) had usable data (*Figure 1*). *Table 2* outlines sociodemographic and injury-related factors, as well as handedness and sex. The majority had sustained an injury over 1 year prior to the study. Nuisance variables were well-balanced between participants in the discovery and the replication set.

### Combined measures of white matter tract microstructure

From all 2850 possible connections, 1,026 survived thresholding. Out of those 1026 connections, 629 survived COMMIT filtering. The PCA applied across dMRI measures from all 629 connections yielded two biologically-interpretable components that together explained 96% of the variance in measures (*Appendix 1—figure 3*). The first appeared to reflect an index of microstructural complexity, whereas the second more closely reflected axonal density. Because we retained two PCs, we performed two PLSc analyses.

### Multi-tract multi-symptom relationships

To parse multi-tract multi-symptom heterogeneity, we performed two PLSc analyses, one using the selected microstructural complexity features and another using the selected axonal density features, along with all 19 symptom features. Each PLSc analysis yielded 19 latent modes of covariance (termed here 'multi-tract multi-symptom relationships'), each consisting of a pair of multi-tract connectivity and multi-symptom features. Based on permutation testing, 16 multi-tract multi-symptom pairs were retained from the microstructural complexity PLSc, and 8 from the axonal density PLSc. *Appendix 1—figures 4 and 5* illustrate all the multi-symptom and multi-tract features (respectively) from the retained

**Table 2.** Table of sample characteristics.

| Demographic and injury data | Discovery set (n = 214) | Replication set (n = 92) |
|---|---|---|
| **Interview Age** | | |
| Mean (SD) | 9.57 (0.496) | 9.54 (0.501) |
| Median [Min, Max] | 10.0 [9.00, 10.00] | 10.0 [9.00, 10.0] |
| **Sex** | | |
| F | 88 (41.1%) | 39 (42.4%) |
| M | 126 (58.9%) | 53 (57.6%) |
| **Pubertal Stage** | | |
| Early | 41 (19.2%) | 18 (19.6%) |
| Mid | 58 (27.1%) | 19 (20.7%) |
| Prepubertal | 115 (53.7%) | 52 (56.5%) |
| Late | 0 (0%) | 3 (3.3%) |
| **Race/Ethnicity** | | |
| Asian | 2 (0.9%) | 2 (2.2%) |
| Hispanic | 27 (12.6%) | 18 (19.6%) |
| Multiple | 18 (8.4%) | 8 (8.7%) |
| Non-Hispanic Black | 14 (6.5%) | 11 (12.0%) |
| Non-Hispanic White | 151 (70.6%) | 52 (56.5%) |
| Other | 2 (0.9%) | 1 (1.1%) |
| **Combined Family Income** | | |
| < 5 K | 5 (2.3%) | 5 (5.4%) |
| $5,000 - $11,999 | 5 (2.3%) | 1 (1.1%) |
| $12,000-$15,999 | 3 (1.4%) | 2 (2.2%) |
| $16,000-$24,999 | 5 (2.3%) | 3 (3.3%) |
| $25,000-$34,999 | 12 (5.6%) | 4 (4.3%) |
| $35,000-$49,999 | 12 (5.6%) | 5 (5.4%) |
| $50,000-$74,999 | 34 (15.9%) | 16 (17.4%) |
| $75,000-$99,999 | 31 (14.5%) | 13 (14.1%) |
| $100,000-$199,000 | 76 (35.5%) | 27 (29.3%) |
| >$200,000 | 31 (14.5%) | 16 (17.4%) |
| **Handedness** | | |
| LH | 10 (4.7%) | 10 (10.9%) |
| RH | 175 (81.8%) | 69 (75%) |
| Mixed | 29 (13.6%) | 13 (14.1%) |
| **Injury Mechanism** | | |
| Fall/hit by object | 135 (63.1%) | 48 (52.2%) |
| Fight/shaken | 2 (0.9%) | 3 (3.3%) |
| Motor vehicle collision | 14 (6.5%) | 3 (3.3%) |
| Multiple | 10 (4.7%) | 5 (5.4%) |

*Table 2 continued on next page*

*Table 2 continued*

| Demographic and injury data | Discovery set (n = 214) | Replication set (n = 92) |
|---|---|---|
| Unknown | 53 (24.8%) | 33 (35.9%) |
| Time Since Injury (years) | | |
| Mean (SD) | 3.22 (2.79) | 3.23 (2.60) |
| Median [Min, Max] | 2.00 [0.00, 11.0] | 2.50 [0.00, 9.00] |
| Total TBIs | | |
| Unknown | 53 (24.8%) | 33 (35.9%) |
| 1 | 151 (70.6%) | 54 (58.7%) |
| 2 | 9 (4.2%) | 5 (5.4%) |
| 3 | 1 (0.5%) | 0 (0%) |

Note: Participants with "Unknown" Injury Mechanism and Total TBIs reported sustaining a TBI but no mechanism of injury was endorsed.

pairs from the microstructural complexity PLSc. Individual pairs selected for further discussion are shown in *Figures 3 and 4*. *Figure 3* also illustrates the expression of these multi-tract multi-symptom pairs (scatter plots). For each pair, the symptom profiles of two example participants, one with high feature expression, one with low, are shown. These example participants illustrate how these multi-tract multi-symptom features can represent a diversity of symptom profiles.

Across most extracted pairs, the representation of tracts and symptoms formed a continuum, with earlier pairs capturing broader symptom categories and more distributed networks of connections, and later pairs capturing more idiosyncratic symptom/connectivity relationships. The first multi-tract multi-symptom pair from the microstructural complexity PLSc broadly represented most symptoms (*Figure 3A* polar plot) and implicated a broad range of frontal commissural and occipito-temporal association tracts (*Figure 4*, violet brain graph). The third multi-tract multi-symptom pair obtained from the axonal density PLSc represented broadly cognitive problems (*Figure 3B* polar plot). The multi-symptom feature from the third pair obtained from the microstructural complexity PLSc also represented cognitive problems broadly and implicated a wide array of tracts with a mostly frontal focus. Features from subsequent pairs represented individual cognitive problems, such as feature 8 which represented processing speed, executive function (card sorting), and working memory, and implicated almost exclusively frontal tracts. The seventh pair obtained from the same PLSc represented attention problems almost exclusively, along with decreased sleep and processing speed (*Figure 3C* polar plot). This pair implicated mostly frontal tracts, including a connection between the left posterior cingulate and left thalamus, a trajectory that is consistent with the corticospinal tract. This pattern whereby pairs ranged from broadly representing symptom categories and distributed networks to more specific symptom combinations with more localized connections can be best appreciated in *Figure 5*. As can be observed, certain groups of connections tended to be represented only once alongside broad symptom categories (*Figure 5A*, orange rectangles). More consistent connectivity/symptom correspondences were only observed for few, more specific single-symptom/single-connection combinations (*Figure 5A* blue rectangle).

Although this pattern was observed in both PLSc analyses, important exceptions were observed as well. First, the second multi-tract multi-symptom pair obtained from both PLSc analyses strongly represented nausea and vomiting, almost exclusively, and implicated no commissural tracts. Second, sleep problems (especially 'trouble sleeping') were implicated across several pairs. Interestingly, despite being found ubiquitously across pairs, they were not consistently associated with the same connections across pairs. In contrast, nearly every time attention problems were implicated in a pair (3/4 pairs), they were found alongside two connections with trajectories that correspond to parts of the right superior longitudinal fasciculus (right pars opercularis – right post-central sulcus; right par opercularis – right sumpramarginal gyrus). However, this type of consistent symptom/connection correspondence was more often than not absent (*Figure 5*). Out of 200 connections selected for the microstructural complexity PLSc, 2 were found to be most frequently implicated across all retained

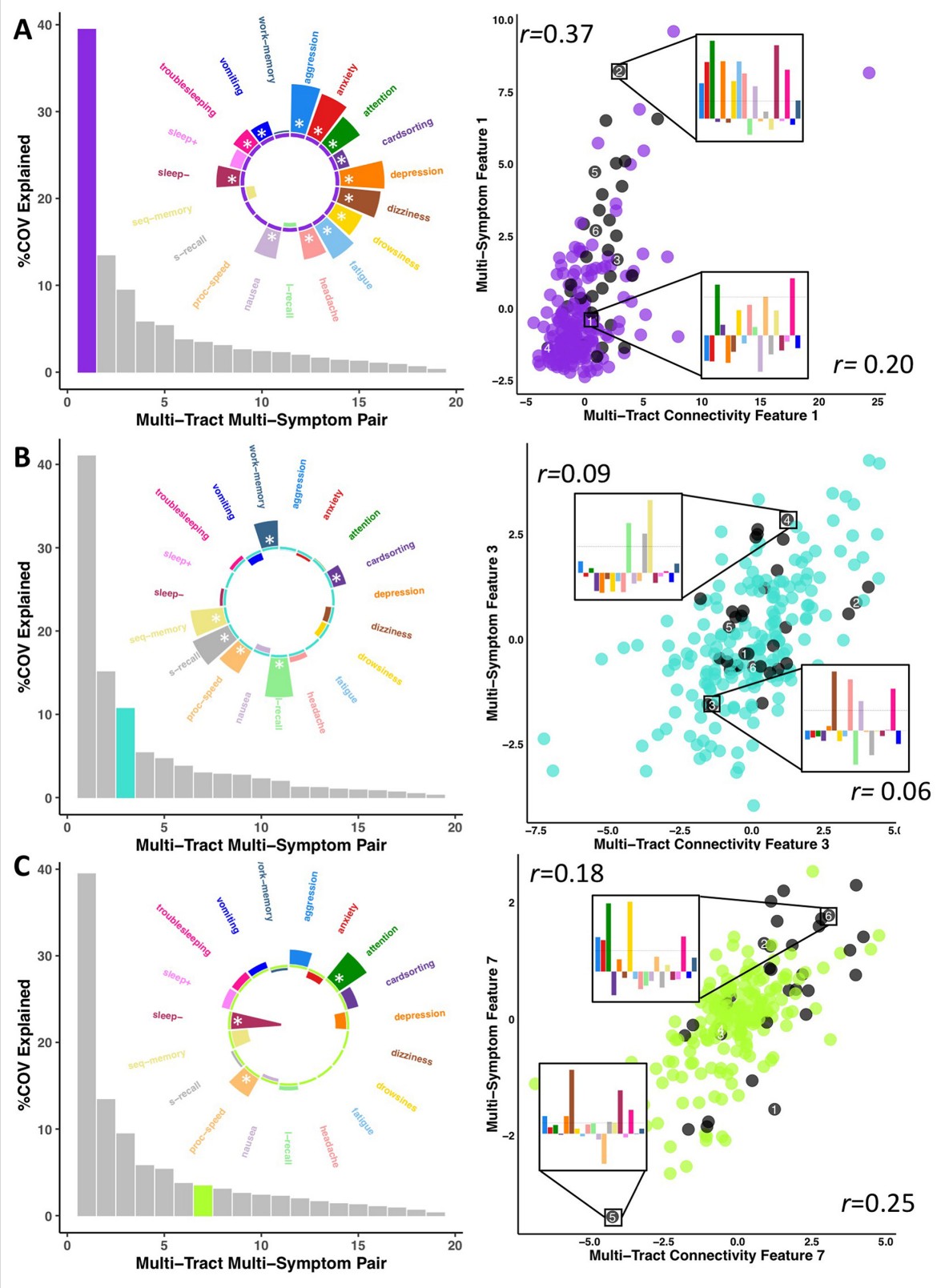

**Figure 3.** Illustration of multi-tract multi-symptom pairs 1 and 7 obtained from the microstructural complexity PLSc (A and C respectively), and pair 3 from the axonal density PLSc (**B**). Left: Polar plots displaying the weights of all 19 symptom measures for each multi-symptom feature. Bars pointing away from the center illustrate positive weights, bars pointing towards the center represent negative weights. White stars illustrate symptoms that significantly contributed to the pair. Bar graphs underneath the polar plots illustrate the % covariance explained by each pair, with the currently-shown

*Figure 3 continued on next page*

*Figure 3 continued*

pair highlighted. Right: Scatter plots showing the expression of multi-tract features (x-axis) and multi-symptom features (y-axis). In each scatter plot, the same 6 participants are labeled (1 through 6). Small bar graphs illustrate the scaled symptom *measures* (i.e.: not the expression of multi-symptom features) for two participants, one expressing low levels of a pair, the other expressing high levels. For each illustrated participant, positive bars illustrate symptoms that are higher than the sample average, negative bars represent symptoms that are lower. The black dashed line illustrates 1 standard deviation above the group mean. Participants with ADHD diagnoses are illustrated in black. Correlation coefficients inset in each scatter plot represent Pearson correlations between expression of multi-tract features (near x-axis), or multi-symptom features (near y-axis) and a binary variable indexing whether or not a participant had a diagnosis of ADHD.

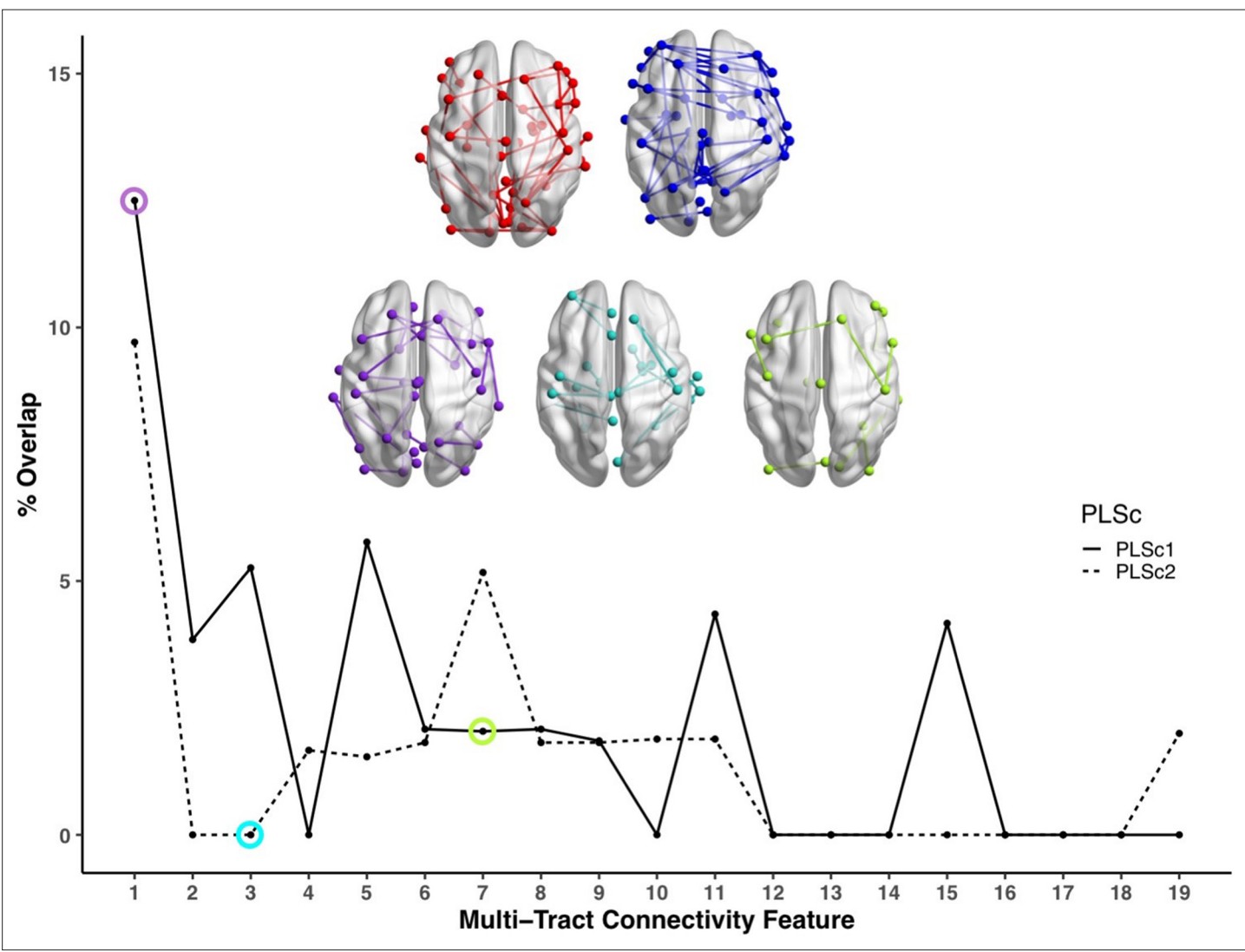

**Figure 4.** Line plot showing the percent overlap between univariate analyses and each multi-tract connectivity feature. Highest overlap occurred for the first multi-tract connectivity feature from both PLSc analyses. Brain renderings shown above graph illustrate which connections were found to be significant for univariate comparisons of microstructural complexity (red), univariate comparisons of axonal density (blue), multi-tract connectivity feature 1 from the microstructural complexity PLSc (violet), multi-tract connectivity feature 3 from the axonal density PLSc (turquoise), and multi-tract connectivity feature 7 from the microstructural complexity PLSc (green). The percent overlap score for each of the three illustrated multi-tract connectivity features are identified in the line plot with a circle of the corresponding color. Univariate brain graphs show connections significant at p < 0.01 for illustrative purposes. Multivariate brain graphs show connections significant at p < 0.05. Brain renderings were visualized with the BrainNet Viewer (*Xia et al., 2013*).

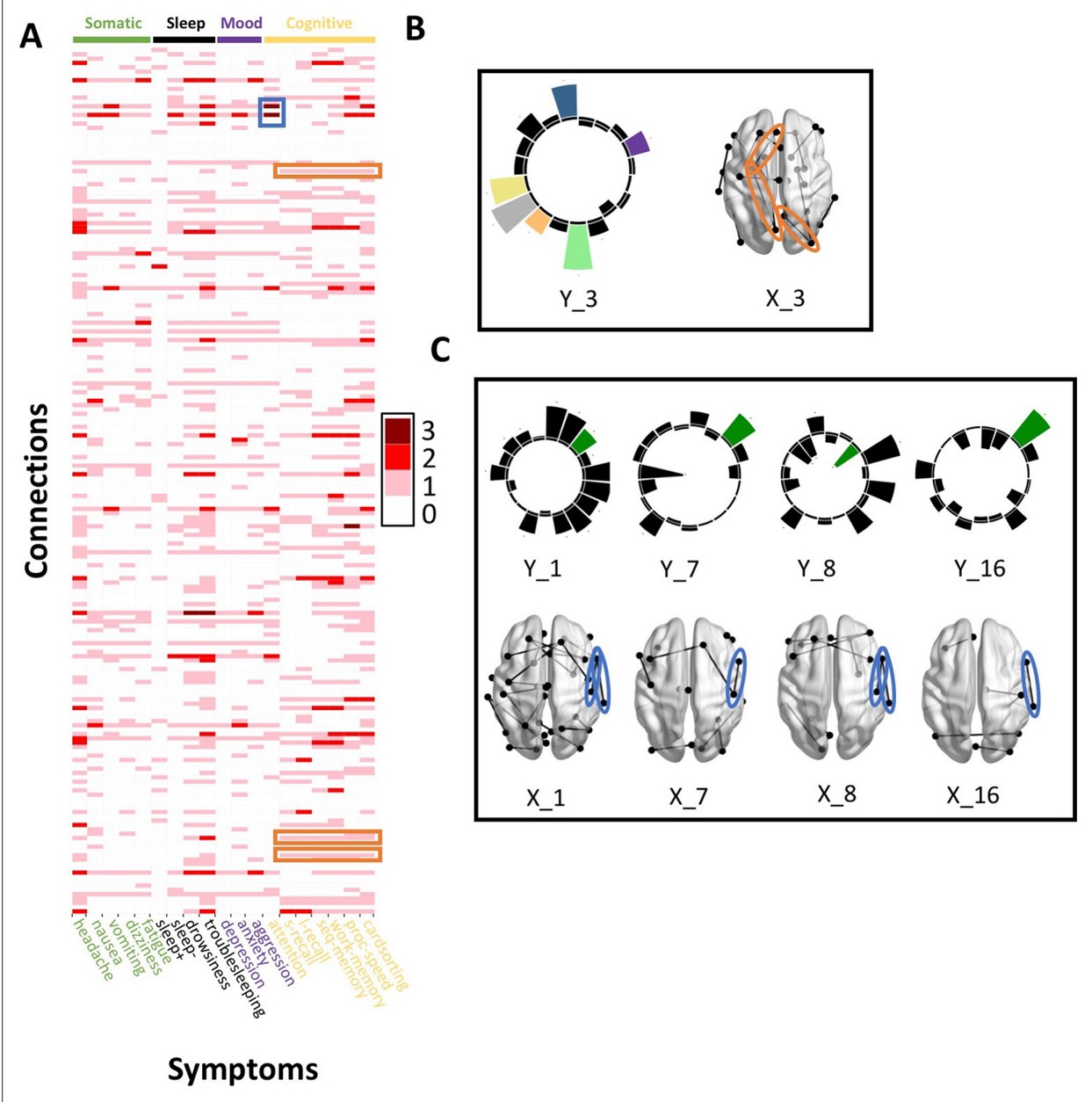

**Figure 5.** Patterns of connection/symptom correspondences across multi-tract multi-symptom pairs. (**A**). Adjacency matrix, illustrating the number of multi-tract multi-symptom pairs from the microstructural complexity PLSc where a given significant connection corresponded to a given significant symptom (based on bootstrap analyses). Darker colors illustrate more consistent correspondences. Symptom categories are illustrated in colors (green: somatic, black: sleep problems, purple: mood problems, yellow: cognitive problems). Orange rectangles highlight three connections (right lateral occipital – right precuneus; left putamen – left rostral anterior cingulate; left putamen – left lingual) that were only present in one multi-tract multi-symptom pair (pair 3), which also represented broadly all cognitive problems. This pair is illustrated in B, where cognitive problems are illustrated in color and all other symptoms are illustrated in black, and the highlighted connections are circled in orange. Although only three connections are highlighted, several such 'broad cognitive problems' connections can be observed. The blue rectangle highlights two connections (right pars opercularis – right post-central; right pars opercularis – right supramarginal) that were present in 4 multi-tract multi-symptom pairs, all of which also implicated attention problems. These pairs are illustrated in panel C, where attention problems are illustrated in color, all other symptoms are illustrated in black, and the highlighted connections are circled in blue.

pairs. These were the two connections mentioned above that were related to attention problems, the right pars opercularis – right post-central gyrus, and the right pars opercularis to the right supramarginal gyrus. They were implicated in 4 pairs each, two of which were the same (pairs 1 and 8). Callosal tracts were not among the most often implicated connections.

Overall, these results illustrate how different symptom profiles are associated with different combinations of tracts. Earlier pairs consisted of broad symptom categories and implicated wider networks of connectivity features, whereas more idiosyncratic pairs consisted of more localized connectivity features that were associated with more symptom-specific profiles. However, some symptoms such as sleep problems were implicated across the spectrum of different multi-tract multi-symptom pairs, illustrating how some symptoms do not demonstrate a one-to-one relationship with connectivity features across multi-tract multi-symptom pairs. No tracts were widely implicated across all pairs.

## Multi-tract multi-symptom features and clinical subgroup membership

We identified 33 individuals in the discovery set with diagnoses of ADHD obtained from the KSADS (*Dodd et al., 2014*). These individuals are shown in black in *Figure 3* (scatter plots). Despite forming a clinical subgroup based on a gold-standard measure of psychiatric pathology, these individuals were heterogeneous in their expression of multi-tract and multi-symptom features. Further, we computed correlations between the expression of these multivariate features and clinical subgroup membership, and found weak, albeit significant correlations (see *Figure 3* scatter plots and *Supplementary file 3*).

## Multivariate vs univariate approaches

We compared microstructural complexity and axonal density scores across all 200 connections between individuals with and without an ADHD diagnosis, and calculated the percent overlap between each multi-tract connectivity feature and the set of tracts found to be significant in univariate comparisons (ostensibly 'ADHD-related' tracts). The percent overlap scores are presented in *Figure 4*. Notably, the highest overlap occurred with multi-tract connectivity feature 1 (10–13%) from both PLSc analyses, which implicated a wide network of white matter tracts and were associated with general problems. In contrast, the overlap with multi-tract connectivity feature 7, which implicated mostly frontal connections and was associated with attention problems almost exclusively, was low (5%). Neither of the two univariate analyses implicated the two connections discussed above (right pars opercularis – right post-central sulcus; right par opercularis – right sumpramarginal gyrus) that were consistently associated with attention problems. These results suggest that the putative 'ADHD-related' connections identified in univariate comparisons of microstructural complexity and axonal density measures between individuals with ADHD and those without are mostly non-overlapping with the connections identified in an attention-problems specific multi-tract multi-symptom pair.

## Relationship with sociodemographic and injury-related factors

*Appendix 1—figure 2* illustrates the expression of multi-tract connectivity features color-coded by sociodemographic strata defined by sex, total combined household income, and race/ethnicity. Qualitatively, no clusters defined by these sociodemographic strata are apparent. Further, correlations between multi-tract/multi-symptom feature expression and binary (or dummy-coded) variables defining each strata are overall weak and non-significant (*Appendix 1—figure 2*). Out of 24 retained multi-tract multi-symptom pairs, time since the latest injury was only significantly correlated to the expression of one multi-tract connectivity features (*Supplementary file 2*) and no multi-symptom features. Only the expression of two multi-tract connectivity features (features 2 and 15 from the microstructural complexity PLSc) were significantly different between groups defined by injury cause (*Appendix 1—figures 6 and 7*, respectively). Only the expression of one multi-tract connectivity features, feature 15 from the microstructural complexity PLSc (*Appendix 1—figure 8*) significantly differed between groups defined by the total number of TBIs.

## Results on the replication dataset

We first analyzed the amount of connectivity/symptom covariance explained by each of the multi-tract/multi-symptom pairs. We found that approximately 26% of the connectivity/symptom covariance in the replication set was explained by the first multi-tract/multi-symptom pair from the microstructural complexity PLSc, and 40% was explained by the corresponding pair from the

axonal density PLSc. However, with permutation testing, these percentages were not found to be significantly higher than expected by chance. We then projected the replication dataset onto the latent spaces obtained using the discovery set, and ran correlations comparing multi-tract multi-symptom feature expression against a binary variable indexing the dataset. We found very low, non-significant correlations (all coefficients lower than 0.01) between multi-tract and multi-symptom feature expression and set membership. Finally, we reran our PCA and PLSc analyses on the replication set, and compared the features obtained against the original results. The loadings obtained from the PCA performed on the replication set are illustrated in *Appendix 1—figure 9*. The loadings for the first two PCs from both sets of analyses were highly correlated ($r = 0.9997$, p < 0.001; $r = 0.996$, p < 0.001). Out of the 200 connections originally selected in the discovery set, 73 were also selected in the replication set. *Appendix 1—figure 10* illustrates the loadings for all the multi-symptom features retained after permutation testing. Correlations between the loadings of these new multi-symptom features and the ones obtained from the original PLSc analyses were low to high (0.087–0.810). The correlations between the loadings of the 73 connections that were common to both PLSc analyses were very low to moderate (0.006–0.227). However, the correlations between expression of multi-symptom features obtained from the PLSc performed on the replication dataset and those obtained by projecting the replication set onto the original latent spaces reached higher values (0.01–0.979), which was also found for those between expression of multi-tract features (0.02–0.67). Altogether, these results suggest that despite having mostly different connectivity inputs due to the feature selection step, the analyses led to similar multi-tract multi-symptom features.

## Sensitivity analyses

We first tested the impact that changing the number of retained connections would have on PLSc results. The percentage of covariance explained by the first multi-tract multi-symptom pair from the microstructural complexity PLSc is illustrated in *Appendix 1—figure 11*. Initially, the percentage of covariance explained increased with an increasing number of connections, and stabilized around 200 connections. These results suggest that a selection of 200 connections is close to the optimal amount that could have been selected.

We then assessed the impact of modifying our resampling approach, attempting instead to shuffle the original connectivity features before the feature selection step. Using this stricter approach yielded no significant multivariate pairs on permutation testing.

We then assessed the impact of using different thresholds to retain connections (t = 85%, 95%, 100%). All 629 connections that had survived COMMIT in the original analyses (t = 90%) were among the 1,142 connections that survived the t = 85% threshold, and the 877 connections that survived the t = 95% threshold. Hence, after COMMIT, the same 629 connections were selected in all three thresholds, which led to identical data going into all subsequent analyses. Differences were only seen for t = 100%, where 258 connections survived thresholding. After COMMIT, 252 connections survived, suggesting that only 6 connections that were considered 'spurious' were found across 100% of participants. These results suggest that highly consistent connections also tended to be the ones found by COMMIT to be 'non-spurious'. Compared to the original threshold, the connectivity features obtained from the t = 100% threshold after PCA (*Appendix 1—figure 3*), and the weights of the multi-symptom features (*Appendix 1—figure 12*) were highly similar. However, using permutation testing, only two multi-tract multi-symptom pairs (12 and 15) were found to be significant (although in the figure, the same multi-symptom features that had been retained using the 90% threshold are shown, to facilitate comparison with *Appendix 1—figure 4*). Correlations between expression of multi-tract connectivity features at t = 90% and t = 100% are illustrated in the appendix (*Appendix 1—figure 13*). Correlations between the expression of the corresponding multi-tract connectivity features (e.g.: multi-tract feature 1 from t = 90%, multi-tract feature 1 from t = 100%) had high correlations overall, with exceptions arising around the middle features (7-14). As can be observed in both *Appendix 1—figures 4 and 12*, these middle features appeared to be switched in order. For instance, multi-symptom feature 8 from the microstructural complexity PLSc using the t = 90% threshold was similar to the multi-symtom feature 7 from the PLSc using the t = 100% threshold, and the expression of these two features were highly correlated as well. Overall, these results indicate that features obtained from the PLSc analyses were similar across thresholds (*Appendix 1—figure 13*).

## Discussion

In the present study, we leveraged novel dMRI methods and a double-multivariate approach to parse heterogeneity in the relationship between white matter structure and symptoms in a large sample of previously-concussed children. By applying PLSc on biologically interpretable measures of dMRI obtained from PCA, we found cross-demographic multi-tract multi-symptom features that captured information about structure/symptom relationships that traditional approaches missed. More representative multi-tract multi-symptom pairs represented broader symptom categories and implicated wider networks of connections, whereas more idiosyncratic pairs represented more specific symptom combinations and implicated more localized connections. Whereas certain symptom/tract correspondences were consistent across the pairs that implicated them, more often than not, tracts were not consistently associated with the same symptoms across pairs. This finding was especially apparent for sleep problems, which were implicated across most pairs. These results suggest that rather than a clean and consistent set of one-to-one symptom/tract relationships, concussions may instead be composed of subsets of symptom combinations that are associated with combinations of structural alterations of different white matter tracts.

Defining concussions as a clinical syndrome characterized by a set of symptoms stemming from a set of alterations of brain structure and function, multi-tract multi-symptom pairs can be thought of as subtypes of concussion (i.e.: of structure/symptom relationships), not of concussion patients. Patients with concussions can express these different concussion subtypes to varying degrees. We theorize that the combination of these different subtypes and how much they are expressed is what determines the clinical syndrome a person will display. The pairs explaining the most covariance can be interpreted as the subtypes that are most commonly expressed across individuals. Whether these subtypes are dataset-specific remains to be explored. Multi-tract relationships may be driven by the metabolic demands imposed by the network structure of the brain, which is known to predict the course of several brain diseases (*Crossley et al., 2014*), by biomechanical constraints imposed by the skull and other structures exposing certain areas to more shearing strain (*Hernandez et al., 2019*), or by both factors simultaneously (*Anderson et al., 2020*). These possibilities need to be tested further. Multi-symptom relationships may be driven by feedback mechanisms, with symptoms potentiating each other. One example is sleep: sleep disturbances were implicated in nearly all multi-tract multi-symptom pairs, but showed no consistent correspondence with any particular connection. This result suggests that sleep disturbances across concussed patients may not be associated with the same neural substrate, but may instead arise, in some concussions, as a consequence of other symptoms. Previous theoretical work has proposed that sleep may play a central role in concussions, given that it can arise as a consequence of certain symptoms but can also potentiate other symptoms (*Kenzie et al., 2018*). The current findings are consistent with this previous work.

Concussion heterogeneity has been identified as a major obstacle (*Kenzie et al., 2017*; *Hawryluk and Bullock, 2016*) in response to decades of failed attempts to translate basic science findings into successful clinical trials and novel therapies. Heterogeneity in symptoms, impact of injury on brain structure and function, and pre-injury factors pose a particular problem for most concussion neuroimaging studies which have traditionally employed univariate comparisons between concussed and healthy or orthopedic injury control groups, or between patients with and without persistent symptoms (*Dodd et al., 2014*; *Hulkower et al., 2013*). These sources of inter-subject variability are believed to be problematic because they decrease the statistical power needed for group comparisons and multivariable models to detect the often-subtle effects of concussions (*Maas et al., 2013*). To overcome this challenge, landmark initiatives such as the IMPACT (*Maas et al., 2013*), InTBIR (*Tosetti et al., 2013*), CENTER TBI (*Maas et al., 2015*), and TRACK TBI (*Bodien et al., 2018*) aim to standardize and pool multi-center data collected across sociodemographic strata, to identify and statistically correct for pre-injury factors known to impact brain structure, and develop diagnostic and prognostic tools leveraging multimodal data and increasingly sophisticated machine-learning approaches.

In this study, we posited that disease-specific concussion heterogeneity is also problematic because by pooling across patients, idiosyncratic patterns of connectivity that may be more symptom-specific are sacrificed in favour of shared ones. By assuming that symptoms map cleanly and consistently onto shared connectivity abnormalities in a one-to-one fashion, erroneous inferences could be made about relationships between group-level patterns of connectivity differences and specific symptoms. Our results are consistent with this idea: univariate comparisons between a clinical subgroup defined

by a diagnosis of ADHD and the rest of the sample identified connectivity features that mostly overlapped with the first multi-tract multi-symptom pair obtained from both PLSc analyses. These pairs, which accounted for the most covariance, reflected general problems and not specifically ADHD. Both these pairs and the univariate 'ADHD-related' connections implicated a distributed network of tracts. Instead, a multi-tract multi-symptom pair that more uniquely represented attention problems implicated mostly frontal connections, including a connection that was part of the corticospinal tract. These findings are consistent with prior literature showing differences in white matter structure, especially in the corticospinal tract, among children with ADHD (*Silk et al., 2016*; *Wu et al., 2020*; *Puzzo et al., 2018*). Nonetheless, children with ADHD were heterogeneous in the expression of this more attention-specific multi-tract multi-symptom pair, suggesting that this clinical subgroup of children with TBIs may have important differences that can be further investigated. Across pairs, nearly every time attention problems were implicated, they were found alongside two connections with trajectories that correspond to parts of the right superior longitudinal fasciculus. This result is consistent with previous work that has shown differences in the structure of the right superior longitudinal fasciculus in children and adults with ADHD (*Cortese et al., 2013*; *Hamilton et al., 2008*; *Konrad et al., 2010*; *Makris et al., 2008*; *Wolfers et al., 2015*). Overall, these results suggest that univariate comparisons in concussed children, even when performed in such a way as to identify a diagnosis-specific set of connectivity features, identified only the most consistent group-level connectivity differences at the expense of more symptom-specific idiosyncratic ones.

The present findings must be contrasted to the nascent literature addressing concussion heterogeneity. A few recent studies have parsed inter-subject heterogeneity in concussion symptoms, using clustering analyses to group concussion patients into discrete subtypes (*Langdon et al., 2020*; *Si et al., 2018*; *Yeates et al., 2019*). The symptoms displayed by these reported subgroups differed from those implicated in the multi-symptom features found in the present study. Differences between symptom profiles arose because our multi-symptom features are associated with white matter structure and not driven by variability in symptoms alone. Other prior studies have attempted to address inter-subject heterogeneity in white matter structure in concussions (*Stojanovski et al., 2019b*; *Taylor et al., 2020*; *Ware et al., 2017*). Using two different approaches, these studies generated point summaries that accounted for the high-dimensional variability of white matter structure to better distinguish patients from controls. These prior studies, using a variety of approaches, have all focused on parsing down inter-subject heterogeneity in symptoms or white matter structure. Our approach focuses instead on disease-specific heterogeneity in structure/symptom relationships.

The present results should be considered in light of methodological limitations. Data on mTBI occurrence was collected retrospectively. Participants did not have baseline data, and additionally had highly variable times since injury. Most individuals with concussions recover from their injury (*Leddy et al., 2012*) which should have led to a concussed group where most participants were similar to healthy controls. Interestingly, our PCA yielded combinations of diffusion measures that differed from those of two prior studies that have used this approach (*Chamberland et al., 2019*; *Geeraert et al., 2020*). These prior studies used samples of typically developing children without neurological insults. To the extent that the PCs reported in these prior studies reflect healthy neurotypical brains, our PCs suggest that our concussed sample was not as similar in white matter structure to healthy controls as expected. However, variable time since injury made the interpretation of patterns of microstructure difficult. Further, due to the cross-sectional nature of this data, the difference between symptoms and pre-existing characteristics are difficult to discern, especially since some behavioral measures often believed to be symptoms of concussion, such as attention problems, can also be risk factors for brain injury (*Guberman et al., 2020b*). Heterogeneity has several forms, including in symptoms, duration, severity, neuropathology (*Bigler and Maxwell, 2012*), lesion location (*Ware et al., 2017*), sociodemographics (*Maas et al., 2013*), genetics (*Stojanovski et al., 2019a*), behavior (*Guberman et al., 2020b*), pre-injury comorbidities (*Yue et al., 2019*), and environmental differences, including access to and quality of care (*Yue et al., 2020*). These factors have been theorized to interact in complex ways (*Kenzie et al., 2017*). This study only addressed a minority of these complex relationships, further studies integrating more variable sets are needed to address these other drivers of heterogeneity. The connectivity features that were studied came from connectomes, not from well-known bundles. This choice was made to obtain a larger coverage of white matter structure, but as a result, sacrificed interpretability. The reason is that the connectivity features studied here may or may not represent true

anatomical units, and hence cannot be interpreted easily on their own. Instead, we relied on patterns across tracts, such as counting the number of connections that corresponded to the same symptom across different multi-tract multi-symptom pairs. Future studies should contrast this approach against procedures used to extract large well-known bundles. Lastly, methodological choices are a central part of this study. As illustrated by the sensitivity analyses and analyses performed on the replication set, some parts of the analytical pipeline were robust to methodological variations, whereas others were not. For instance, the PLSc procedure was robust to variations in input data. This was evidenced by the non-significant correlations between multi-tract multi-symptom feature expressions and a variable indexing the dataset, as well as the similarities observed after running the PLSc analyses on the replication set. However, the univariate feature selection approach was not as robust, as shown by the low overlap of connections when using different input datasets, as well as the loss of significant permutation test results when using a stricter resampling method that reshuffled prior to feature selection. Importantly, this study was not attempting to identify a single best analytical approach for studying multi-tract multi-symptom relationships. Rather, this project aimed to present a fundamentally different way of understanding concussions. Among the myriad of options available, we found a compromise between novel, cutting-edge techniques, and established, well-known ones. However, for parts of the analytical pipeline, refinements can be made. Future iterations of this work will need to exert better control of time since injury, perform longitudinal follow-up, develop predictive models for clinical outcomes, and assess the impact of choices made among the panoply of alternatives along key steps of the analytical pipeline.

Conversely, this study leveraged some of the most recent and important advances in dMRI to address the major limitations of conventional approaches. We used high-quality multi-shell dMRI data (*Jones et al., 2013*), as well as modeling approaches, tractography techniques, and microstructural measures robust to crossing fibers, partial volume effects, and connectivity biases (*Girard et al., 2014*; *Descoteaux et al., 2009*; *Tournier et al., 2007*; *Raffelt et al., 2012*; *Dell'Acqua et al., 2013*; *Chamberland et al., 2019*). We used PCA to combine dMRI measures into meaningful indices of white matter structure. Lastly, we used gold-standard measures of psychiatric illness to divide the sample into meaningful clinical subgroups.

In conclusion, leveraging advanced dMRI and a pattern-learning algorithm to parse white matter structure/symptom heterogeneity, we have found clinically-meaningful, cross-demographic multi-tract multi-symptom relationships. As the field moves towards large-scale studies which aim to statistically control for sociodemographic sources of heterogeneity to detect a putative consistent white matter signature of concussion across patients, the fundamental insight of this study should be taken into consideration: when pooling across patients, disease-specific multi-tract multi-symptom heterogeneity is lost, leading to the loss of informative, clinically-meaningful, symptom-specific patterns of connectivity abnormalities. Future studies aiming to better understand the relationship between white matter abnormalities and concussion symptomatology should look beyond the group-comparison design and consider multi-tract multi-symptom heterogeneity.

## Acknowledgements

The authors thank François Rheault, Derek Beaton, Manon Edde, Guillaume Theaud, and Arnaud Boré for comments and other contributions that were helpful in this project. Data used in the preparation of this article were obtained from the Adolescent Brain Cognitive Development[SM] (ABCD) Study (https://abcdstudy.org), held in the NIMH Data Archive (NDA). This is a multisite, longitudinal study designed to recruit more than 10,000 children age 9–10 and follow them over 10 years into early adulthood. The ABCD Study is supported by the National Institutes of Health and additional federal partners under award numbers U01DA041048, U01DA050989, U01DA051016, U01DA041022, U01DA051018, U01DA051037, U01DA050987, U01DA041174, U01DA041106, U01DA041117, U01DA041028, U01DA041134, U01DA050988, U01DA051039, U01DA041156, U01DA041025, U01DA041120, U01DA051038, U01DA041148, U01DA041093, U01DA041089, U24DA041123, U24DA041147. A full list of supporters is available at https://abcdstudy.org/federal-partners.html. A listing of participating sites and a complete listing of the study investigators can be found at https://abcdstudy.org/consortium_members/. ABCD consortium investigators designed and implemented the study and/or provided data but did not necessarily participate in the analysis or writing of this report. This manuscript reflects the views of the authors and may not reflect the opinions or views of the NIH or ABCD

consortium investigators. The ABCD data repository grows and changes over time. The ABCD data used in this report came from https://dx.doi.org/10.15154/1503209.

---

# Additional information

## Competing interests

Maxime Descoteaux: works as Chief Scientific Officer for IMEKA. He holds the following patents: DETERMINATION OF WHITE-MATTER NEURODEGENERATIVE DISEASE BIOMARKERS (Patent Application No.: 63/222,914), PROCESSING OF TRACTOGRAPHY RESULTS USING AN AUTOEN-CODER (Patent Application No.: 17/337,413). The other authors declare that no competing interests exist.

## Funding

| Funder | Grant reference number | Author |
|---|---|---|
| Canadian Institutes of Health Research | Vanier Canada Graduate Scholarship | Guido I Guberman |
| Government of Ontario | Ontario Graduate Scholarship | Sonja Stojanovski |
| Hospital for Sick Children | Restracomp Research Fellowship | Sonja Stojanovski |
| Université de Sherbrooke | Institutional Research Chair | Maxime Descoteaux |
| Natural Sciences and Engineering Research Council of Canada | CREATE Grant | Maxime Descoteaux |

The funders had no role in study design, data collection and interpretation, or the decision to submit the work for publication.

## Author contributions

Guido I Guberman, Conceptualization, Data curation, Formal analysis, Investigation, Methodology, Project administration, Resources, Software, Validation, Visualization, Writing – original draft, Writing – review and editing; Sonja Stojanovski, Conceptualization, Data curation, Methodology, Visualization, Writing – review and editing; Eman Nishat, Data curation, Resources, Writing – review and editing; Alain Ptito, Resources, Supervision, Writing – review and editing; Danilo Bzdok, Anne L Wheeler, Conceptualization, Supervision, Writing – review and editing; Maxime Descoteaux, Conceptualization, Resources, Software, Supervision, Writing – review and editing

## Author ORCIDs

Guido I Guberman ⬤ http://orcid.org/0000-0002-4422-2225

## Ethics

Human subjects: The data used in this study were obtained from the Adolescent Brain Cognitive Development Study. All aspects related to ethical standards were managed by the ABCD Study team. The Institutional Review Board of the McGill University Faculty of Medicine and Health Sciences reviewed the application and confirmed that no further ethics approvals were required.

## Decision letter and Author response

Decision letter https://doi.org/10.7554/eLife.70450.sa1
Author response https://doi.org/10.7554/eLife.70450.sa2

---

# Additional files

## Supplementary files
- Transparent reporting form
- Supplementary file 1. Table listing labels retained in the DKT + aseg parcellation.

• Supplementary file 2. Table listing p-values of correlations between the expression of all retained multi-tract connectivity features and the time since the latest injury. Note: PLSc1: Microstructural complexity PLSc; PLSc2: Axonal Density PLSc.

• Supplementary file 3. Table listing p values of correlations between the expression of all retained multi-tract connectivity features and a variable indexing ADHD. Note: PLSc1: Microstructural complexity PLSc; PLSc2: Axonal Density PLSc.

## Data availability

All data used in this project were obtained from the Adolescent Brain Cognitive Development Study. This dataset is administered by the National Institutes of Mental Health Data Archive and is freely available to all qualified researchers upon submission of an access request. All relevant instructions to obtain the data can be found in https://nda.nih.gov/abcd/request-access. The Institutional Review Board of the McGill University Faculty of Medicine and Health Sciences reviewed the application and confirmed that no further ethics approvals were required.

The following previously published dataset was used:

| Author(s) | Year | Dataset title | Dataset URL | Database and Identifier |
|---|---|---|---|---|
| Casey BJ, Cannonier T, Conley MI, Cohen AO, Barch DM, Heitzeg MM, Soules ME, Teslovich T, Dellarco DV, Garavan H, Orr CA, Wager TD, Banich MT, Speer NK, Sutherland MT, Riedel MC, Dick AS, Bjork JM, Thomas KM, Chaarani B, Mejia MH, Hagler DJ, Cornejo DM, Sicat CS, Harms MP, Dosenbach NUF, Rosenberg M, Earl E, Bartsch H, Watts R, Polimeni JR, Kuperman JM, Fair DA, Dale AM | 2018 | Adolescent Brain Cognitive Development Study | https://nda.nih.gov/abcd | NIMH Data Archive Collection, #2573 |

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

## Appendix 1

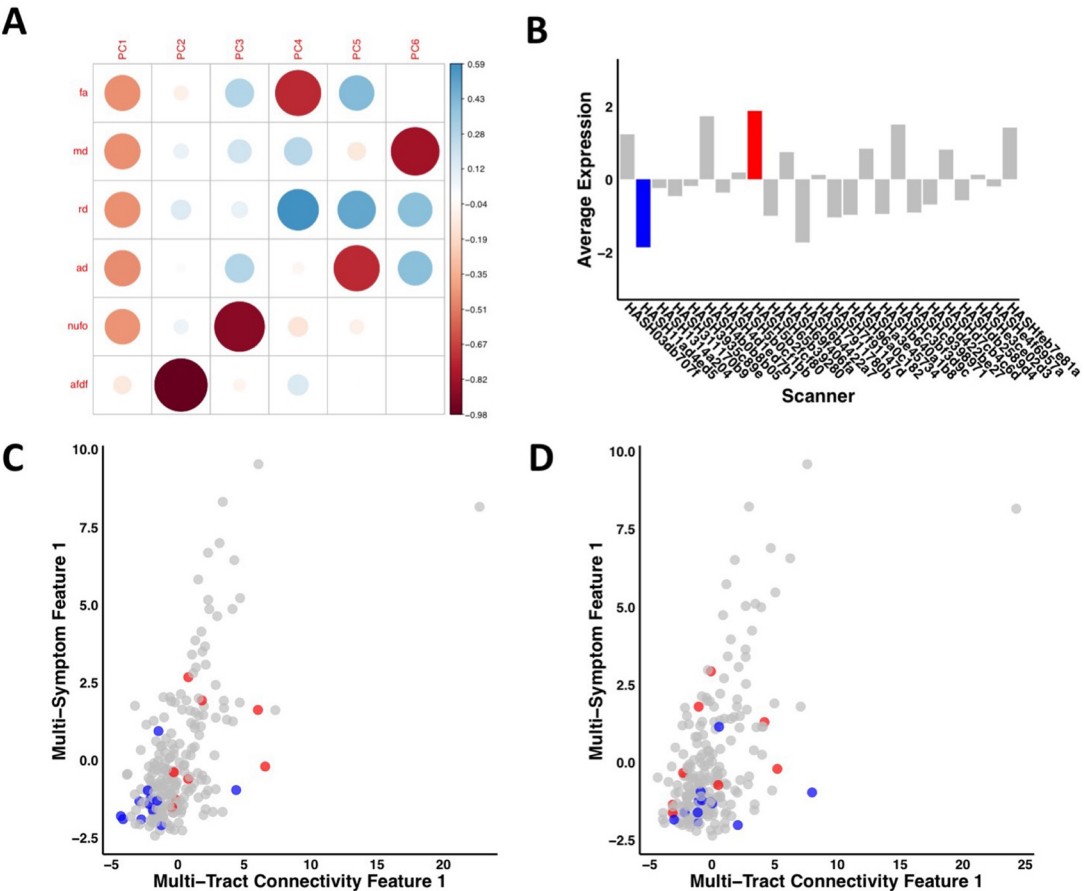

**Appendix 1—figure 1.** Illustration of the effects of regressing out scanner. (**A**) Weights of each diffusion measure for each principal component obtained after running a principal component analysis on data that was processed without regressing out scanner. (**B**) Barplot illustrating the expression of multi-tract connectivity feature 1 averaged across all participants for each scanner. The blue bar illustrates the scanner with the lowest multi-tract connectivity feature 1 expression, and the red bar illustrates the scanner with the second highest multi-tract connectivity feature 1 expression (the scanner with the highest expression only had one participant, so it was not chosen for illustrative purposes). (**C**) Scatter plot illustrating expression of multi-tract multi-symptom pair 1 from the microstructural complexity PLSc using data that was processed without regressing out scanner. The blue dots illustrate participants from the scanner with the lowest average multi-tract connectivity feature 1 expression, the red dots illustrate participants from the scanner with the second-highest feature 1 expression. These two groups are distinguishable in their multi-tract connectivity feature 1 expression. (**D**) Scatter plot illustrating expression of multi-tract multi-symptom pair 1 from the microstructural complexity PLSc using data where scanner had been regressed out. The same participants identified in scatter plot C are illustrated in scatter plot D. After regressing out scanner, these two groups are not distinguishable in their multi-tract connectivity feature 1 expression.

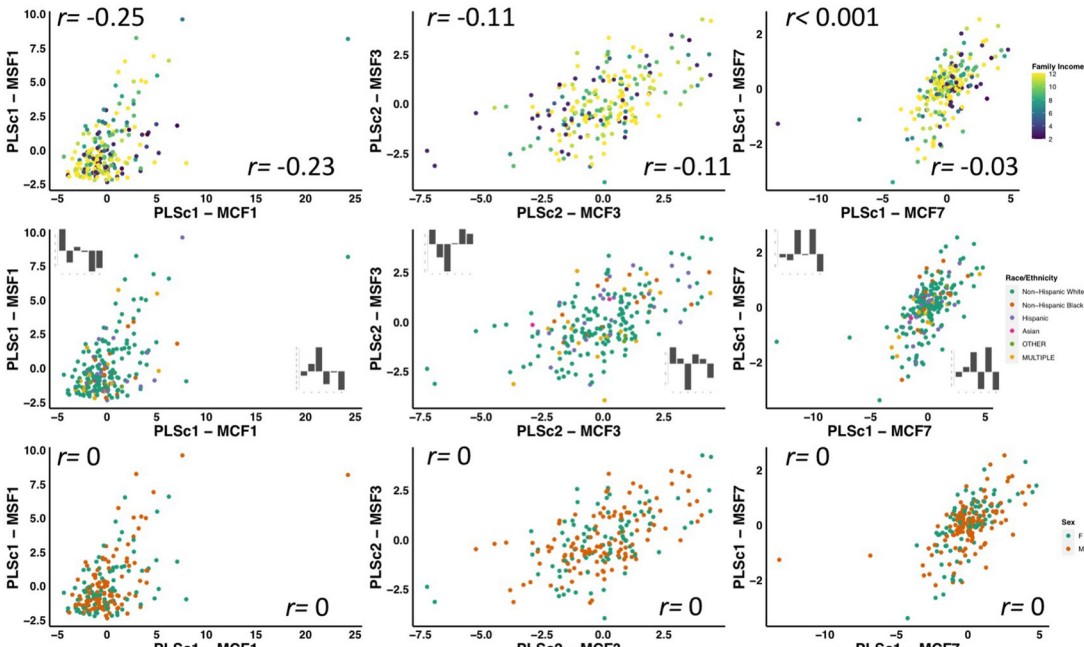

**Appendix 1—figure 2.** Scatter plots illustrating the expression of three multi-tract multi-symptom pairs (first column: pair 1 from the microstructural complexity PLSc, second column: pair 3 from the axonal density PLSc, third column: pair 7 from the microstructural complexity PLSc), color-coded by total family income (first row), race/ ethnicity (second row), and sex (third row). The upper and bottom rows illustrate the correlation coefficient for the expression of multi-tract features (over x-axis) and multi-symptom features (over y-axis) and variables representing family income (top), and sex (bottom). For race/ethnicity, correlations were performed between multivariate feature expression and dummy-coded variables representing each specific race/ethnicity. The correlation coefficients are presented in the bar graphs following the same order as listed in the color code.

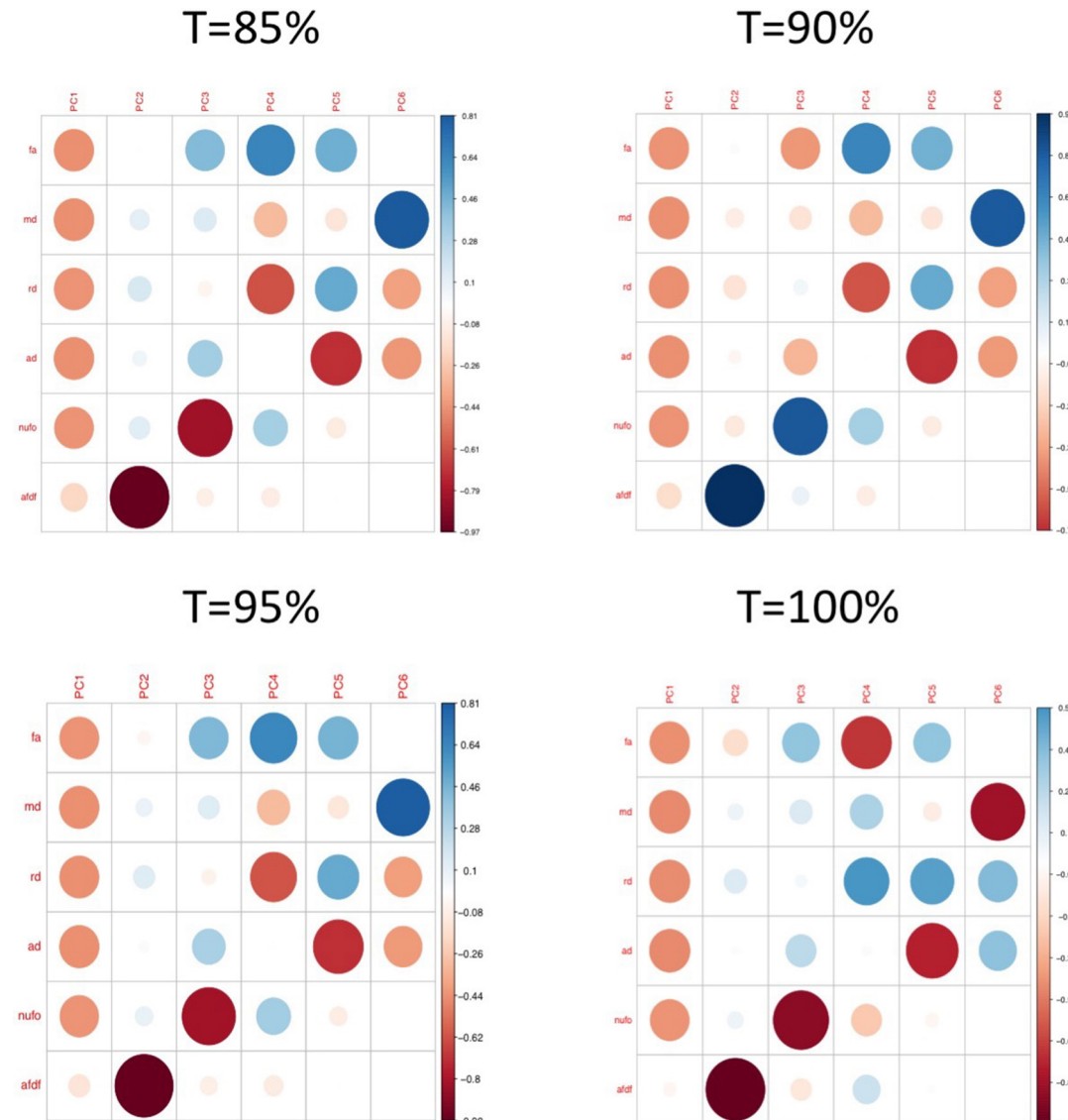

**Appendix 1—figure 3.** Plots illustrating the weights of each diffusion measure for each principal component for different connectome thresholds (85%, 90%, 95%, 100%). The interpretation of the first two principal components are consistent across thresholds.

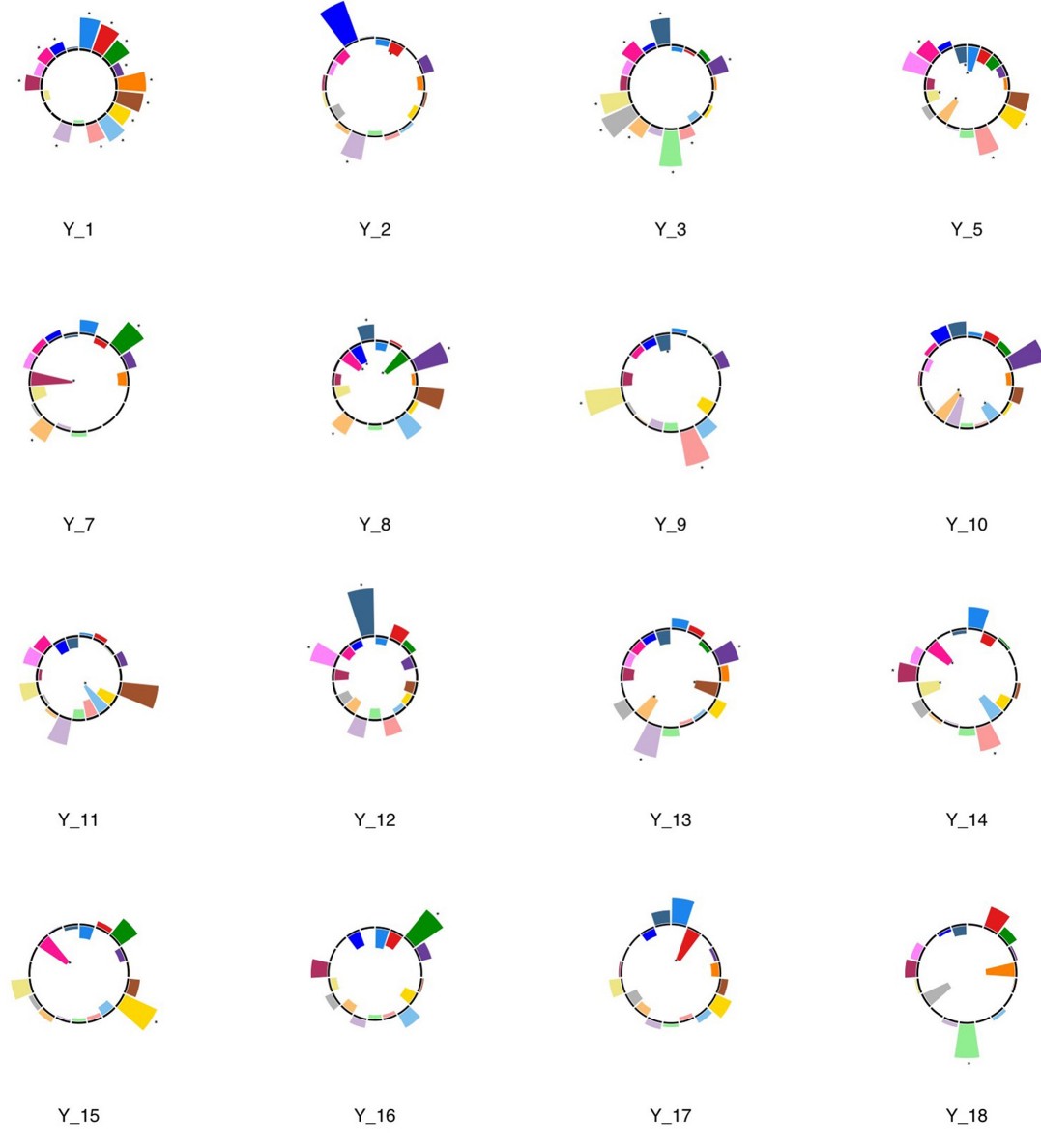

T=90%

**Appendix 1—figure 4.** Polar plots illustrating the weights of each symptom measure for every retained multi-symptom feature obtained from the microstructural complexity PLSc performed using all 19 symptom measures as well as connectivity features selected from connectomes thresholded at T = 90%. Black stars indicate symptoms that significantly contributed to the multi-tract multi-symptom pair based on bootstrapping analyses.

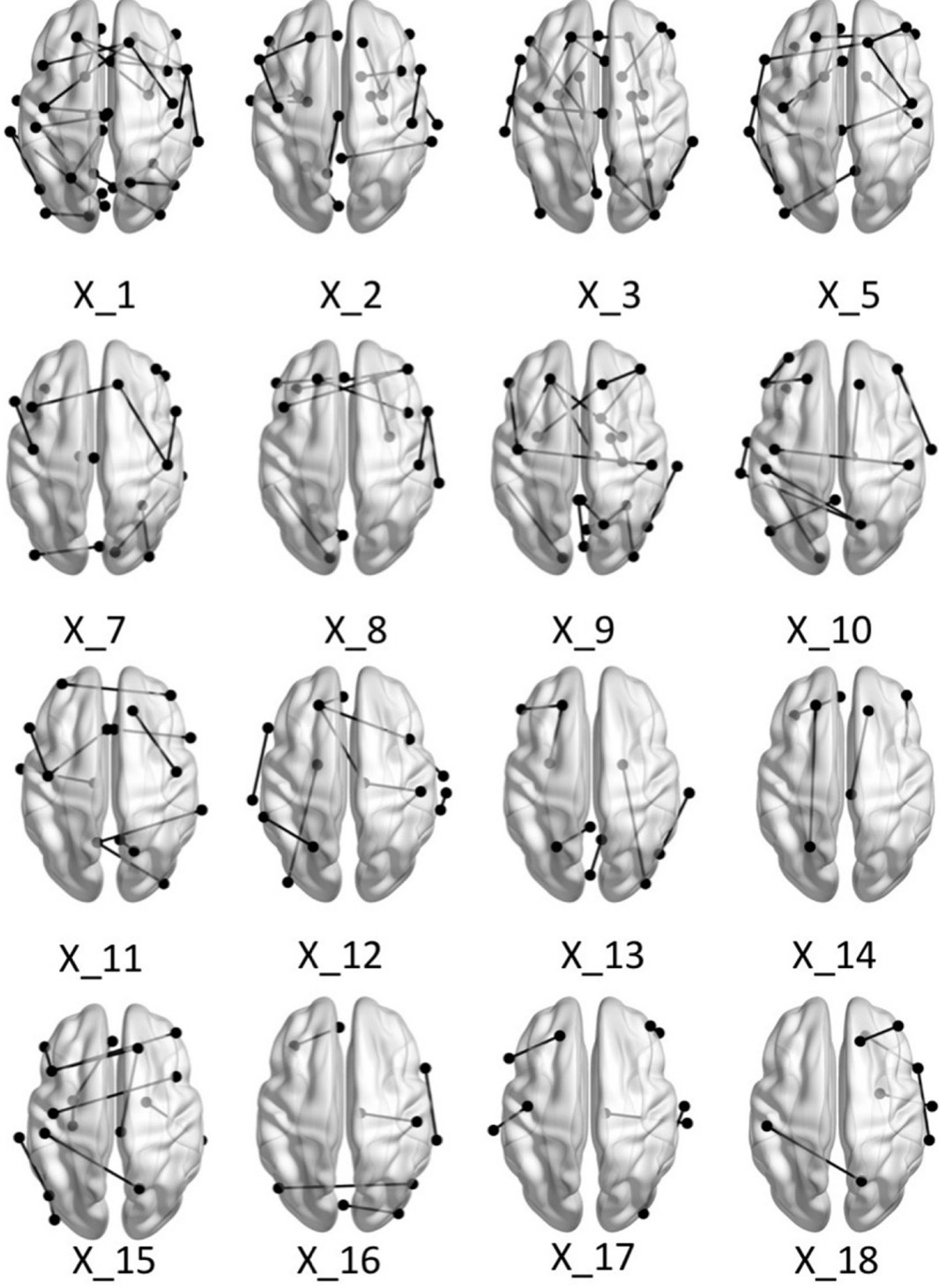

**Appendix 1—figure 5.** Brain graphs illustrating the connections that were found to be significant ($P < 0.05$ based on bootstrap analysis) for each of the retained multi-tract features from the microstructural complexity PLSc. Brain renderings were visualized with the BrainNet Viewer (*Xia et al., 2013*).

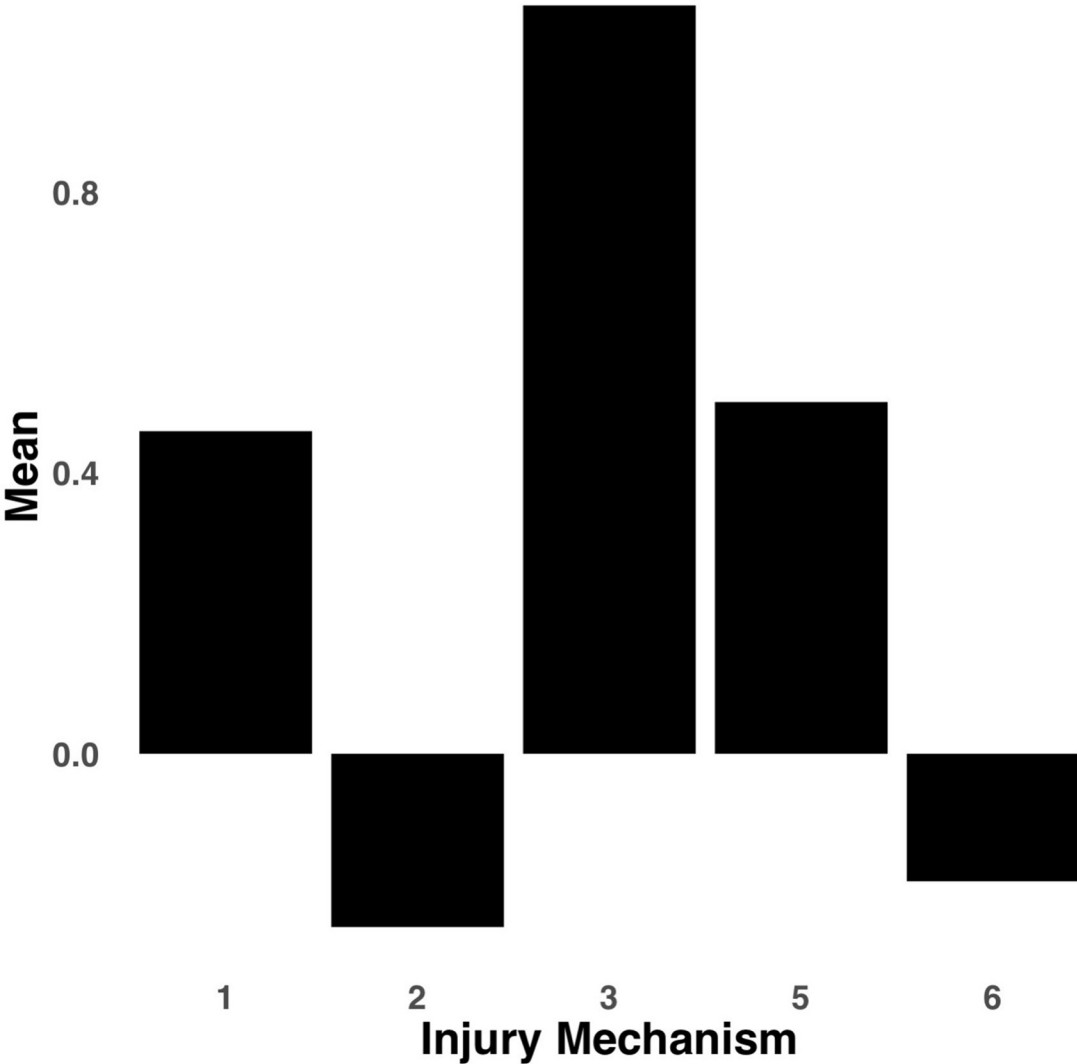

**Appendix 1—figure 6.** Bar graph illustrating the expression of multi-tract connectivity feature 2 from the microstructural complexity PLSc, averaged according to subgroups of participants defined by Injury Mechanism. 1: Fall/hit by object; 2: Fight/shaken; 3: Motor vehicle collision; 4: Multiple; 5: Unknown.

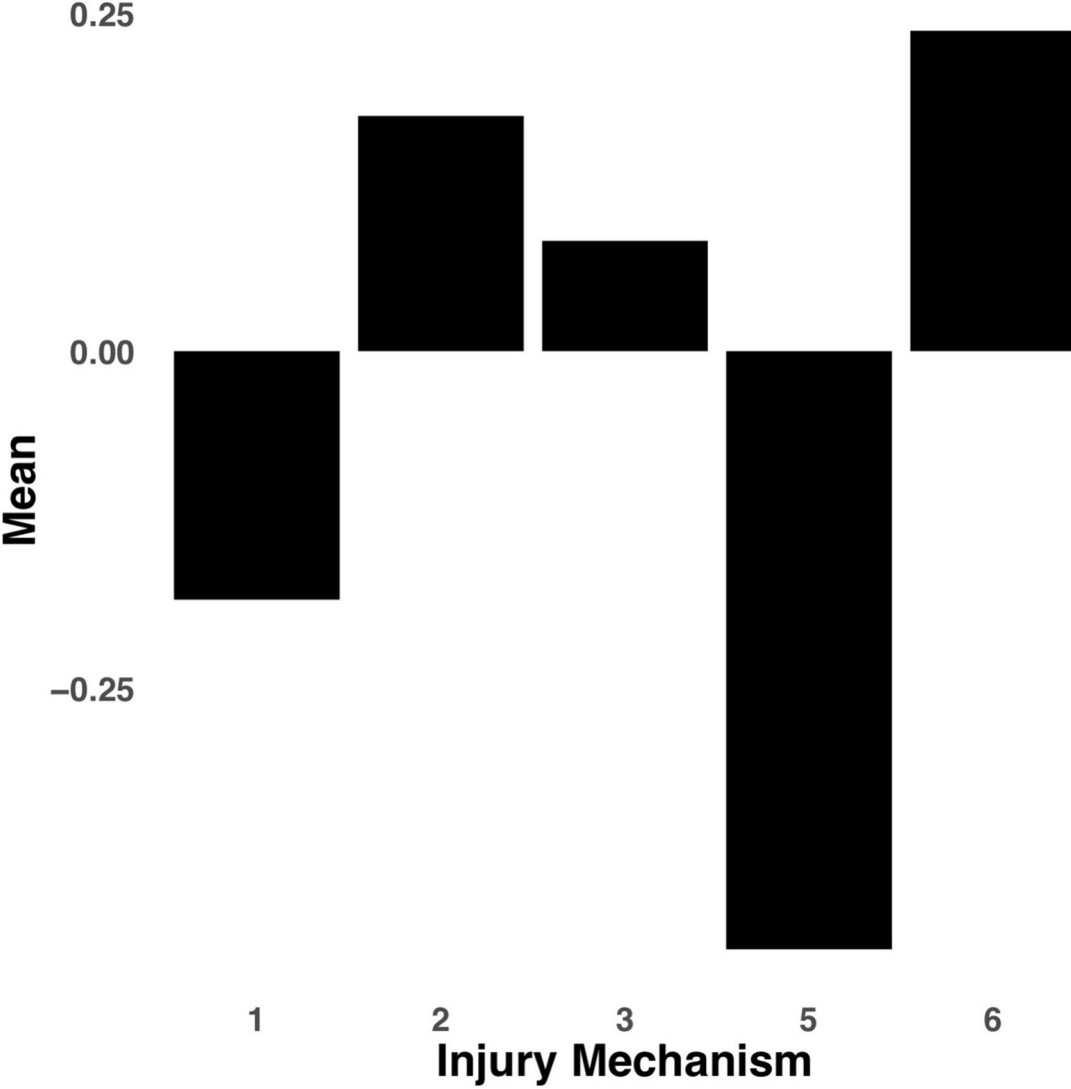

**Appendix 1—figure 7.** Bar graph illustrating the expression of multi-tract connectivity feature 15 from the microstructural complexity PLSc, averaged according to subgroups of participants defined by Injury Mechanism. 1: Fall/hit by object; 2: Fight/shaken; 3: Motor vehicle collision; 4: Multiple; 5: Unknown.

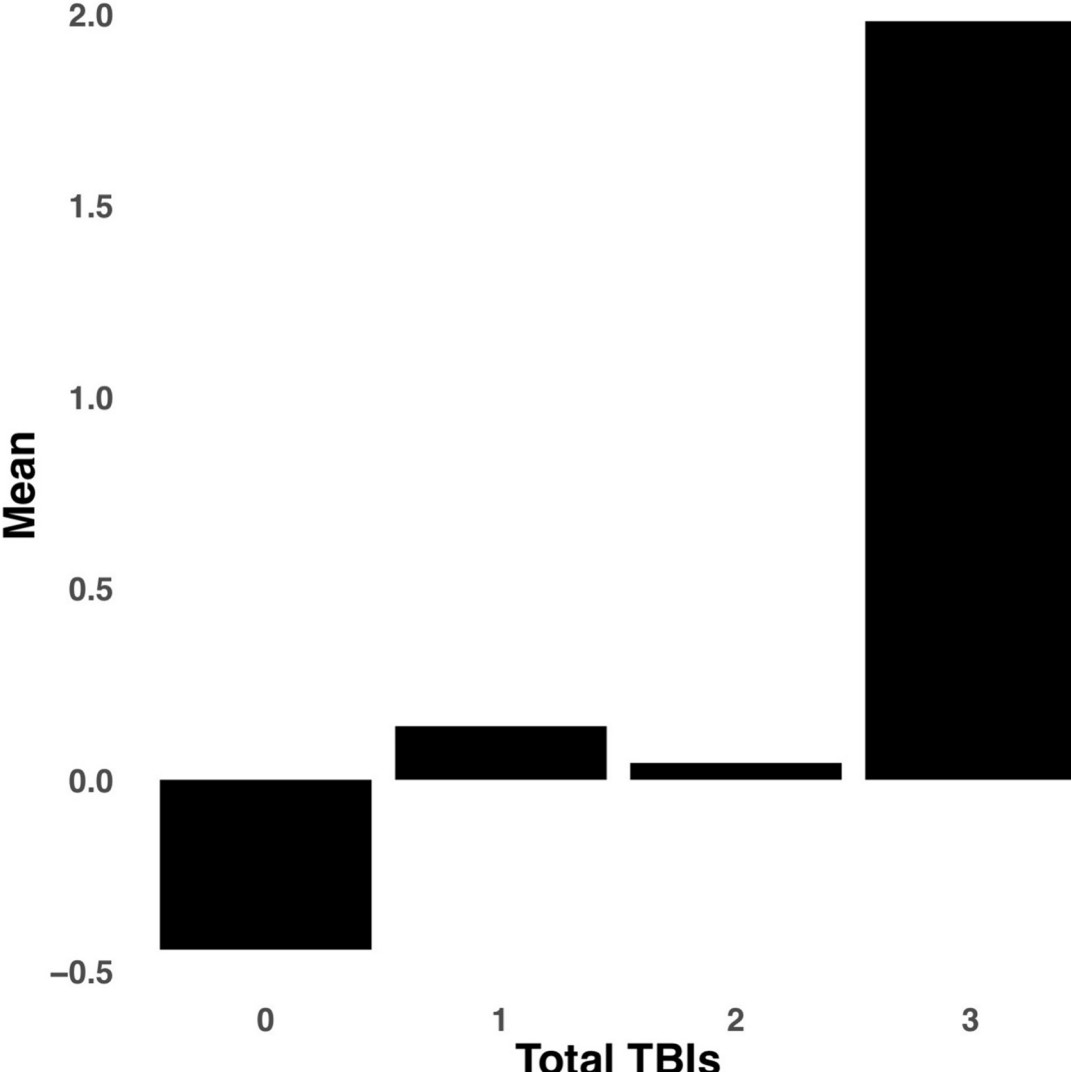

**Appendix 1—figure 8.** Bar graph illustrating the expression of multi-tract connectivity feature 15 from the microstructural complexity PLSc, averaged according to subgroups of participants defined by Total TBIs. 0: Unknown. Other numbers represent the total number of TBIs.

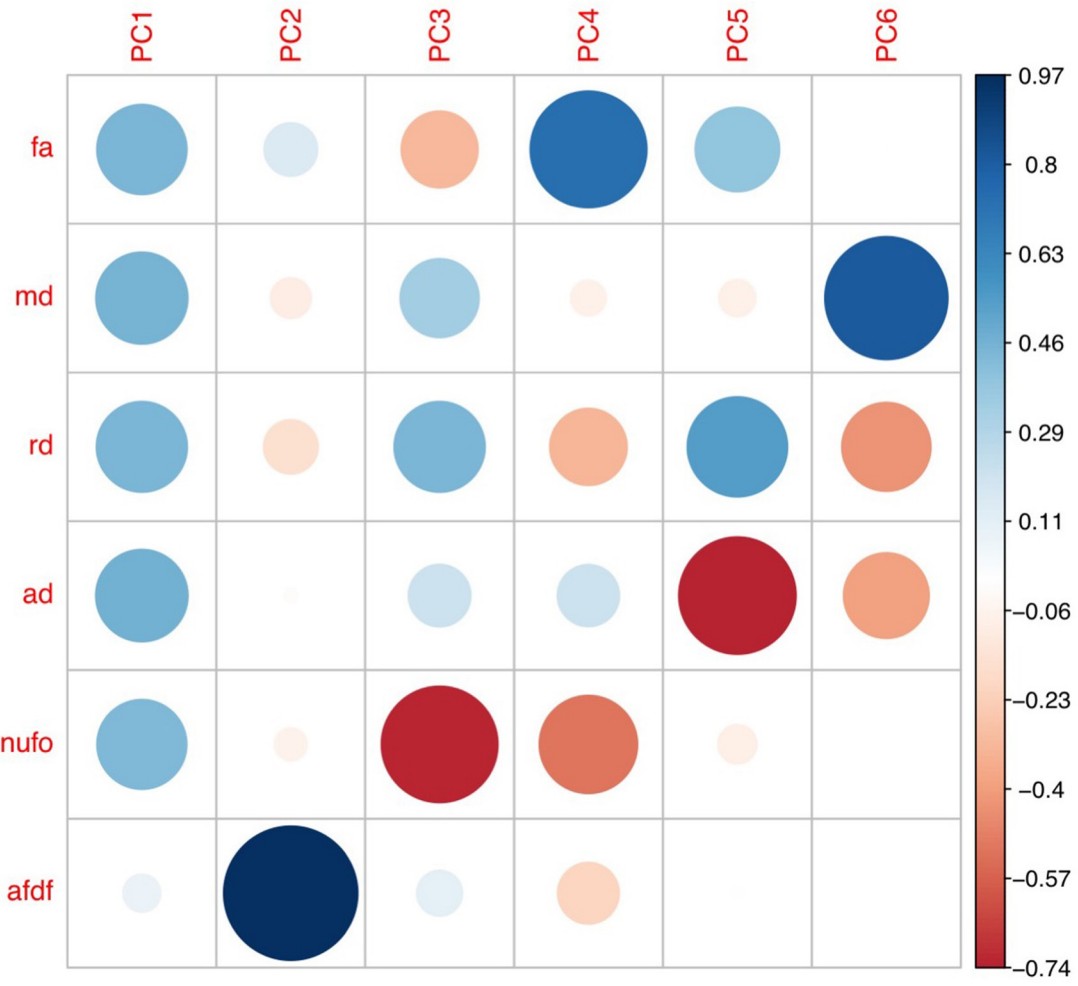

**Appendix 1—figure 9.** Plot illustrating the weights of each diffusion measure for each principal component for the PCA performed on the replication set data using a threshold of 90% during processing.

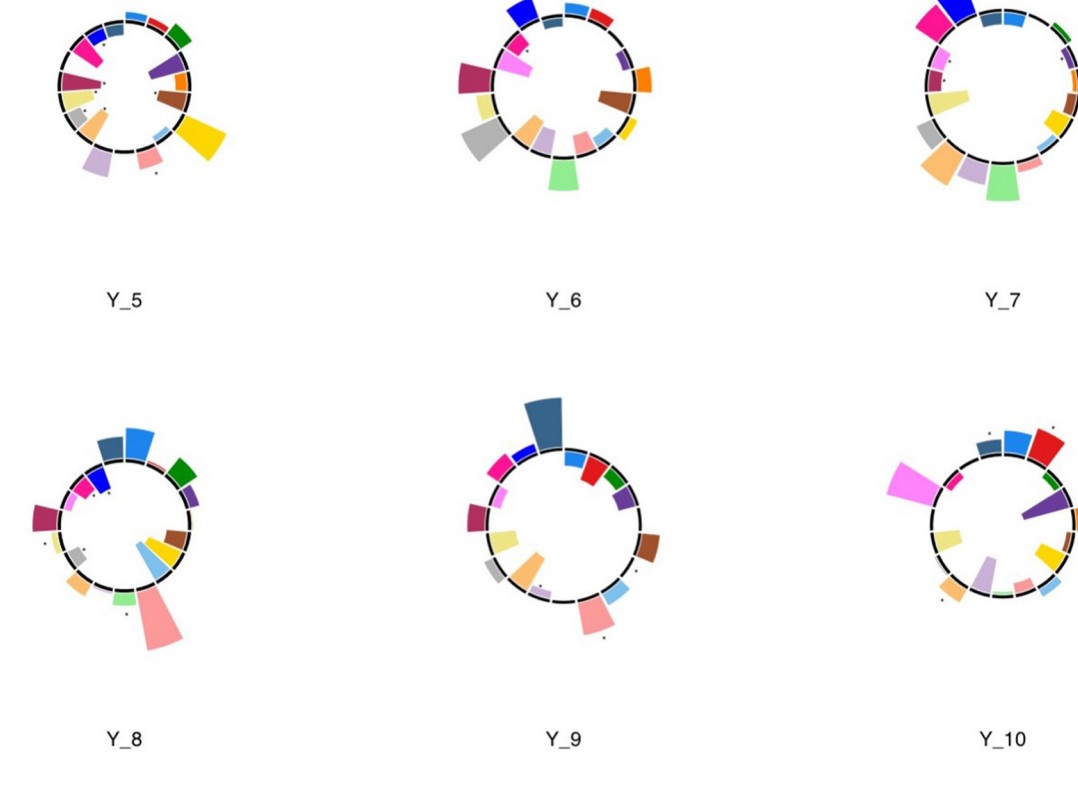

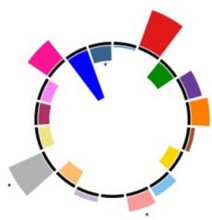

**Appendix 1—figure 10.** Polar plots illustrating the weights of each symptom measure for every retained multi-symptom feature obtained from the microstructural complexity PLSc performed using all 19 symptom measures as well as connectivity features selected from connectomes thresholded at T = 90% using the replication dataset. Black stars indicate symptoms that significantly contributed to the multi-tract multi-symptom pair based on bootstrapping analyses.

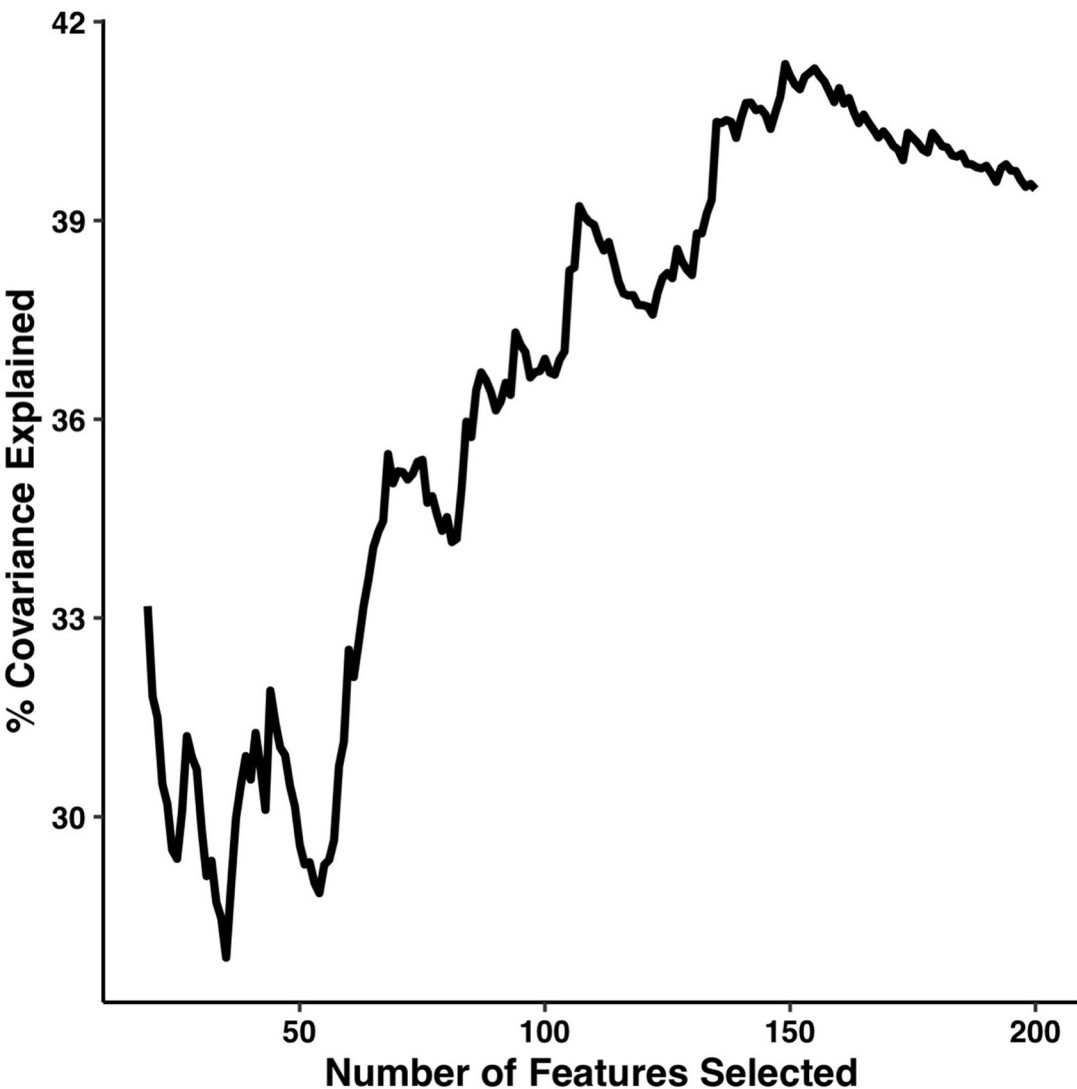

**Appendix 1—figure 11.** Percentage of covariance explained by the first multi-tract multi-symptom pair from the microstructural complexity PLSc as a function of the number of features selected from the univariate feature selection step. The connectivity features are selected based on decreasing strength of correlation with any symptom. The number of features tested ranged from 19 to 214.

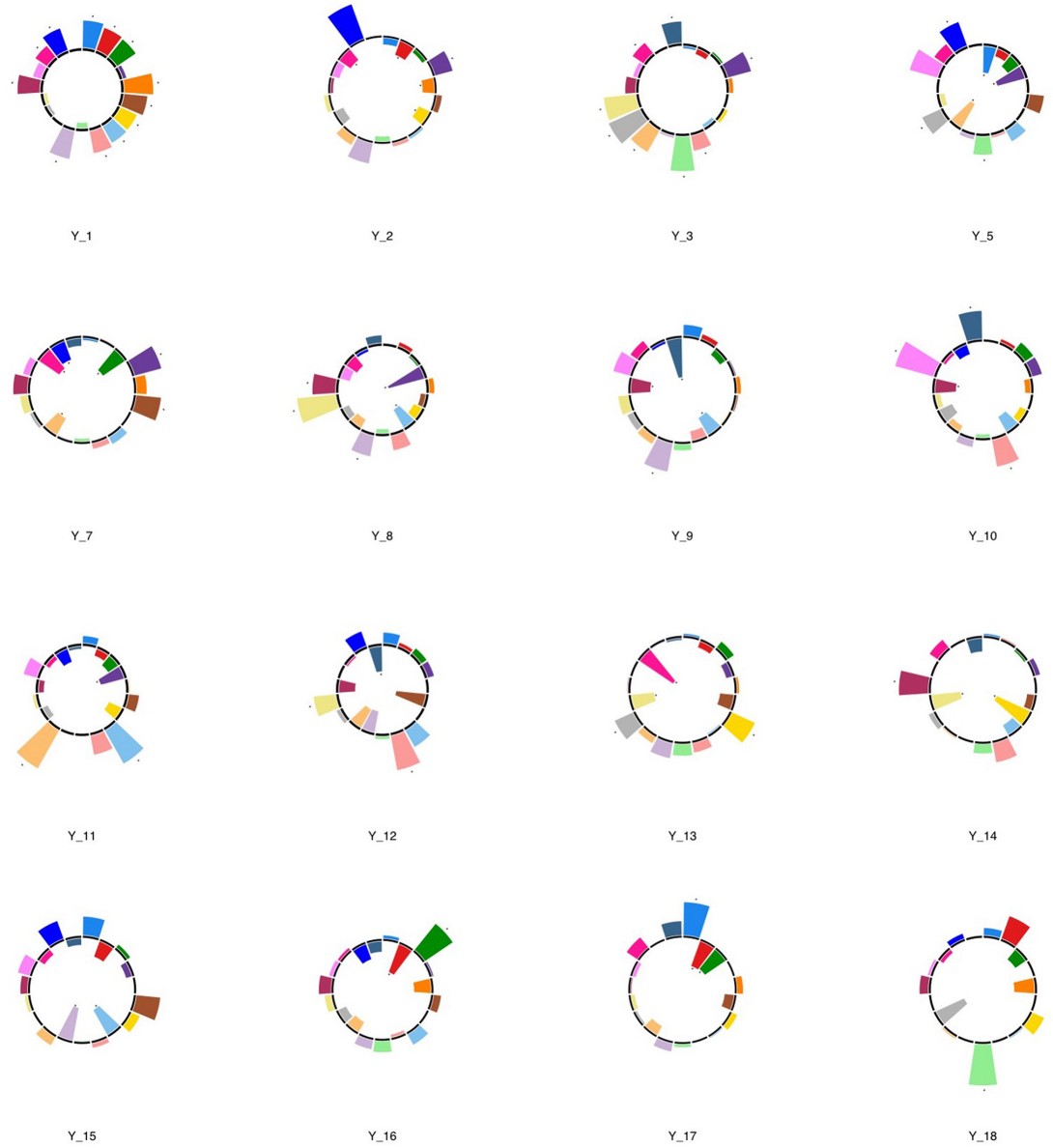

T=100%

**Appendix 1—figure 12.** Polar plots illustrating the weights of each symptom measure for multi-symptom features that were obtained from the microstructural complexity PLSc performed using all 19 symptom measures as well as connectivity features selected from connectomes thresholded at T = 100%. Only the multi-symptom features that were found to be significant in the corresponding PLSc performed at a threshold of T = 90% are shown here, for comparison with those multi-symptom features (*Appendix 1—figure 4*). All Black stars indicate symptoms that significantly contributed to the multi-tract multi-symptom pair based on bootstrapping analyses.

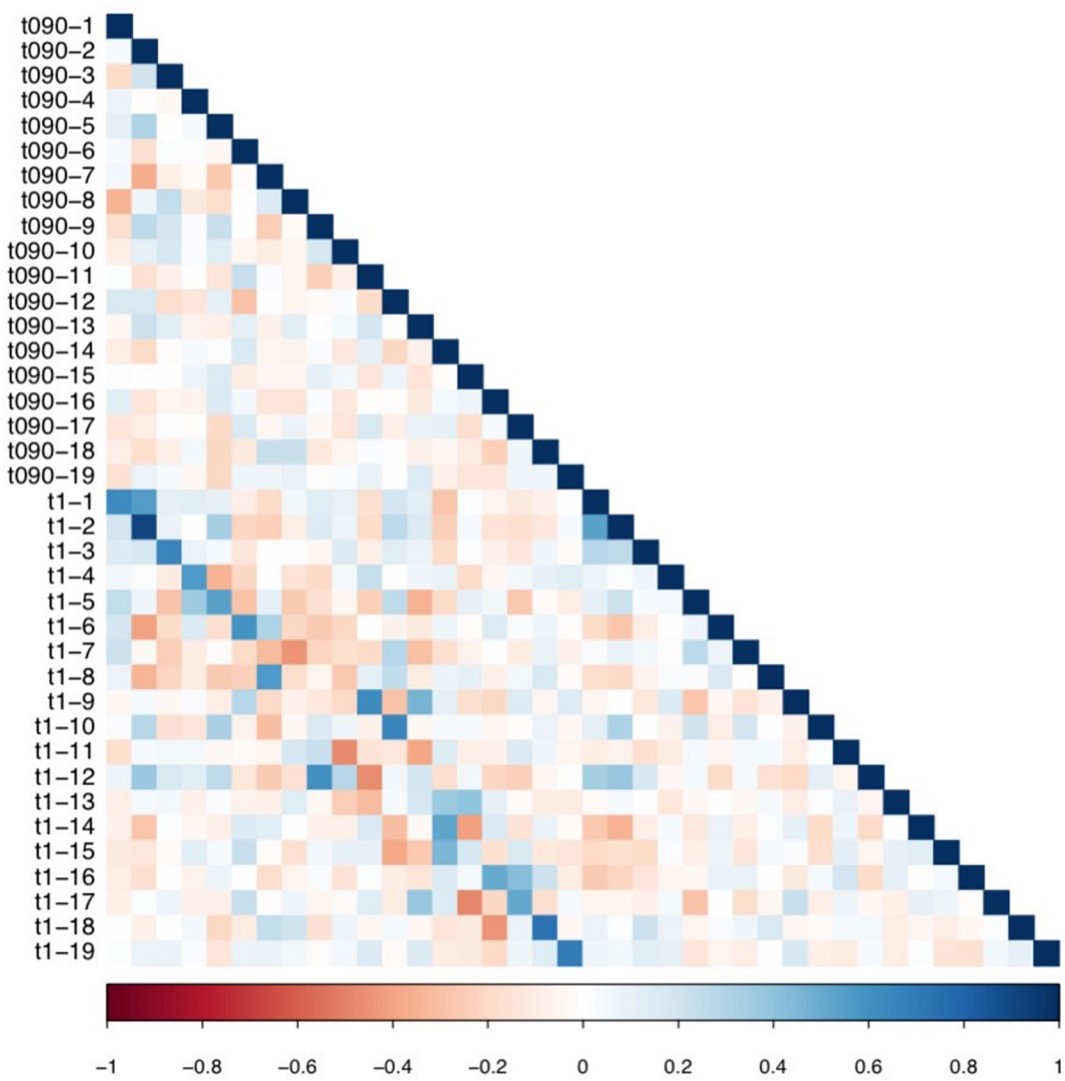

**Appendix 1—figure 13.** Matrix illustrating correlation coefficients between the expression of every pair of multi-tract connectivity features obtained from the microstructural complexity PLSc. The matrix illustrates the correlation between features obtained from the PLSc analysis performed on connectivity features obtained from the 90% and 100% thresholds. Given that this matrix is symmetrical, only the bottom triangular is shown. The main diagonals illustrate autocorrelations. These matrices illustrate how corresponding multi-tract connectivity features between thresholds (e.g.: multi-tract connectivity feature 1 from T = 90%, multi-tract connectivity feature 1 from T = 100%) are highly correlated.

