## [Editor Report]

This manuscript aims to address an important issue in the study of concussion: both the brain damage caused by concussion, as well as the behavioral symptoms that result vary widely across individuals. The study uses novel and interesting methods to relate multi-variate diffusion MRI data with multi-variate symptom-related data. The methods of analysis are sophisticated and well-executed and the results are quite interesting. The methods developed here could have a broad impact in their application to the many other neurological diseases that have heterogeneous outcomes.

---

## [Decision Letter]

**Decision letter after peer review:**

Thank you for submitting your article "Multi-tract multi-symptom relationships in pediatric concussion" for consideration by *eLife*. Your article has been reviewed by 2 peer reviewers, and the evaluation has been overseen by Drs. Shackman (Reviewing Editor) and Büchel (Senior Editor). The reviewers have discussed their reviews with one another, and the Reviewing Editor has drafted this to help you prepare a revised submission.

While the Reviewers highlighted several strengths of the manuscript, they also raised some potentially important concerns about the manuscript in its present form.

Essential revisions:

1. The need to quantitatively investigate heterogeneity

– One weakness of the study stems from a possible, and understandable confusion between heterogeneity and complexity. The study successfully targets the potential complexity of links between observed symptoms and underlying neurobiological mechanisms. Authors use a method that can identify multi-tract multi-symptom relationships, elaborating the inter-relatedness of different brain network features as well as the network of symptoms. Nevertheless, there is no analysis in the manuscript that explicitly studies heterogeneity, such as analyses on inter-subject variability (possibly comparing this to the variability within a healthy sample) or the grouping of subjects and variability of symptom/track profiles assigned to the groups. The manuscript would benefit greatly from such analyses.

– The only analysis that is related to the heterogeneity is Figure 2, which is a qualitative analysis. The black dots did not constitute a clear cluster and extended over a large portion of the scatter plot(s). I would appreciate a more quantitative analysis.

Better integration of goals and methods in the Results

Before reading the methods, I had a hard time understanding the results

– If authors want to keep the same structure, they should include short explanations within the results, describing briefly "what and how is done" in each section. For example, the section "Combined measures of white matter tract microstructure" starts with "The PCA yielded two biologically-interpretable components…", and it is not clear what features PCA was applied to and why. Similarly, it is not clear what PLSc stands for, or how and why it was used

– Comparisons between univariate and multivariate approaches should be explained better in the results. I had to read it twice and then go to the Methods section to understand what was done there. The explanation in the Methods section is much better and cleaner than what is given in the Results section.

2. The need to address potentially spurious connections

– There are concerns that some of the connections represented in the 76x76 connectivity matrix used here are spurious connections due to limitations of tractography that cannot be resolved with current methods, especially as the streamlines derived in tractography come close to cortex. These limitations can affect results even when using the state of the art tractography algorithm that was used here, (as demonstrated in Girard et al., 2020, Neuroimage, 221: 117201).

– The computational tractography arsenal includes several additional ways that could have been used to somewhat mitigate these concerns, including post-processing of the streamlines (e.g., with SIFT or LiFE), using surface-enhanced tractography methods (these may have been used here, but not mentioned) to increase the quality of tractography, and/or focusing on well-defined anatomical structures in the core of the white matter that can be well-detected using bundle recognition algorithms, and where tractography uncertainty due to the indeterminacy of the signal is of less concern. These issues seem particularly important given the substantial difference in the results between the 95% threshold and 100% threshold requirement for inclusion of an edge in the analysis

– Regarding my comments about tractography methods, I think that improvements to the tractography methods would improve the confidence in the results that are presented here. At the very least, one way to deal with these concerns is to discuss them, and to acknowledge that the estimation procedures used here produce not "tracts" but rather the edges of a connectivity graph, and that these may or may not represent anatomical units.

3. Concerns with the analytic approach and stability across feature selection

– Data were collected at multiple sites, and site differences should be considered. Authors say the scanner variable was regressed out from the connectivity data, which is fine, but it would be nice to see the site differences and the effect of regressing it out in a more systematic way.

– With the bootstrap analysis, how is it guaranteed that you get the same pairs each time? Please clarify.

– I wonder whether the procedure used to assess the statistical significance of the multi-tract multi-symptom pairs (lines 463-467) doesn't unfairly favor the ones in the unshuffled data. Shouldn't this procedure also include the preliminary feature selection step (lines 427-430) also in the shuffled data? It seems like it should, because some of the correlation in the PLSc is due to these univariate correlations, which could arise by chance in high-dimensional data.

– Uni vs multivariate 1 of 2. Is it any surprising or even interesting that the correlations between multi-tract and multi-symptom features were higher than all univariate correlations? With PLSc, latent modes are constructed (i.e., coefficients of tracts and symptoms are determined) to maximize the correlations. Thus, it is an expected outcome that those correlations would be higher than univariate correlations. Unless I am missing something, which is possible, I don't think this fact can be reported as a finding. Authors need to clarify this point, or revise the way in which these analyses and results are framed and interpreted.

– Uni vs multivariate 2 of 2. I also have a few questions regarding comparisons of univariate and mulivariate correlations: Line 122 – 124 makes a statement comparing these values, but doesn't provide quantitative information. Also, I might be misunderstanding, but don't the highest PLSc values have to be at least as high as the highest bi-variate correlation by construction? Are the differences between these values higher than would be expected by chance? In Figure 3, the two univariate brain graphs (red and blue) look very similar to each other. Are they as similar as they seem there? Are the multi-tracts implicated by PLSc1 also similar to the multi-tracts implicated in PLSc2?

– Why 200 features? Would the results change substantially if you use 100? Is there some way to select a reasonably small number of features empirically (e.g., based on a scree plot?).

4. Concerns with the approach to psychopathology

– One reviewer noted that: It is not the best practice to combine all possible psychiatric conditions within a single group and call it "adverse psychiatric outcome". This adversely affects the interpretability of findings. Knowing that one of the multi-tract features correlates with the adverse psychiatric outcome is not much informative and clinically valuable. Authors need to devise a more appropriate approach. At minimum, they need to provide a compelling motivation for the current approach.

– While the other reviewer noted that: Some comments regarding the participants that were classified as having psychopathology (marked in black in the scatter plots in Figure 2). It is not clear why these participants were selected based on the CBCL scores, which are indirect measures of psychopathology, rather than the more direct KSADS psychopathology scores? If there are 55% with psychopathological KSADS score, that should provide more than 28 participants marked here. I am also not sure that I agree with the statement that the 28 individuals highlighted in black would be grouped together in clinical studies, given their psychopathology probably varies widely. It is not clear to me whether the paragraph on lines 125 – 131 claims that these participants are different from the other participants in terms of their multi-tract/multi-symptom scores or are not different from the other participants. It seems to me that they might be different based on the first multi-tract/multi-symptom scores (which are both slightly higher in this group), but there might also not be enough subjects to demonstrate this difference statistically. I think that this would also be consistent with the subsequently-described overlap between this score and the univariate scores (lines 132-141).

5. Concerns with claims of replication

– The presentation of the results doesn't make it clear whether the structure that was identified in the discovery dataset does in fact replicate in the replication set. Similarly, it hard to judge what to make of the differences in the results of the logistic regression between the samples. If I understand correctly, the logistic regression in the discovery sample indicates as predictive of psychopathology features that are different than the feature that was indicated in the discovery set. But this might lead us to wonder: How should we interpret this? For example, if I had a measurement from a new child that has had a concussion, which of the two models would I use to predict whether they might develop a psychopathology? The one derived in the discovery set or the one remaining feature that was indicated in the discovery set? Authors need to clearly explain the logic and evidence that supports their claim.

– Regarding my comments about the comparison to the replication set, I think that a few direct comparisons might help. For example, how much of the covariance in the discovery set is explained by the same multi-tract/multi-symptom pair that explained more than 40% of the covariance in the microstructural complexity PLSc in the discovery set? Or, possibly even more rigorous: if you ran a PLSc on the discovery set how similar would the feature sets be? I realize that this might not be possible, given the smaller sample size and the choice of 200 features in the feature-selection phase.

6. The need for a more sober and precise Discussion

– Authors say in the discussion that "these results recapitulated well-known findings from the concussion literature and revealed new insights…". Yet it is not clear what those well-known findings are and what new insights were revealed

– Overall, the narrative in the Results section would benefit from a better organization

– Authors should consider reporting more quantitative results, including tables to summarize findings, and precise comparisons of their results with published work.

[Editors' note: further revisions were suggested prior to acceptance, as described below.]

Thank you for resubmitting your work entitled "Multi-tract multi-symptom relationships in pediatric concussion" for further consideration by *eLife*. Your revised article has been evaluated by Drs. Shackman (Reviewing Editor) and Büchel (Senior Editor), in close consultation with the 2 expert Reviewers.

Both Reviewers expressed some enthusiasm for the revision, noting that

"I think the authors did a great job of reorganizing the manuscript and addressing almost all issues raised by reviewers," and

"The authors have made substantial changes to the manuscript in their revision and extensive additional analysis has been added in response to the comments made on the previous version. I believe that many of these changes help improve the article, and could provide some more clarity about the robustness of the method proposed and its application. In particular, new analyses that describe the robustness of the approach, and new comparisons with the replication sample could help establish confidence in the conclusions."

Nevertheless, there was consensus that additional major revisions are required, as detailed below

Abstract

– In the abstract, results are limited to findings in ADHD and the replication set, but both are rather secondary/supporting results. There is virtually nothing about the main findings. How do certain combinations of symptoms relate to certain combinations of tracts? I would give specific results in the abstract with the highest importance, possibly mentioning specific symptom-tract couples, rather than sharing very generic comments, such as "These clinically-informative, cross demographic multi-tract multi-symptom relationships recapitulated well-known findings…" Such text is more suitable for the Conclusions.

– The current Conclusions in the abstract is again extremely generic and does not fit well the findings. The manuscript does not include much on "stratification strategies" or "predictive biomarkers". It would be better to discuss how specific findings of the current study would fuel future studies.

Abstract/Introduction/Results/Discussion

– Overall, the main goal (not methodological but rather scientific) of the study should be clearer, both in the abstract and other sections (especially, Results). It should be clear for the reader what they learn new about concussion/TBI and how those findings can be used in practice (e.g., in research, in clinics, …).

– The new focus on ADHD is reasonable, but I would have liked to see more treatment in the manuscript of the relationship between ADHD and concussion and what we learn from the current analysis (if at all) about this relationship. As it is right now, the choice to analyze ADHD is not strongly motivated by the Introduction, and the interpretation of the results does not clearly help us understand what we learned here, beyond the fact that manifestations of concussion in participants with ADHD are heterogeneous.

Results/Discussion

– The text in the introduction (page 3, lines 87-96), nicely summarizes the main contribution: "… display a common symptom but will have overall different symptom profiles and different white matter damage profiles… a variety of white matter structure alterations ("multi-tract") may be related to a variety of symptoms ("multi-symptoms") in different ways." This is very clear and interesting. What is missing is that results do not demonstrate this interesting phenomenon clearly. There is one example in Discussions (page 21, lines 600-6004), but it is limited to ADHD symptoms. It would be great if more results/discussions were included about specific symptoms and how they are associated with specific tracts. If possible, talk more about symptom-tract pairs; specifically showing how same/similar symptoms might be related to different tracts. Or how different combinations of symptoms are linked to different tract combinations. This is the most important part of the manuscript, and the results should reflect this fact.

– The inclusion of new control analysis that uses COMMIT to identify spurious connections is great. I think that the added text on lines 666 – 690 raise important caveats and future directions, which will give the readers not as familiar with the details of dMRI research important context to understand the work. On the other hand, it is very hard to judge the similarity between the results in Appendix 1 Figure 4 and Appendix 1 Figure 14. In particular, there seem to be no black stars in Figure 14 (despite the caption mentioning them). Does this mean that no MCF symptom features are significant in this analysis when COMMIT is used? That would be a real problem for the robustness of this approach to issues related to tractography that are mentioned in this part of the discussion.

– Figure 3 (and the comparison between univariate and multivariate measures) is still hard to understand.

A) The authors stated in the response to the reviews that "Figure 3 has been modified, illustrating now two univariate brain graphs (contrasting PC1 and PC2 scores as a function of ADHD diagnosis), and three multivariate brain graphs (PLSc1 – pair 1, PLSc2 – pair 3, and PLSc1 – pair 7)." I see the latter, but not the former. Figure 3 also has two correlation coefficients inset in each plot, (e.g., "r=0.38" and "r=0.26" in the right-hand side of the first row). What do these coefficients refer to?

B) Relatedly, the methods say (starting on line 279) that "Using threshold of p<0.05, we computed univariate comparisons of connectivity (PC scores) between individuals with and those without a diagnosis of ADHD thus identifying putative "ADHD-related" univariate connectivity features." This sounds reasonable, but I don't see the conclusion from this analysis in the Results section. Is there a univariate difference between the PC scores of individuals diagnosed with ADHD and those that were not? Was this comparison conducted only within the group that had a psychopathological KSADS scores? In other words, on line 273 in the phrase "We divided the sample", does the word "sample" here refer only to (how many?) participants who had any psychopathology? Is only their data presented in Figure 3? Or is that all 214 of the discovery set participants?

C) Furthermore, I am still struggling to understand whether I should interpret Figure 3 as indicating a difference between the subjects marked in black (now subjects diagnosed with ADHD) and the other subjects in the multivariate analysis. As far as I can tell, these individuals (black dots in the scatter plots) are not different in any discernible way from the other participants in these plots (colored dots) -- certainly not in pair 3 (the middle line), where the black dots and colored dots seem completely intermingled. Is there some way to evaluate the difference between the groups across all of the multi-symptom/multi-tract features?

D) In the same section, what is the statistical test that supports the statement "Further, we computed correlations between the expression of these multivariate features and clinical subgroup membership, and found weak, albeit significant correlations"? It would be good to apply such a statistical test to all of the multi-tract multi-symptom features, and report the results of this test as part of this section, just as it would be good to report the results of a direct test of the "univariate" PCA differences. I believe that this would provide a stronger assessment of multivariate vs. univariate approaches.

– This might just be a wording issue, but I don't understand the following statement (L413- 416): "Despite forming a clinical subgroup based on a gold-standard measure of psychiatric pathology, these individuals were differentiable by their expression of multi-tract and multi-symptom features". Maybe the problem is the word "differentiable"? If this is really what you meant, may I suggest something a bit more direct, such as "…, these individuals were heterogeneous in their expression of multi-tract and multi-symptom features"?

– The analysis on the replication set is much improved in the revision, but questions still remain here as well. First, are 12% and 33% of the covariance that were explained in the replication set, using MCF1 in both PLSc analysis more than would be expected by chance? I think that this could be established through a permutation test in the replication set or some other statistical test. Another question that arises from the comparison is that it looks as though the first set of symptom weights obtained in the PLSc for microstructural complexity in the replication set (Y_1 in Appendix 1, Figure 11) is completely inverted relative to the set of weights in this PLSc conducted on the discovery set. Is this anti-correlation somehow trivial? Or is it significant (either statistically or conceptually)? What do we learn from the fact that the features selected in the discovery and replication sets are quite different? Does performing the COMMIT analysis in both the discovery and replication set improve the situation? Surely the feature selection step is affected by the 35% of connections that COMMIT finds to be spurious in the discovery set? Why not remove them with COMMIT and proceed from there? I realize that you can't try all the different permutations of analysis pipelines, but using COMMIT seems like a principled way to improve the robustness specifically of this part of the pipeline. I also think that the probe for correlation between expression of the multi-symptom multi-tract features and dataset is clever, but some of the explanations that now appear in the response to the reviews (and particularly about this point) should probably appear in the Discussion section, to help readers understand what the point of this analysis is. In general, I wonder whether, now that lines 618 – 626 have been removed, the authors could discuss the implications of the analysis of the replication set in the Discussion. I think that having a replication set analysis is a real strength of the paper.

Discussion

– While it is discussed in Discussions how a multivariate approach better captured ADHD-related symptoms than a univariate approach, this is not clear in Results. When I first read it in the discussions, I was surprised but then remembered the parts in the results (scattered all around) that supported this claim. I would make this comparison clearer under the "Results / Multivariate vs Univariate Approaches" section.

---

## [Author Response]

Essential revisions:1. The need to quantitatively investigate heterogeneity– One weakness of the study stems from a possible, and understandable confusion between heterogeneity and complexity. The study successfully targets the potential complexity of links between observed symptoms and underlying neurobiological mechanisms. Authors use a method that can identify multi-tract multi-symptom relationships, elaborating the inter-relatedness of different brain network features as well as the network of symptoms. Nevertheless, there is no analysis in the manuscript that explicitly studies heterogeneity, such as analyses on inter-subject variability (possibly comparing this to the variability within a healthy sample) or the grouping of subjects and variability of symptom/track profiles assigned to the groups. The manuscript would benefit greatly from such analyses.

Thank you for this helpful comment. We agree that our use of the term “heterogeneity” in this context was not as clear as we thought. What the study is meant to target is specifically heterogeneity in how white matter structure alterations (or combinations of them) relate to symptoms. We argue that there are multiple ways that white matter abnormalities are related to symptoms. It is these different relationships that we call multi-tract multi-symptom relationships, and these are the relationships that we wanted to study. Multi-tract multi-symptom relationships reflect heterogeneity of the disease, rather than that of individual subjects. Although “disease heterogeneity” is a factor in “disease complexity”, when revising the manuscript we opted for the first term because it more clearly conveys the multiplicity of structure/symptom relationships that we are aiming to study.

We have modified the manuscript clarify this point and more accurately reflect the primary goal of our investigation.

Modifications:

“This example illustrates an additional, hitherto ignored source of heterogeneity, whereby a variety of white matter structure alterations (“multi-tract”) may be related to a variety of symptoms (“multi-symptoms”) in different ways. This disease-specific type of heterogeneity will henceforth be referred to as multi-tract multi-symptom heterogeneity for brevity.” (lines 84-88)

“In the present study we leveraged novel dMRI methods and a double-multivariate approach to parse heterogeneity in the relationship between white matter structure and symptoms in a large sample of previously-concussed children.” (lines 590-592)

“In contrast to this nascent literature focusing on inter-subject heterogeneity in symptoms or white matter structure, our approach focuses on disease-specific heterogeneity in structure/symptom relationships.” (lines 664-666)

– The only analysis that is related to the heterogeneity is Figure 2, which is a qualitative analysis. The black dots did not constitute a clear cluster and extended over a large portion of the scatter plot(s). I would appreciate a more quantitative analysis.

We fully agree that the black dots did not form a clear cluster, and that is in fact one of the core arguments that we were attempting to make. What would have ostensibly been a cluster based on more rudimentary features like Child Behavior Checklist scores are in fact quite different with respect to their expression of these multi-tract multi-symptom features.

To address your comment (along with comment 4c), we changed the variable we used to form these psychopathology subgroups and implemented a quantitative analysis. Specifically, we used the Schedule for Affective Disorders and Schizophrenia for School-Aged Children (KSADS) questionnaire to identify children with a diagnosis of ADHD. We modified Figure 2 scatter plots to indicate children with ADHD diagnoses as black data points (see modified Figure 2 scatter plots). We then computed Pearson correlations between expression of multi-tract features and a binary score for having an ADHD diagnosis or not, and did the same for multi-symptom features. This simple but effective approach allowed us to quantitatively assess how much the expression of multi-tract multi-symptom features is related to having a diagnosis of ADHD. If individuals with ADHD formed a cluster in their expression of multi-tract and multi-symptom features, we would have expected a strong (i.e.: close to 1 or -1) correlation between their expression and a binary variable indicating the presence of an ADHD diagnosis. For this particular analysis, we found weak correlations (highest was 0.3) that were nonetheless statistically significant (see modified Figure 2). Modifications in the manuscript related to this new analysis can be found in the introduction (lines 97-100), methods (lines 267-278), results (lines 417-436), and discussion (lines 626-636).

Modifications:

“Analyses comparing clinical subgroups defined by the presence of attention-deficit/hyperactivity disorder showed that multi-tract multi-symptom analyses identified disease-specific connectivity patterns that were missed by single-tract single-symptom approaches.” (lines 97-100)

“To compare information captured by the PLSc and univariate approaches, we first divided participants based on whether they had obtained a psychiatric diagnosis. Parents of all participants completed the Kiddie-Schedule for Affective and Psychiatric Disorders in School Age Children (KSADS), a gold-standard tool, to assess the presence of pediatric psychiatric disorders.^38^ We divided the sample into clinical subgroups based on whether they had obtained a diagnosis of attention-deficit/hyperactivity disorder (ADHD). We selected this diagnosis because its behavioural manifestations can be easily related to some of the presently-studied concussion symptoms (e.g.: attention problems). It was also the second-most common diagnosis in our sample (33/214). Using a threshold of p<0.05, we computed univariate comparisons of connectivity (PC scores) between individuals with and those without a diagnosis of ADHD, thus identifying putative “ADHD-related” univariate connectivity features.” (lines 267-278)

“To compare univariate and multivariate approaches, we identified 33 individuals with diagnoses of ADHD obtained from the KSADS.^9^ These individuals are shown in black in Figure 3 (scatter plots). Despite forming a clinical subgroup based on a gold-standard measure of psychiatric pathology, these individuals were differentiable by their expression of multi-tract and multi-symptom features. Further, we computed correlations between the expression of these multivariate features and clinical subgroup membership, and found weak, albeit significant correlations (see Figure 3 scatter plots).

We compared microstructural complexity and axonal density scores across all 200 connections between individuals with and without an ADHD diagnosis, and calculated the percent overlap between each multi-tract connectivity feature and the set of tracts found to be significant in univariate comparisons (ostensibly “ADHD-related” tracts). The percent overlap scores are presented in Figure 4. Notably, the highest overlap occurred with multi-tract connectivity feature 1 (9-13%) from both PLSc analyses. The second-highest overlap was with multi-tract connectivity feature 7 from the microstructural complexity PLSc (6%). Its corresponding multi-symptom feature represented almost exclusively attention problems. Both the univariate “ADHD-related” brain graphs of microstructural complexity and of axonal density implicated several callosal tracts (16/64 and 31/76 respectively). Multi-tract feature 1 also implicated several callosal connections (13/39). However, fewer callosal connections were implicated in multi-tract connectivity feature 7 (2/10) (Figure 4).” (lines 417-436)

“Our results are consistent with this idea: univariate comparisons between a clinical subgroup defined by a diagnosis of ADHD and the rest of the sample identified connectivity features that mostly overlapped with the first multi-tract multi-symptom pair obtained from both PLSc analyses. These pairs, which accounted for the most covariance, reflected general problems and not specifically ADHD. Both these pairs and the univariate “ADHD-related” connections implicated several callosal tracts. Instead, a multi-tract multi-symptom pair that more uniquely represented attention problems implicated fewer callosal tracts. This pair implicated right-lateralized connections, consistent with prior literature showing more left-right asymmetry in FA in children with ADHD compared to controls.^45,46^ These results suggest that univariate comparisons in concussed children, even when performed in such a way as to identify a diagnosis-specific set of connectivity features, identified only the most consistent group-level connectivity differences at the expense of more symptom-specific idiosyncratic ones.” (lines 623-636)

2. Better integration of goals and methods in the ResultsBefore reading the methods, I had a hard time understanding the results– If authors want to keep the same structure, they should include short explanations within the results, describing briefly "what and how is done" in each section. For example, the section "Combined measures of white matter tract microstructure" starts with "The PCA yielded two biologically-interpretable components…", and it is not clear what features PCA was applied to and why. Similarly, it is not clear what PLSc stands for, or how and why it was used.

Thank you for pointing out these issues in the previous version of our manuscript. We have modified the structure of the manuscript so that the results follow the methods and have added brief recapitulations of the particular techniques in the Results section.

Modifications:

“The PCA applied across dMRI measures from all connections yielded two biologically-interpretable components that together explained 97% of the variance in measures (Figure S3).” (lines 390-392)

“To compare univariate and multivariate approaches, we identified 33 individuals with diagnoses of ADHD obtained from the KSADS.^9^” (lines 417-418)

“We first analyzed the amount of connectivity/symptom covariance explained by each of the multi-tract/multi-symptom pairs." (lines 484-485)

“We then projected the replication dataset onto the latent spaces obtained using the discovery set, and ran correlations comparing multi-tract multi-symptom feature expression against a binary variable indexing the dataset.” (lines 488-491)

“We first tested the impact that changing the number of retained connections would have on PLSc results.” (lines 506-507)

“We then assessed the impact of modifying our resampling approach, attempting instead to shuffle the original connectivity features before the feature selection step.” (lines 543-544)

“Finally, we used COMMIT to compare results after accounting for spurious streamlines against the original results.” (lines 557-558)

– Comparisons between univariate and multivariate approaches should be explained better in the results. I had to read it twice and then go to the Methods section to understand what was done there. The explanation in the Methods section is much better and cleaner than what is given in the Results section.

Thank you for pointing this out. We have modified this description in three important ways. First, as suggested, we have added a more detailed description of the results (lines 417-436, see modifications in response 1b). Second, to address this comment and one made below (Comment 4c), we have changed the contrast, using instead a KSADS diagnosis of ADHD. Third, using this different contrast, we felt it would be a stronger to select a multi-tract multi-symptom pair that was more specific to attention problems (PLSc1 – pair 7). That is why Figure 3 has been modified, illustrating now two univariate brain graphs (contrasting PC1 and PC2 scores as a function of ADHD diagnosis), and three multivariate brain graphs (PLSc1 – pair 1, PLSc2 – pair 3, and PLSc1 – pair 7). We hope that by providing more details as well as a new comparison of subgroups defined by a single diagnosis, we have made these results clearer.

3. The need to address potentially spurious connections– There are concerns that some of the connections represented in the 76x76 connectivity matrix used here are spurious connections due to limitations of tractography that cannot be resolved with current methods, especially as the streamlines derived in tractography come close to cortex. These limitations can affect results even when using the state of the art tractography algorithm that was used here, (as demonstrated in Girard et al., 2020, Neuroimage, 221: 117201).– The computational tractography arsenal includes several additional ways that could have been used to somewhat mitigate these concerns, including post-processing of the streamlines (e.g., with SIFT or LiFE), using surface-enhanced tractography methods (these may have been used here, but not mentioned) to increase the quality of tractography, and/or focusing on well-defined anatomical structures in the core of the white matter that can be well-detected using bundle recognition algorithms, and where tractography uncertainty due to the indeterminacy of the signal is of less concern. These issues seem particularly important given the substantial difference in the results between the 95% threshold and 100% threshold requirement for inclusion of an edge in the analysis– Regarding my comments about tractography methods, I think that improvements to the tractography methods would improve the confidence in the results that are presented here. At the very least, one way to deal with these concerns is to discuss them, and to acknowledge that the estimation procedures used here produce not "tracts" but rather the edges of a connectivity graph, and that these may or may not represent anatomical units.

We agree with the reviewer, it is at times difficult to exhaustively account for spurious connections, even with state-of-the-art techniques. To address this comment, we added to the manuscript’s methods section explanations on the limitations of the diffusion MRI techniques we used and those inherent to all tractography in general (lines 326-330). We also recognize in the manuscript’s discussion how the connections we have studied are not necessarily anatomical units, and that future studies will explore using well-recognized bundles (lines 704-711). Finally, we address in the manuscript how the multivariate approach we employed focuses less on the identity of individual tracts and more on patterns across tracts (lines 708-710).

In addition, to address your comment in a more quantitative manner, we implemented a post-tractography algorithm called COMMIT, which weighs streamlines by how much they explain the diffusion signal. This technique is known to remove false positive streamlines (https://pubmed.ncbi.nlm.nih.gov/25167548/). We defined spurious streamlines as those with COMMIT weights of 0. Given that we were not using streamlines per se but rather diffusion measures averaged across a connection, removing streamlines was not expected to have a large effect on the measures we used. However, when connections are composed of streamlines that are entirely spurious, COMMIT can lead to the removal of all its streamlines, and hence of the entire connection. Thus, we calculated how many connections were entirely removed by COMMIT. Using a criterion of 90%, to be consistent with our original methods, we found that 129 (65%) of the 200 originally-selected connections survived this correction. Multi-symptom features obtained using this subset of connections are highly similar to those obtained in the original analyses (compare Figures S3-S6 with the new supplemental Figure S14). Further, the correlations between the loadings of the corresponding multi-tract features for each PLSc, as well as the correlations between the loadings of the corresponding multi-symptom features for each PLSc were very high overall (the first 8 features had correlations ranging from.78 to 0.999), suggesting that the features were very similar. Further, the correlations between the expression of corresponding multi-tract features as well as the expression of multi-symptom features were also very high (the first 8 ranging from 0.78 to 0.999 as well), suggesting that the latent spaces were very similar between the two sets of analyses. These results suggest that even after accounting for spurious streamlines, PLSc results are highly consistent. These results have been outlined in a new section in the results entitled “Accounting for spurious streamlines”. Modifications to the manuscript related to this new analysis can be found in the methods (lines 330-340), results (lines 557-573), and discussion (lines 714-717).

Modifications:

“Most tractography techniques, including the one presently used, depend on propagating the local diffusion model across voxels. This approach has inherent limitations in voxels where the local model lacks the information necessary to inform on the appropriate path for streamline propagation, leading inevitably to the creation of spurious streamlines.^40,41^” (lines 326-330)

“To assess the impact of spurious connections on our results, we implemented an approach called Convex optimization modeling for microstructure informed tractography (COMMIT).^42^ This technique assigns weights to streamlines based on how they explain the diffusion signal. After running COMMIT, we identified streamlines with weights of 0, which we considered to be spurious streamlines. When all the streamlines of a connection had a weight of 0, the entire connection was considered spurious. We identified non-spurious connections and only kept those that were found to be non-spurious across 90% of participants. We then analyzed how many of the original 200 studied connections were found to be non-spurious. Finally, we reran our PLSc analyses on the subset of non-spurious connections and compared these results against those obtained from the original analyses.” (lines 330-340)

“Finally, we used COMMIT to compare results after accounting for spurious streamlines against the original results. We found that from the 200 studied connections, 129 (65%) were categorized as non-spurious. Multi-tract multi-symptom features obtained using this subset of non-spurious connections were highly similar to the original multivariate features (compare Figure S4 to Figure S14). Further, the correlations between the loadings of the corresponding multi-tract features for each PLSc, as well as the correlations between the loadings of the corresponding multi-symptom features for each PLSc were very high overall (the first 8 features had correlations ranging from.78 to 0.999), suggesting that these PLSc analyses were performing very similar transformations on the inputs. Further, the correlations between the expression of corresponding multi-tract features as well as the expression of multi-symptom features were also very high (the first 8 ranging from 0.78 to 0.999 as well), suggesting that the resulting features were very similar between the two sets of analyses. These results suggest that even after accounting for spurious streamlines, our PLSc results were highly consistent.” (lines 557-573)

“The connectivity features that were studied came from connectomes, not from well-known bundles. This choice was made to obtain a larger coverage of white matter structure, but as a result, sacrificed interpretability. The reason is that the connectivity features studied here may or may not represent true anatomical units, and hence cannot be interpreted easily on their own. Instead, we relied on patterns across tracts, such as counting the number of connections that implicated the corpus callosum in different multi-tract features. Future studies should contrast this approach against procedures to extract large well-known bundles.” (lines 704-711)

“For instance, the PLSc procedure was robust to variations in input data. This was evidenced by the consistent results after using a post-processing streamline filtering approach, as well as the similarities observed after running analyses on the replication set.” (lines 714-717)

4. Concerns with the analytic approach and stability across feature selection– Data were collected at multiple sites, and site differences should be considered. Authors say the scanner variable was regressed out from the connectivity data, which is fine, but it would be nice to see the site differences and the effect of regressing it out in a more systematic way.

Thank you for this suggestion, we agree that we should provide more details about site differences (which to be precise, are differences in scanner, given that some sites contained multiple scanners). We have included a figure in supplementary material (new Figure S1) illustrating the expression of multi-tract multi-symptom pair 1 with and without regressing out scanner. The figure also illustrates the weights of each diffusion measure for the principal components obtained after performing a principal component analysis on data that had been processed without regressing out scanner. Lastly, the figure also includes a barplot illustrating the average expression of multi-tract connectivity feature 1 across participants of a given scanner of before regressing out scanner. Average expression of this feature after regressing out scanner was not included given that, by definition, it was 0 for each scanner. In this figure, we highlighted the scanner with the highest average expression, and the one with the second-lowest (we did not illustrate the scanner with the lowest expression because we only had one participant with data from that scanner).

The plot illustrating the principal component weights demonstrates that regressing out scanner does not fundamentally alter the connectivity features (compare these weights with those shown in Figure S2). The barplots illustrate how different expression of multi-tract features can be as a function of scanner. Finally, the scatter plots illustrate how, without regressing out scanner, participants are clearly distinguishable in their expression of multi-tract connectivity feature 1 by their scanner, whereas after regressing out scanner, participants are no longer distinguishable by their scanner. These results are referenced in lines 207-208 and can be found in detail in Figure S1 and described in its caption.

Modifications:

“An illustration of the impact of regressing out scanner from connectivity data can be found in Figure S1.” (lines 207-208)

“Figure S1. Illustration of the effects of regressing out scanner. A. Weights of each diffusion measure for each principal component obtained after running a principal component analysis on data that was processed without regressing out scanner. B. Barplot illustrating the expression of multi-tract connectivity feature 1 averaged across all participants for each scanner. The blue bar illustrates the scanner with the highest multi-tract connectivity feature 1 expression, and the red bar illustrates the scanner with the second lowest multi-tract connectivity feature 1 expression (the scanner with the lowest expression only had one participant, so it was not chosen for illustrative purposes). C. Scatter plot illustrating expression of multi-tract multi-symptom pair 1 from the microstructural complexity PLSc using data that was processed without regressing out scanner. The blue dots illustrate participants from the scanner with the highest average multi-tract connectivity feature 1 expression, the red dots illustrate participants from the scanner with the second-lowest feature 1 expression. These two groups are distinguishable in their multi-tract connectivity feature 1 expression. D. Scatter plot illustrating expression of multi-tract multi-symptom pair 1 from the microstructural complexity PLSc using data where scanner had been regressed out. The same participants identified in scatter plot C are illustrated in scatter plot D. After regressing out scanner, these two groups are not distinguishable in their multi-tract connectivity feature 1 expression.” (lines 926-939)

– With the bootstrap analysis, how is it guaranteed that you get the same pairs each time? Please clarify.

Thank you for this question, it is indeed a technical subtlety that we did not explain in the original manuscript. The PLSc is performed using singular value decomposition (SVD). The pairs are ordered by the percentage of covariance in inputs that they explain. When inputs are shuffled, the obtained pairs are expected to be different. The only inherent order there is to the pairs is the percentage of covariance explained. Hence, that was the indicator we used to compare the original pairs against the pairs obtained from bootstrap and permutation testing. We have added a sentence specifying this in the methods section of the revised manuscript (lines 259-260).

Modifications:

“Although the pairs are expected to differ between iterations, they are always ordered by the percentage of covariance in inputs they explain.” (lines 259-260)

– I wonder whether the procedure used to assess the statistical significance of the multi-tract multi-symptom pairs (lines 463-467) doesn't unfairly favor the ones in the unshuffled data. Shouldn't this procedure also include the preliminary feature selection step (lines 427-430) also in the shuffled data? It seems like it should, because some of the correlation in the PLSc is due to these univariate correlations, which could arise by chance in high-dimensional data.

Thank you for this suggestion. We agree, the connections selected by the univariate feature selection stage could partly be due to chance given the high-dimensional data, and it is necessary to find a way of accounting for this potential issue. However, the optimal approach to account for this issue is not entirely clear. The preliminary feature selection step picks out the optimal connections for PLSc, given that they will already have the highest correlations with symptoms. Shuffling prior to the feature selection stage will lead to selection of connections that are more weakly correlated with symptoms. However, by including the feature selection step, the pipeline will still include an optimization stage that maximizes the relationship between connections and symptoms. Hence, shuffling in this way will lead to comparisons against null distributions with higher covariance explained. Stated more simply, this approach is stricter.

It is necessary to keep in mind that this permutation approach was not used to determine which pairs are “real” or “meaningful”. The original purpose for using it was: 1. As a heuristic to reduce the number of pairs to interpret (which in retrospect did not help much given the high number of factors retained); 2. To provide a more “objective” way of subselecting pairs, compared to the more traditional but more “subjective” approach of a scree test. Ultimately, none of these selection approaches guarantee that a pair is meaningful. The results of the permutation testing procedure also don’t ultimately impact the pairs themselves and hence the rest of the results.

Nonetheless, in response to this helpful comment, we have performed this analysis. No pairs were found to be significant when using this approach. We have presented the results of this analysis in a modified section on Sensitivity Analyses (lines 543-545).

Modification:

“We then assessed the impact of modifying our resampling approach, attempting instead to shuffle the original connectivity features before the feature selection step. Using this stricter approach yielded no significant multivariate pairs on permutation testing.” (lines 543-545)

– Uni vs multivariate 1 of 2. Is it any surprising or even interesting that the correlations between multi-tract and multi-symptom features were higher than all univariate correlations? With PLSc, latent modes are constructed (i.e., coefficients of tracts and symptoms are determined) to maximize the correlations. Thus, it is an expected outcome that those correlations would be higher than univariate correlations. Unless I am missing something, which is possible, I don't think this fact can be reported as a finding. Authors need to clarify this point, or revise the way in which these analyses and results are framed and interpreted.

We agree with the reviewer, it is not a surprising result. It is however a reassuring first indication that this PLSc is picking up something. But we agree that the manuscript in its original form placed too much emphasis on this admittedly trivial result, we have thus omitted it from the modified manuscript.

– Uni vs multivariate 2 of 2. I also have a few questions regarding comparisons of univariate and mulivariate correlations: Line 122 – 124 makes a statement comparing these values, but doesn't provide quantitative information.

Thank you for pointing this out. In response to the previous comment, we have removed the statement comparing univariate against multivariate correlation coefficients in the revised version of the manuscript.

Also, I might be misunderstanding, but don't the highest PLSc values have to be at least as high as the highest bi-variate correlation by construction?

To our knowledge, this does not necessarily have to be the case. PLSc is performed by applying a singular-value-decomposition to the covariance matrix between two sets of inputs. It is a fundamentally different operation than computing a correlation coefficient. More specifically, PLSc has a loss function that is based on a) internal covariation of each variable set and b) the strength of association with an outcome. As such, there are distinct sources of deviation that can arise when comparing ensuing association strength to other correlation approaches.

Are the differences between these values higher than would be expected by chance?

This question is interesting, but given that, as per the previous comment, the comparison between univariate vs multivariate correlations were removed, we have not tested this idea.

In Figure 3, the two univariate brain graphs (red and blue) look very similar to each other. Are they as similar as they seem there?

Thank you for catching this. The two brain graphs you are mentioning in Figure 4 looked identical because of a mistake in the code. We have fixed this issue and have made another important modification to the revised Figure 4. As stated in response to comment 1b, we have changed the contrast, comparing instead individuals with and without a KSADS diagnosis of ADHD. Hence, the red and blue brain graphs are contrasting the microstructural complexity and axonal density measures between children with and without ADHD diagnoses (using an α of 0.01 in the figure only to reduce the number of significant connections and improve the illustration). Additionally, we have changed one of the multi-tract features illustrated in the figure, showing now multi-tract feature 7 from the microstructural complexity PLSc. We selected this feature because its associated multi-symptom feature appears to be selectively representing attention problems.

Are the multi-tracts implicated by PLSc1 also similar to the multi-tracts implicated in PLSc2?

We performed our analytical pipeline (univariate feature selection -> PLSc) twice because we kept two principal components from our formed PCA model. With each of these two PCs, we performed the univariate feature selection twice. Hence, the connections that went into the two PLSc analyses are different. From the 200 connections selected for the first PLSc analysis, 48 (24%) were also selected in the second PLSc analysis. However, calculating the %Overlap measure to compare corresponding pairs in both PLSc analyses is not possible, given that this measure requires that the two brain graphs come from the same set. The expression of corresponding multi-tract features from both PLSc analyses showed correlations that ranged from weak (lowest was -0.04) to moderate (highest was 0.67). These results suggest that despite having mostly different inputs, some of the latent spaces derived from these two PLSc analyses share some similarities. We have not presented these results in the manuscript given that we used instead similar analyses on the replication dataset to assess how robust the PLSc procedure is to differences in inputs (see response 5a).

– Why 200 features? Would the results change substantially if you use 100? Is there some way to select a reasonably small number of features empirically (e.g., based on a scree plot?).

Thank you for this question. To quantify the impact of changing the number of features retained on the results, we successively took a larger number of connections from the list of top correlated connections, and plotted the % covariance explained as a function of the number of connections selected (starting from 19, the same number of symptoms). This analysis suggests that, in general, for the first latent factor, % covariance explained decreases with an increasing number of connections, increases again at around 80 features, and then stabilizes around 200 connections. These results suggest that some features included may be redundant, but that selecting 200 connections yields features that explain nearly the maximum amount of covariance possible across the different possible number of features that could be selected. These results are illustrated in the new Figure S12 and outlined in the Sensitivity Analyses paragraph of the results, lines 506-542.

To answer the first part of this question, our reasoning for selecting 200 connections was as follows: to extract connectivity features, we built connectomes using atlases, giving us several small putative connections, rather than full, well-known bundles. This decision was made because we believed that otherwise, we would lose “coverage”: a connectome built only from major bundles excludes several connections between other areas of the brain. However, unlike major bundles, connections from connectivity matrices are not necessarily individual anatomical units. Hence, to improve interpretation of multivariate connectivity features, we believed we needed to instead consider patterns of connections rather than attempt to interpret individual connections. One example of this approach in the manuscript was when we discussed how many of connections involved the corpus callosum in different brain graphs (Figure 4). If we had selected fewer connections, we reasoned that we would be sacrificing our capacity to make this type of interpretations. Hence, we aimed to include several connectivity features. However, a constraint of singular value decomposition (SVD) is that we require at least as many observations (e.g.: subjects) as we have features (e.g.: connections). The most features we could use is 214 (the number of participants in the discovery dataset). That is why we selected a number close to 214. However, we did not have a concrete reason for choosing 200 over 214 or other nearby numbers. Studies employ a number of different approaches to subselect features from a larger set. Two recent studies that informed our project used a correlation-based feature selection step like ours, and selected enough features so as to constitute 80% of their sample size (CITS). Our feature selection has a similar level of granularity (200/214 = 93%). The reasoning behind our choice of 200 features as well as the similar granularity employed in these two studies have been added to the revised methods section (lines 210-218).

To answer the second part of this question, unfortunately, there is no empirical way of selecting the number of features that go into the PLSc. A scree plot, or an equivalent approach, is a way of selecting the number of singular vectors (e.g.: multi-tract multi-symptom pairs/latent factors/principal components/canonical variates) by analyzing the successive singular values (% variance/covariance explained), not the number of features that go into the analysis. When comparing successive singular vectors within the same SVD, a scree approach can be used. But comparing singular values when using different features involves a comparison of singular values across different SVDs.

Modifications

“Feature selection. Given the constraints on the number of connectivity features that can be included in the partial least squares correlation (PLSc) analysis, we performed a univariate feature selection based on Pearson correlations. This solution is becoming increasingly adopted for high-dimensional variable sets (Figure 2D).^47,48^ We selected the 200 connectivity features most correlated with any symptom score, to maximize the number of features included. Given our discovery dataset size, selecting 200 connectivity features corresponded to 93% of our sample, a level of granularity comparable to other recent neuroimaging studies employing a feature selection step prior to multivariate analyses.^35,36^“(lines 210-218).

“We first tested the impact that changing the number of retained connections would have on PLSc results. The percentage of covariance explained by the first multi-tract multi-symptom pair from the microstructural complexity PLSc is illustrated in Figure S12. Initially, the percentage of covariance explained decreases with an increasing number of connections. However, starting around 80 connections, this percentage increases, and then stabilizes around 200 connections to a level close to its initial value. These results suggest that while some connectivity features included may be redundant, a selection of 200 connections is not far from the optimal amount that could have been selected.” (lines 506-542).

4. Concerns with the approach to psychopathology– One reviewer noted that: It is not the best practice to combine all possible psychiatric conditions within a single group and call it "adverse psychiatric outcome". This adversely affects the interpretability of findings. Knowing that one of the multi-tract features correlates with the adverse psychiatric outcome is not much informative and clinically valuable. Authors need to devise a more appropriate approach. At minimum, they need to provide a compelling motivation for the current approach.

Thank you for this suggestion, we agree that the measure we used was rather coarse. To address your comment, we selected instead a diagnosis of ADHD based on responses to the KSADS. We selected this diagnosis because its behavioural manifestations can be easily related to some of the presently-studied concussion symptoms (e.g.: attention problems), and because it was the second-most common diagnosis in our sample. We hope this chance improves the interpretability of the results. Please see our response to comment 1b for a list of relevant modifications.

– While the other reviewer noted that: Some comments regarding the participants that were classified as having psychopathology (marked in black in the scatter plots in Figure 2). It is not clear why these participants were selected based on the CBCL scores, which are indirect measures of psychopathology, rather than the more direct KSADS psychopathology scores? If there are 55% with psychopathological KSADS score, that should provide more than 28 participants marked here.

Thank you for this suggestion. In addressing the previous question, we have switched to using ADHD based on the KSADS questionnaire instead of CBCL scores.

I am also not sure that I agree with the statement that the 28 individuals highlighted in black would be grouped together in clinical studies, given their psychopathology probably varies widely. It is not clear to me whether the paragraph on lines 125 – 131 claims that these participants are different from the other participants in terms of their multi-tract/multi-symptom scores or are not different from the other participants. It seems to me that they might be different based on the first multi-tract/multi-symptom scores (which are both slightly higher in this group), but there might also not be enough subjects to demonstrate this difference statistically. I think that this would also be consistent with the subsequently-described overlap between this score and the univariate scores (lines 132-141).

Thank you for bringing up these points. In addressing the previous two comments, we have also addressed this one. We can say more confidently now, given the use of a single psychiatric diagnosis obtained from a gold-standard questionnaire, that this new subgroup of individuals with ADHD would indeed be more likely to be grouped together in clinical trials. We have also computed correlations between expression of multi-tract multi-symptom features and the presence/absence of an ADHD diagnosis. These correlations can be found in the modified Figure 3 scatter plots and in lines 421-423. Overall, these correlations are low to moderate, but statistically significant. The strength of these correlations does indeed follow the pattern that can be observed in Figure 4, whereby the overlap between univariate “ADHD-related” connections and multi-tract features was higher in pairs 1 and 7 than in pair 3. But qualitatively, there are still differences that can be observed in multi-tract multi-symptom feature expression among individuals with an ADHD diagnosis. That is the argument being made in this section, that differences in the expression of these features would be ignored in analyses that would group these individuals together.

Modifications:

“Further, we computed correlations between the expression of these multivariate features and clinical subgroup membership, and found weak, albeit significant correlations (see Figure 3 scatter plots).” (lines 421-423)

5. Concerns with claims of replication– The presentation of the results doesn't make it clear whether the structure that was identified in the discovery dataset does in fact replicate in the replication set.

Thank you for bringing up this point. If you are asking whether the same multivariate structure was obtained in the replication set, we did not determine this because originally, we did not rebuild a PLSc model from scratch on the replication set. Instead, we only projected the replication data onto the latent space that we had obtained from the discovery set. The challenge with redoing the PLSc on the replication set is that given that the sets have different sizes, we are inputting a different amount of connectivity features than in the original PLSc, which would unsurprisingly lead to some differences between the latent factors obtained. Nonetheless, to address your comment, we performed this analysis, redoing the feature selection step using 92 connections (the maximum number possible). We found that 20 of the original 200 connections were included in the 92 selected for this PLSc. The multi-symptom features are illustrated in Figure S11. Correlations between the loadings of these new multi-symptom features and the ones obtained from the original PLSc analyses were low to high (0.1 to 0.92). The correlations between the loadings of the 20 connections that were common to both PLSc analyses were very low to moderate (0.01 to 0.3). However, the correlations between expression of multi-symptom features were moderate to very high (0.3 to 0.99), and those between expression of multi-tract features were low to moderate (0.1 to 0.5), suggesting that despite having different inputs and hence different weights, the latent spaces obtained through these analyses share some similarities. Modifications to the manuscript relevant to these analyses can be found in a new section of the methods entitled *Analyses on replication dataset* (lines 312-324) and in a new section of the results entitled *Results on the replication dataset* (lines 484-504).

Modifications:

“To assess the robustness of our analyses, we first computed the percentage of connectivity/symptom covariance explained in the replication set by the first multi-tract multi-symptom pair of both PLSc analyses performed on the discovery set. We then selected, from the replication set, the same 200 connectivity features originally selected in the discovery set, and projected them, along with symptom features, onto the latent spaces obtained using the discovery set. To assess whether differences existed in multi-tract multi-symptom expression between participants from each set, we performed correlations comparing multi-tract multi-symptom feature expression against a binary variable indexing the dataset. Finally, we reran our feature selection procedure as well as the PLSc analyses on the discovery set, and compared the number of connectivity features that coincided in both analyses. We also performed correlations comparing the loadings of every corresponding multi-tract and multi-symptom feature, as well as the expression of these features.” (lines 312-324)

“We first analyzed the amount of connectivity/symptom covariance explained by each of the multi-tract/multi-symptom pairs. We found that approximately 12% of the connectivity/symptom covariance in the replication set was explained by the first multi-tract/multi-symptom pair from the microstructural complexity PLSc, and 33% was explained by the corresponding pair from the axonal density PLSc. We then projected the replication dataset onto the latent spaces obtained using the discovery set, and ran correlations comparing multi-tract multi-symptom feature expression against a binary variable indexing the dataset. We found very low, non-significant correlations (all coefficients lower than 0.01) between multi-tract and multi-symptom feature expression and set membership. Finally, we reran our PLSc analyses on the replication set, and compared the features obtained against the original results. Only 20 of the 200 connections originally selected in the discovery set were also selected in the replication set. Figure S11 illustrates the loadings for all the multi-symptom features retained after permutation testing. Correlations between the loadings of these new multi-symptom features and the ones obtained from the original PLSc analyses were low to high (0.1 to 0.92). The correlations between the loadings of the 20 connections that were common to both PLSc analyses were very low to moderate (0.01 to 0.3). However, the correlations between expression of multi-symptom features were better (0.3 to 0.99), which was also found for those between expression of multi-tract features (0.1 to 0.5). Altogether, these results suggest that despite having mostly different connectivity inputs due to the feature selection step, the analyses led to similar multi-tract multi-symptom features.” (lines 484-504).

Similarly, it hard to judge what to make of the differences in the results of the logistic regression between the samples. If I understand correctly, the logistic regression in the discovery sample indicates as predictive of psychopathology features that are different than the feature that was indicated in the discovery set. But this might lead us to wonder: How should we interpret this? For example, if I had a measurement from a new child that has had a concussion, which of the two models would I use to predict whether they might develop a psychopathology? The one derived in the discovery set or the one remaining feature that was indicated in the discovery set? Authors need to clearly explain the logic and evidence that supports their claim.

Thank you for raising this point. After much consideration, we have decided to remove the section on psychiatric diagnosis prediction from the manuscript, for two important reasons. First, we believe attempting to demonstrate a clinical applicability is rushed, given that, as we stated before, this manuscript is presenting a novel way of looking at concussions, not an optimized analytical pipeline. To achieve a practical outcome, the methods require optimization. Second, given that our data is cross-sectional, we cannot state for certain that our models are predicting outcomes.

– Regarding my comments about the comparison to the replication set, I think that a few direct comparisons might help. For example, how much of the covariance in the discovery set is explained by the same multi-tract/multi-symptom pair that explained more than 40% of the covariance in the microstructural complexity PLSc in the discovery set? Or, possibly even more rigorous: if you ran a PLSc on the discovery set how similar would the feature sets be? I realize that this might not be possible, given the smaller sample size and the choice of 200 features in the feature-selection phase.

Thank you for these excellent suggestions. We performed the analysis looking at connectivity/symptom covariance explained in the replication set using multi-tract/multi-symptom pairs. We found that approximately 12% of the connectivity/symptom covariance in the replication set was explained by the first multi-tract/multi-symptom pair from the microstructural complexity PLSc, and 33% was explained by the corresponding pair from the axonal density PLSc. We also performed additional comparisons using the replication set. As mentioned above in response to comment 5a, we performed PLSc on the replication set and compared the resulting multivariate features against those obtained with the discovery set. These features had differences compared to the original ones, which were expected given the different inputs, but they also revealed some similarities. In addition, we projected the data from the replication set onto the latent spaces obtained from the discovery set, and we calculated correlations between the expression of multi-tract and multi-symptom features and a binary variable for the set. We found these correlations to be low and non-significant, suggesting that there are no measurable differences in feature expression in the two sets. Altogether, these results suggest that the feature selection step is less robust than we initially thought, but the PLSc step is robust, such that even with different inputs, the features obtained share similarities. These analyses are described in a new section of the Methods entitled *Analyses on the replication dataset*, and a new section of the results entitled *Results on the replication dataset*. Please see our response to comment 5a for the relevant modifications.

6. The need for a more sober and precise Discussion.– Authors say in the discussion that "these results recapitulated well-known findings from the concussion literature and revealed new insights…". Yet it is not clear what those well-known findings are and what new insights were revealed.

Thank you for pointing this out. By well-known findings, we meant the central role of the corpus callosum, a result that is reported in the literature with high frequency. By new insights we mainly meant two things: 1. The possibility that the corpus callosum might be identified so frequently because it is a structure that is implicated in all concussions, but it might not be symptom-specific; 2. The idea that concussions can be seen as a combination of a variety of white matter structure alterations with different relationships to combinations of a variety of symptoms. We have made these specifications in the Discussion (see lines 590-603).

Modifications:

“In the present study we leveraged novel dMRI methods and a double-multivariate approach to parse heterogeneity in the relationship between white matter structure and symptoms in a large sample of previously-concussed children. By applying PLSc on biologically-interpretable measures of dMRI obtained from PCA, we found cross-demographic multi-tract multi-symptom features that captured information about structure/symptom relationships that traditional approaches missed. The most consistent multi-tract multi-symptom pairs implicated several callosal tracts, recapitulating a well-known finding from the concussion literature. However, the present results suggest the involvement of these callosal tracts may be more reflective of general problems than specific symptoms. Instead, specific symptom combinations may be related to combinations of different white matter alterations. These results thus reveal new insights about white matter structure/symptom relationships.” (lines 590-603)

– Overall, the narrative in the Results section would benefit from a better organization.

Thank you for this suggestion, in an effort to respond to a similar comment above, we have moved the Results section to appear after the Methods, and have added small sentences recapitulating the analyses performed, hopefully improving readability and organization of the manuscript.

– Authors should consider reporting more quantitative results, including tables to summarize findings, and precise comparisons of their results with published work.

Thank you for this suggestion. To address this comment and previous comments made by the other reviewer, we have aimed to increase the amount of quantitative results we present throughout the entire manuscript. For example, when assessing whether the expression of multi-tract multi-symptom features cut across strata defined by “external” variables, such as diagnoses (e.g.: ADHD) and socioeconomic variables (sex, household income, race/ethnicity), we have computed correlations between expression of these features and these variables. Examples of these analyses can be found in our response to comment 1b, as well as Figure S2. These simple yet clear comparisons allowed us to demonstrate more quantitatively that the expression of multi-tract and multi-symptom features is not strongly related to these “external” variables. The revised manuscripts contains several other additional quantitative results, as well as more comparisons to published work (e.g.: lines 215-218, or lines 629-633). We hope these modifications increase clarity in the results.

Modifications:

“Given our discovery dataset size, selecting 200 connectivity features corresponded to 93% of our sample, a level of granularity comparable to other recent neuroimaging studies employing a feature selection step prior to multivariate analyses.^34,35^” (lines 215-218)

“Instead, a multi-tract multi-symptom pair that more uniquely represented attention problems implicated fewer callosal tracts. This pair implicated right-lateralized connections, consistent with prior literature showing more left-right asymmetry in FA in children with ADHD compared to controls.^48,49^” (lines 629-633)

[Editors' note: further revisions were suggested prior to acceptance, as described below.]

Both Reviewers expressed some enthusiasm for the revision, noting that"I think the authors did a great job of reorganizing the manuscript and addressing almost all issues raised by reviewers," and"The authors have made substantial changes to the manuscript in their revision and extensive additional analysis has been added in response to the comments made on the previous version. I believe that many of these changes help improve the article, and could provide some more clarity about the robustness of the method proposed and its application. In particular, new analyses that describe the robustness of the approach, and new comparisons with the replication sample could help establish confidence in the conclusions."Nevertheless, there was consensus that additional major revisions are required, as detailed belowAbstract– In the abstract, results are limited to findings in ADHD and the replication set, but both are rather secondary/supporting results. There is virtually nothing about the main findings. How do certain combinations of symptoms relate to certain combinations of tracts? I would give specific results in the abstract with the highest importance, possibly mentioning specific symptom-tract couples, rather than sharing very generic comments, such as "These clinically-informative, cross demographic multi-tract multi-symptom relationships recapitulated well-known findings…" Such text is more suitable for the Conclusions.

Thank you for pointing this out. We have modified the abstract’s results to highlight specific multi-tract multi-symptom pairs:

“Early multi-tract multi-symptom pairs explained the most covariance and represented broad symptom categories, such as a general problems pair, or a pair representing all cognitive symptoms, and implicated more distributed networks of white matter tracts. Further pairs represented more specific symptom combinations, such as a pair representing attention problems exclusively, and were associated with more localized white matter abnormalities. Symptom representation was not systematically related to tract representation across pairs. Sleep problems were implicated across most pairs, but were related to different connections across these pairs.” (lines 34-41)

– The current Conclusions in the abstract is again extremely generic and does not fit well the findings. The manuscript does not include much on "stratification strategies" or "predictive biomarkers". It would be better to discuss how specific findings of the current study would fuel future studies.

We agree with the reviewer, these statements were not closely fitting with the results. We have modified the abstract’s conclusion accordingly:

“Using a double-multivariate approach, we identified clinically-informative, cross-demographic multi-tract multi-symptom relationships. These results suggest that rather than clear one-to-one symptom-connectivity disturbances, concussions may be characterized by subtypes of symptom/connectivity relationships. The symptom/connectivity relationships identified in multi-tract multi-symptom pairs were not apparent in single-tract/single-symptom analyses. Future studies aiming to better understand connectivity/symptom relationships should take into account multi-tract multi-symptom heterogeneity.” (lines 45-52)

Abstract/Introduction/Results/Discussion– Overall, the main goal (not methodological but rather scientific) of the study should be clearer, both in the abstract and other sections (especially, Results). It should be clear for the reader what they learn new about concussion/TBI and how those findings can be used in practice (e.g., in research, in clinics, …).

Thank you for this suggestion. To address your comment, we have made additions in the abstract (lines 45-52), the results (lines 391-393, 447-462), and discussion (lines 634-637, 711-714).

“Using a double-multivariate approach, we identified clinically-informative, cross-demographic multi-tract multi-symptom relationships. These results suggest that rather than clear one-to-one symptom-connectivity disturbances, concussions may be characterized by subtypes of symptom/connectivity relationships. The symptom/connectivity relationships identified in multi-tract multi-symptom pairs were not apparent in single-tract/single-symptom analyses. Future studies aiming to better understand connectivity/symptom relationships should take into account multi-tract multi-symptom heterogeneity.” (lines 45-52)

“To parse multi-tract multi-symptom heterogeneity, we performed two PLSc analyses, one using the selected microstructural complexity features and another using the selected axonal density features, along with all 19 symptom features.” (lines 391-393)

“Overall, these results illustrate how different symptom profiles are associated with different combinations of tracts. Earlier pairs consisted of broad symptom categories and implicated wider networks of connectivity features, whereas more idiosyncratic pairs consisted of more symptom-specific combinations of connectivity features were associated with more symptom-specific profiles. However, some symptoms such as sleep problems were implicated across the spectrum of different multi-tract multi-symptom pairs, illustrating how some symptoms do not demonstrate a one-to-one relationship with connectivity features across multi-tract multi-symptom pairs. No tracts were widely implicated across all pairs.” (lines 447-462)

“These results suggest that rather than a clean and consistent set of one-to-one symptom/tract relationships, concussions may instead be composed of different symptom combinations that are associated with combinations of structural alterations of different white matter tracts.” (lines 634-637)

“These prior studies, using a variety of approaches, have all focused on parsing down inter-subject heterogeneity in symptoms or white matter structure. Our approach focuses instead on disease-specific heterogeneity in structure/symptom relationships.” (lines 711-714)

– The new focus on ADHD is reasonable, but I would have liked to see more treatment in the manuscript of the relationship between ADHD and concussion and what we learn from the current analysis (if at all) about this relationship. As it is right now, the choice to analyze ADHD is not strongly motivated by the Introduction, and the interpretation of the results does not clearly help us understand what we learned here, beyond the fact that manifestations of concussion in participants with ADHD are heterogeneous.

Thank you for your suggestion. The focus on ADHD was driven by the data (given that it was one of the most common diagnoses in our dataset), and by practicality (given that ADHD has high levels of attention problems, which happened to be one of our symptom scores). We added a brief sentence in the Introduction to introduce the idea of studying ADHD (lines 59-61), and we expanded on the discussion of the results implicating ADHD (lines 689-692).

“Concussion afflicts approximately 600 per 100,000 individuals every year.^1^ It is associated with several psychiatric conditions, such as post-traumatic stress disorder and attention-deficit/hyperactivity disorder (ADHD).” (lines 59-61)

“Nonetheless, children with ADHD were heterogeneous in the expression of this more attention-specific multi-tract multi-symptom pair, suggesting that this clinical subgroup of children with TBIs may have important differences that can be further investigated.” (lines 689-692)

Results/Discussion– The text in the introduction (page 3, lines 87-96), nicely summarizes the main contribution: "… display a common symptom but will have overall different symptom profiles and different white matter damage profiles… a variety of white matter structure alterations ("multi-tract") may be related to a variety of symptoms ("multi-symptoms") in different ways." This is very clear and interesting. What is missing is that results do not demonstrate this interesting phenomenon clearly. There is one example in Discussions (page 21, lines 600-6004), but it is limited to ADHD symptoms. It would be great if more results/discussions were included about specific symptoms and how they are associated with specific tracts. If possible, talk more about symptom-tract pairs; specifically showing how same/similar symptoms might be related to different tracts. Or how different combinations of symptoms are linked to different tract combinations. This is the most important part of the manuscript, and the results should reflect this fact.

Thank you for this excellent suggestion. Your comment allowed us to take another look at our results, which allowed us to notice interesting patterns that we had missed the first time. We have modified the manuscript to focus more on this specific set of results (lines 398-462).

“Appendix 1 Figures 4 and 5 illustrate all the multi-symptom and multi-tract features (respectively) from the retained pairs from the microstructural complexity PLSc. Individual pairs selected for further discussion are shown in Figures 3 and 4. Figure 3 also illustrates the expression of three multi-tract multi-symptom pairs (scatter plots). For each pair, the symptom profiles of two example participants, one with high feature expression, one with low, are shown. These example participants illustrate how these multi-tract multi-symptom features can represent a diversity of symptom profiles.

Across most extracted pairs, the representation of tracts and symptoms formed a continuum, with earlier pairs capturing broader symptom categories and more distributed networks of connections, and later pairs capturing more idiosyncratic symptom/connectivity relationships. The first multi-tract multi-symptom pair from the microstructural complexity PLSc broadly represented most symptoms (Figure 3A polar plot) and implicated a broad range of frontal commissural and occipito-temporal association tracts (Figure 4, violet brain graph). The third multi-tract multi-symptom pair obtained from the axonal density PLSc represented broadly cognitive problems (Figure 3B polar plot). The multi-symptom feature from the third pair obtained from the microstructural complexity PLSc also represented cognitive problems broadly and implicated a wide array of tracts with a mostly frontal focus, whereas features from subsequent pairs represented individual cognitive problems, such as feature 8 which represented processing speed, executive function (card sorting), and working memory, and implicated almost exclusively frontal tracts. The seventh pair obtained from the same PLSc represented attention problems almost exclusively, along with decreased sleep and processing speed (Figure 3C polar plot). This pair implicated mostly frontal tracts, including a connection between the left posterior cingulate and left thalamus, a trajectory that is consistent with the corticospinal tract. This pattern whereby pairs ranged from broadly representing symptom categories and distributed networks to more specific symptom combinations with more localized connections can be best appreciated in Figure 6. As can be observed, certain groups of connections tended to be represented only once alongside broad symptom categories. More consistent connectivity/symptom correspondences were only observed for few, more specific single-symptom/single-connection combinations.

Although this pattern was observed in both PLSc analyses, important exceptions were observed as well. First, the second multi-tract multi-symptom pair obtained from both PLSc analyses strongly represented nausea and vomiting, almost exclusively, and implicated no commissural tracts. Second, sleep problems (especially “trouble sleeping”) were implicated across several pairs. Interestingly, despite being found ubiquitously across pairs, they were not consistently associated with the same connections across pairs. In contrast, nearly every time attention problems were implicated in a pair (3/4 pairs), they were found alongside two connections with trajectories that correspond to parts of the right superior longitudinal fasciculus (right pars opercularis – right post-central sulcus; right par opercularis – right sumpramarginal gyrus). However, this type of consistent symptom/connection correspondence was more often than not absent (Figure 5).

Out of 200 connections selected for the microstructural complexity PLSc, 2 were found to be most frequently implicated across all retained pairs. These were the two connections mentioned above that were related to attention problems, the right pars opercularis – right post-central gyrus, and the right pars opercularis to the right supramarginal gyrus. They were implicated in 4 pairs each, two of which were the same (pair 1 and 8). Callosal tracts were not among the most often implicated connections.

Overall, these results illustrate how different symptom profiles are associated with different combinations of tracts. Earlier pairs consisted of broad symptom categories and implicated wider networks of connectivity features, whereas more idiosyncratic pairs consisted of more symptom-specific combinations of connectivity features were associated with more symptom-specific profiles. However, some symptoms such as sleep problems were implicated across the spectrum of different multi-tract multi-symptom pairs, illustrating how some symptoms do not demonstrate a one-to-one relationship with connectivity features across multi-tract multi-symptom pairs. No tracts were widely implicated across all pairs.” (lines 398-462)

Answering this question also allowed us to create a new figure (Figure 5 shown below) which we believe more clearly illustrates the relationships between symptoms and connections. In addressing your comment, we have also come up with a new way of interpreting the findings that we believe is more clearly conveys the main message of the manuscript (lines 638-645).

“Defining concussions as a clinical syndrome characterized by a set of symptoms stemming from a set of alterations of brain structure and function, multi-tract multi-symptom pairs can be thought of as subtypes of concussion (i.e.: of structure/symptom relationships), not of concussion patients. Patients with concussions can express these different concussion subtypes to varying degrees. We theorize that the combination of these different subtypes and how much they are expressed is what determines the clinical syndrome a person will display. The pairs explaining the most covariance can be interpreted as the subtypes that are most commonly expressed across individuals.” (lines 638-645)

– The inclusion of new control analysis that uses COMMIT to identify spurious connections is great. I think that the added text on lines 666 – 690 raise important caveats and future directions, which will give the readers not as familiar with the details of dMRI research important context to understand the work. On the other hand, it is very hard to judge the similarity between the results in Appendix 1 Figure 4 and Appendix 1 Figure 14. In particular, there seem to be no black stars in Figure 14 (despite the caption mentioning them). Does this mean that no MCF symptom features are significant in this analysis when COMMIT is used? That would be a real problem for the robustness of this approach to issues related to tractography that are mentioned in this part of the discussion.

Thank you for your feedback on this addition and thank you also for your question. That was a mistake on our part, we have now corrected the figure to illustrate the symptoms that significantly contributed to each multi-symptom feature.

– Figure 3 (and the comparison between univariate and multivariate measures) is still hard to understand.A) The authors stated in the response to the reviews that "Figure 3 has been modified, illustrating now two univariate brain graphs (contrasting PC1 and PC2 scores as a function of ADHD diagnosis), and three multivariate brain graphs (PLSc1 – pair 1, PLSc2 – pair 3, and PLSc1 – pair 7)." I see the latter, but not the former.

Thank you for pointing this out. This was actually a typo on our part in the response to the reviewer’s previous comment. We had meant to refer to Figure 4, the line graph that contains the brain graphs.

Figure 3 also has two correlation coefficients inset in each plot, (e.g., "r=0.38" and "r=0.26" in the right-hand side of the first row). What do these coefficients refer to?

Thank you for asking this question, in answering it we realized we did not include a description of these coefficients in the Figure 3 caption. We have now added it. To answer your question, these coefficients are from a Pearson correlation between expression of a multi-symptom feature (near Y axis), or a multi-tract feature (near X axis), and a binary variable indexing whether or not each participant had an ADHD diagnosis. The further away from 0, the stronger the correlation between the multivariate feature expression and whether or not participants had an ADHD diagnosis. In other words, the more individuals with ADHD differ from non-ADHD individuals in their expression of a multivariate feature, the higher the coefficient.

“Further, we computed correlations between the expression of these multivariate features and clinical subgroup membership, and found weak, albeit significant correlations (see Figure 3 scatter plots).” (lines 468-470)

“Correlation coefficients inset in each scatter plot represent Pearson correlations between expression of multi-tract features (near x-axis), or multi-symptom features (near y-axis) and a binary variable indexing whether or not a participant had a diagnosis of ADHD.” (lines 499-502)

B) Relatedly, the methods say (starting on line 279) that "Using threshold of p<0.05, we computed univariate comparisons of connectivity (PC scores) between individuals with and those without a diagnosis of ADHD thus identifying putative "ADHD-related" univariate connectivity features." This sounds reasonable, but I don't see the conclusion from this analysis in the Results section. Is there a univariate difference between the PC scores of individuals diagnosed with ADHD and those that were not? Was this comparison conducted only within the group that had a psychopathological KSADS scores? In other words, on line 273 in the phrase "We divided the sample", does the word "sample" here refer only to (how many?) participants who had any psychopathology? Is only their data presented in Figure 3? Or is that all 214 of the discovery set participants?

The significant univariate differences between PC scores of individuals with ADHD and those without are the red and blue brain graphs illustrated at the top of Figure 4. These comparisons were performed across the 214 individuals in the discovery dataset, which is what we meant by “sample”. The Figure 3 scatter plots illustrate data for all the 214 participants in the discovery set. We did not provide a detailed summary of these hundreds of univariate tests, but instead focused on how many of the connections identified as significant in these univariate tests were also implicated in the multi-tract features. These were the results presented in Figure 4. The manuscript has been modified to increase clarify regarding this analysis.

“To compare information captured by the PLSc and univariate approaches, we first identified, among the 214 participants from the discovery set, those that had obtained a psychiatric diagnosis.” (lines 290-292)

“We identified 33 individuals in the discovery set with diagnoses of ADHD obtained from the KSADS.” (lines 464-465)

C) Furthermore, I am still struggling to understand whether I should interpret Figure 3 as indicating a difference between the subjects marked in black (now subjects diagnosed with ADHD) and the other subjects in the multivariate analysis. As far as I can tell, these individuals (black dots in the scatter plots) are not different in any discernible way from the other participants in these plots (colored dots) -- certainly not in pair 3 (the middle line), where the black dots and colored dots seem completely intermingled. Is there some way to evaluate the difference between the groups across all of the multi-symptom/multi-tract features?

Thank you for bringing up these points, ensuring their clarity is vital for the manuscript. The objective of Figure 3 is to illustrate that individuals who would be considered “similar”, or considered as belonging to the same clinical subgroup due to their sharing of a psychiatric diagnosis, are spread out in these multi-tract multi-symptom “spaces”. Consider the opposite scenario: imagine that individuals with ADHD and those without were perfectly split in these multivariate spaces. In such a scenario, this multivariate approach would be trivial, since we could as easily discern individuals with ADHD from those without based on their diagnosis. But in these multivariate spaces, individuals with ADHD are spread out, and that is what we meant by “discernible”. But we understand that calling these individuals “discernible” made it sound as if we meant “discernible from non-ADHD individuals”. What we meant to say is that individuals with ADHD are discernible from each other in multivariate space. We have followed the excellent suggestion made in comment 7f to better reflect this point (lines 466-468).

Despite forming a clinical subgroup based on a gold-standard measure of psychiatric pathology, these individuals were heterogeneous in their expression of multi-tract and multi-symptom features. (lines 466-468)

To answer your second question, what we did to evaluate the difference between the groups across specific multi-tract multi-symptom pairs was to compute correlations between expression of multi-tract features, or multi-symptom features, and a binary variable indexing whether or not individuals have an ADHD diagnosis (please see our response to question 7b). These analyses were what led to the coefficients illustrated in the Figure 3 scatter plots. To address your comment however, we have computed these correlations across all retained multi-tract multi-symptom features, and have displayed this information in a new table in the supplementary material (Supplementary File 3) (line 470).

D) In the same section, what is the statistical test that supports the statement "Further, we computed correlations between the expression of these multivariate features and clinical subgroup membership, and found weak, albeit significant correlations"? It would be good to apply such a statistical test to all of the multi-tract multi-symptom features, and report the results of this test as part of this section, just as it would be good to report the results of a direct test of the "univariate" PCA differences. I believe that this would provide a stronger assessment of multivariate vs. univariate approaches.

Thank you for this suggestion. What we computed were Pearson correlations between the expression of multi-tract and multi-symptom features and a binary variable indexing clinical subgroup membership. These are the correlations described in the response to the previous comment (7d). As you suggested, we have now computed these correlations across all multi-tract multi-symptom features, and have reported the results in a new table entitled Supplementary File 3. The results of univariate PCA differences were reported in the form of the red and blue brain graphs illustrated at the top of Figure 4. Given the large amount of connections identified, we thought it would not be clear to provide a list of significant connections, which is why we opted to illustrate these connections in brain graphs. Further, the individual connections identified in these univariate analyses were not of particular interest. What was of interest was how many of these connections were also implicated in our multi-tract features. These are the results we presented in the Figure 4 line graph.

– This might just be a wording issue, but I don't understand the following statement (L413- 416): "Despite forming a clinical subgroup based on a gold-standard measure of psychiatric pathology, these individuals were differentiable by their expression of multi-tract and multi-symptom features". Maybe the problem is the word "differentiable"? If this is really what you meant, may I suggest something a bit more direct, such as "…, these individuals were heterogeneous in their expression of multi-tract and multi-symptom features"?

Thank you for this excellent suggestion. We believe your suggestion captures exactly what we wanted to express, so we have modified the sentence accordingly (lines 466-468).

“Despite forming a clinical subgroup based on a gold-standard measure of psychiatric pathology, these individuals were heterogeneous in their expression of multi-tract and multi-symptom features.” (lines 466-468)

– The analysis on the replication set is much improved in the revision, but questions still remain here as well. First, are 12% and 33% of the covariance that were explained in the replication set, using MCF1 in both PLSc analysis more than would be expected by chance? I think that this could be established through a permutation test in the replication set or some other statistical test.

Thank you for your comment. We followed your suggestion and performed permutation testing, using 2000 permutations and shuffling at each iteration the subject labels in the connection dataset and recalculated the percent covariance explained in the shuffled replication dataset by the first multi-tract multi-symptom pair. The percentages of covariance explained (which were recalculated after COMMIT filtering (see comment 7i)), were not significantly higher than chance. These results were reported in lines 521-526.

“We found that approximately 26% of the connectivity/symptom covariance in the replication set was explained by the first multi-tract/multi-symptom pair from the microstructural complexity PLSc, and 40% was explained by the corresponding pair from the axonal density PLSc. However, with permutation testing, these percentages were not found to be significantly higher than expected by chance.” (lines 521-526)

Another question that arises from the comparison is that it looks as though the first set of symptom weights obtained in the PLSc for microstructural complexity in the replication set (Y_1 in Appendix 1, Figure 11) is completely inverted relative to the set of weights in this PLSc conducted on the discovery set. Is this anti-correlation somehow trivial? Or is it significant (either statistically or conceptually)?

Thank you for your question, we understand how this might appear confusing. When performing matrix decomposition techniques such as PCA and PLSc, what we are doing is finding linear combinations of variables, which define new “spaces” (or axes). The direction of these axes is trivial, as long as the relationship between the weights is maintained (so whatever was positive became negative and whatever was negative became positive). This property is called reflection invariance.

What do we learn from the fact that the features selected in the discovery and replication sets are quite different? Does performing the COMMIT analysis in both the discovery and replication set improve the situation? Surely the feature selection step is affected by the 35% of connections that COMMIT finds to be spurious in the discovery set? Why not remove them with COMMIT and proceed from there? I realize that you can't try all the different permutations of analysis pipelines, but using COMMIT seems like a principled way to improve the robustness specifically of this part of the pipeline.

Thank you for these excellent questions, these are all important discussion points. First, the fact that the features selected in the discovery and replication sets are quite different suggests to us that the feature selection step was less robust than we initially believed. However, despite these differences, the results comparing the loadings and the expression of multi-tract multi-symptom features suggests that after the PLSc analyses, these features share some similarities, ultimately suggesting that the PLSc procedure is quite robust.

In our previous resubmission, we demonstrated how the multi-tract and multi-symptom features obtained from PLSc after performing COMMIT are highly similar (~0.99 correlation) to the features obtained from PLSc without performing COMMIT, highlighting once again the robustness of the PLSc approach. However, we agree that using COMMIT, or another streamline filtering approach, is a principled way to improve robustness of the pipeline. It is certainly our intention that future studies inspired by ours consider doing a filtering approach, which is why we have decided to follow your advice and repeat all analyses using the COMMIT filtered data. To do so, we modified the analytical pipeline, by performing COMMIT after the post-processing stage (i.e.: after thresholding and imputation), *before* the PCA (see modifications to Figure 2 and its caption, found below, and the modifications to the methods, where the COMMIT section now follows the section on *Additional Data Transformations* and precedes the section on PCA). This meant that the PCA and all following steps were performed on the COMMIT filtered connections.

To answer your second question, redoing the univariate feature selection step on the replication set with the COMMIT-filtered connections, 73 of the original 200 connections selected in the discovery set were selected. This represents a major improvement in robustness, even for this feature selection step (compared to the 20/200 connections that were selected in the previous iteration). Since we have modified the analytical pipeline though and since testing the impact of COMMIT on our pipeline was not part of the main research questions, we did not report this comparison between the results of this feature selection step on COMMIT-filtered connections and on non-filtered connections. We only reported the results on COMMIT-filtered connections.

Given that all analyses were performed again, a detailed summary comparing these new results with the prior ones is necessary.

– Results from the PCA, despite including now only COMMIT-filtered connections, are nearly identical.

– Early multi-symptom features are nearly identical to the previous iteration of the analytical pipeline (see Author response image 1, with the new set of multi-symptom features from the microstructural complexity PLSc shown in Figure 4, and the same features but from the previous, non-COMMIT-filtered pipeline shown on the right). However, fewer multi-tract multi-symptom pairs were found to be significant in this iteration of the pipeline.

**Author response image 1. sa2fig1:** 

– Expression of the selected multi-tract multi-symptom features in Figure 3 is nearly identical.– Correlations between multi-tract and multi-symptom feature expression and a variable indexing ADHD were nearly identical (see Figure 3).

– Percent overlap scores are different, although the overall interpretations are the same: multi-tract feature 1 continues to be the one with the highest overlap with the univariate comparisons, and multi-tract feature 7, which continues to be specific for attention problems, has a lower percent overlap score with these univariate comparisons (see Figure 4).

– The validation analysis where we tested the impact of using a different number of features shows a different pattern than in the first iteration of this pipeline. Now, the % covariance explained for the first multi-tract multi-symptom pair does not have an initial drop before increasing again. Instead, it starts off low, and increases with increasing numbers of connections, suggesting that the choice of 200 connections was nearly optimal (see figure 11).

– The focus on the corpus callosum was dropped. The reason was that we were drawing conclusions about the role of the corpus callosum based on which multi-tract features implicated more callosal tracts. However, the total numbers were so low across pairs (e.g.: mult-tract feature 7 implicated 2 callosal tracts, but 10 overall), that we felt this conclusion was not very strong. The new version of the manuscript focuses more on symptom/connection correspondences as per earlier suggestions.

– Using the COMMIT-filtered data, the results with the T=85%, 90%, and 95% thresholds are identical, because this thresholding step was less severe than COMMIT. However, encouragingly, none of the connections that were lost to this thresholding were connections that would have passed COMMIT. The only difference was found with the T=100% threshold. However, even with this threshold, only 6 connections that would have survived COMMIT were lost due to this thresholding. These results and their implications can be found in lines 560-568.

“All 629 connections that had survived COMMIT in the original iteration (t=90%) were among the 1142 connections that survived the t=85% threshold, and the 877 connections that survived the t=95% threshold. Hence, after COMMIT, the same 629 connections were selected in all three thresholds, which led to identical data going into all subsequent analyses. Differences were only seen for t=100%, where 258 connections survived thresholding. After COMMIT, 252 connections survived, suggesting that only 6 connections that were considered “spurious” were found across 100% of participants. These results suggest that highly consistent connections tended to be the ones found by COMMIT to be “non-spurious”.” (lines 560-568)

– All other results are nearly identical to the ones presented in the previous iteration of the analysis.

I also think that the probe for correlation between expression of the multi-symptom multi-tract features and dataset is clever, but some of the explanations that now appear in the response to the reviews (and particularly about this point) should probably appear in the Discussion section, to help readers understand what the point of this analysis is. In general, I wonder whether, now that lines 618 – 626 have been removed, the authors could discuss the implications of the analysis of the replication set in the Discussion. I think that having a replication set analysis is a real strength of the paper.

Thank you for this suggestion. We expanded the section of the Discussion where the implications of the analyses on the replication set were mentioned to include these correlation analyses.

“As illustrated by the sensitivity analyses and analyses performed on the replication set, some parts of the analytical pipeline were robust to methodological variations whereas others were not. For instance, the PLSc procedure was robust to variations in input data. This was evidenced by the consistent results after using a post-processing streamline filtering approach, the non-significant correlations between multi-tract multi-symptom feature expressions and a variable indexing the dataset, as well as the similarities observed after running the PLSc analyses on the replication set.” (lines 743-749)

Discussion– While it is discussed in Discussions how a multivariate approach better captured ADHD-related symptoms than a univariate approach, this is not clear in Results. When I first read it in the discussions, I was surprised but then remembered the parts in the results (scattered all around) that supported this claim. I would make this comparison clearer under the "Results / Multivariate vs Univariate Approaches" section.

Thank you for this suggestion. We have modified the manuscript to improve the clarity of the results supporting this conclusion (lines 476-487).

“Notably, the highest overlap occurred with multi-tract connectivity feature 1 (10-13%) from both PLSc analyses, which implicated a wide network of white matter tracts and were associated with general problems. In contrast, the overlap with multi-tract connectivity feature 7, which implicated mostly frontal connections and was associated with attention problems almost exclusively, was low (5%). Neither of the two univariate analyses implicated the two connections discussed above (right pars opercularis – right post-central sulcus; right par opercularis – right sumpramarginal gyrus) that were consistently associated with attention problems. These results suggest that the putative “ADHD-related” connections identified in univariate comparisons of microstructural complexity and axonal density measures between individuals with ADHD and those without are mostly non-overlapping with the connections identified in an attention-problems specific multi-tract multi-symptom pair.” (lines 476-487)